# Learning with Explanation Constraints

**Rattana Pukdee**[*]
Carnegie Mellon University
rpukdee@cs.cmu.edu

**Dylan Sam**[*]
Carnegie Mellon University
dylansam@andrew.cmu.edu

**J. Zico Kolter**
Carnegie Mellon University
Bosch Center for AI
zkolter@cs.cmu.edu

**Maria-Florina Balcan**
Carnegie Mellon University
ninamf@cs.cmu.edu

**Pradeep Ravikumar**
Carnegie Mellon University
pkr@cs.cmu.edu

## Abstract

As larger deep learning models are hard to interpret, there has been a recent focus on generating explanations of these black-box models. In contrast, we may have apriori explanations of how models should behave. In this paper, we formalize this notion as learning from *explanation constraints* and provide a learning theoretic framework to analyze how such explanations can improve the learning of our models. One may naturally ask, "When would these explanations be helpful?" Our first key contribution addresses this question via a class of models that satisfies these explanation constraints in expectation over new data. We provide a characterization of the benefits of these models (in terms of the reduction of their Rademacher complexities) for a canonical class of explanations given by gradient information in the settings of both linear models and two layer neural networks. In addition, we provide an algorithmic solution for our framework, via a variational approximation that achieves better performance and satisfies these constraints more frequently, when compared to simpler augmented Lagrangian methods to incorporate these explanations. We demonstrate the benefits of our approach over a large array of synthetic and real-world experiments.

## 1 Introduction

There has been a considerable recent focus on generating explanations of complex black-box models so that humans may better understand their decisions. These can take the form of feature importance [31, 35], counterfactuals [31, 35], influential training samples [18, 43], etc. But what if humans were able to provide explanations for how these models should behave? We are interested in the question of how to learn models given such apriori explanations. A recent line of work incorporates explanations as a regularizer, penalizing models that do not exhibit apriori given explanations [33, 32, 15, 36]. For example, Rieger et al. [32] penalize the feature importance of spurious patches on a skin-cancer classification task. These methods lead to models that inherently satisfy "desirable" properties and, thus, are more trustworthy. In addition, some of these empirical results suggest that constraining models via explanations also leads to higher accuracy and robustness to changing test environments. However, there is no theoretical analysis to explain this phenomenon.

We note that such explanations can arise from domain experts and domain knowledge, but also other large "teacher" models that might have been developed for related tasks. An attractive facet of the latter is that we can automatically generate model-based explanations given unlabeled data points. For instance, we can use segmentation models to select the background pixels of images solely on

---

[*]Equal contribution

37th Conference on Neural Information Processing Systems (NeurIPS 2023).

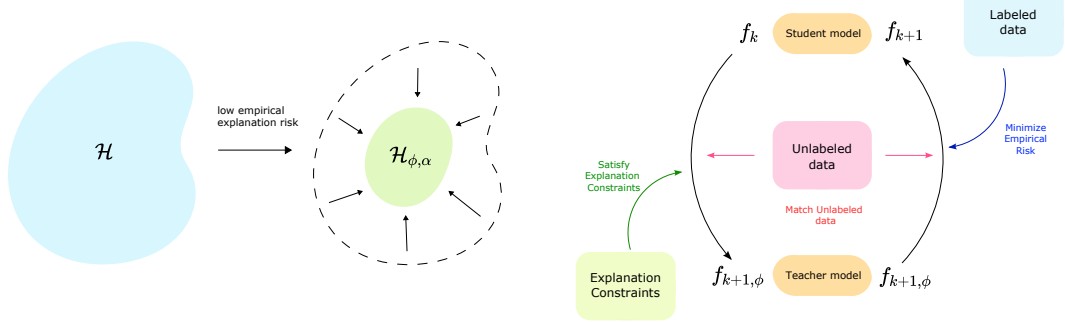

Figure 1: A restricted hypothesis class $\mathcal{H}_{\phi,\alpha}$ (left). Our algorithmic solution to solve a proposed variational objective in Section 5 (right).

unlabeled data, which we can use in our model training. We thus view incorporating explanation constraints from such teacher models as a form of knowledge distillation into our student models [13].

In this paper, we provide an analytical framework for learning from explanations to reason when and how explanations can improve the model performance. We first provide a mathematical framework for model constraints given explanations. Casting explanations as functionals $g$ that take in a model $h$ and input $x$ (as is standard in explainable AI), we can represent domain knowledge of how models should behave as constraints on the values of such explanations. We can leverage these to then solve a constrained ERM problem where we additionally constrain the model to satisfy these explanation constraints. Since the explanations and constraints are provided on randomly sampled inputs, these constraints are random. Nevertheless, via standard statistical learning theoretic arguments [38], any model that satisfies the set of explanation constraints on the finite sample can be shown to satisfy the constraints in expectation up to some slack with high probability. In our work, we term a model that satisfies explanations constraints in expectation, an *CE model* (see Definition 1). Then, we can capture the benefit of learning with explanation constraints by analyzing the generalization capabilities of this class of CE models (Theorem 3.2). This analysis builds off of a learning theoretic framework for semi-supervised learning of Balcan and Blum [1, 2]. We remark that if the explanation constraints are arbitrary, it is not possible to reason if a model satisfies the constraints in expectation based on a finite sample. We provide a detailed discussion on when this argument is possible in Appendix B,D. In addition, we note that our work also has a connection with classical approaches in stochastic programming [16, 4] and is worth investigating this relationship further.

Another key contribution of our work is concretely analyzing this framework for a canonical class of explanation constraints given by gradient information for linear models (Theorem 4.1) and two layer neural networks (Theorem 4.2). We focus on gradient constraints as we can represent many different notions of explanations, such as feature importance and ignoring spurious features. These corollaries clearly illustrate that restricting the hypothesis class via explanation constraints can lead to fewer required labeled data. Our results also provide a quantitative measure of the benefits of the explanation constraints in terms of the number of labeled data. We also discuss when learning these explanation constraints makes sense or is possible (i.e., with a finite generalization bound). We note that our framework allows for the explanations to be noisy, and not fully satisfied by even the Bayes optimal classifier. Why then would incorporating explanation constraints help? As our analysis shows, this is by reducing the estimation error (variance) by constraining the hypothesis class, at the expense of approximation error (bias). We defer the question of how to explicitly denoise noisy explanations to future work.

Now that we have provided a learning theoretic framework for these explanation constraints, we next consider the algorithmic question: how do we solve for these explanation-constrained models? In general, these constraints are not necessarily well-behaved and are difficult to optimize. One can use augmented Lagrangian approaches [33, 7], or simply regularized versions of our constrained problems [32] (which however do not in general solve the constrained problems for non-convex parameterizations but is more computationally tractable). We draw from seminal work in posterior regularization [9], which has also been studied in the capacity of model distillation [14], to provide a variational objective. Our objective is composed of two terms; supervised empirical risk and the discrepancy between the current model and the class of CE models. The optimal solution of our objective is also the optimal solution of the constrained problem which is consistent with our theoretical analysis. Our objective naturally incorporates unlabeled data and provides a simple way to

control the trade-off between explanation constraints and the supervised loss (Section 5). We propose a tractable algorithm that iteratively trains a model on the supervised data, and then approximately projects this learnt model onto the class of CE models. Finally, we provide an extensive array of experiments that capture the benefits of learning from explanation constraints. These experiments also demonstrate that the variational approach improves over simpler augmented Lagrangian approaches and can lead to models that indeed satisfy explanations more frequently.

## 2   Related Work

**Explainable AI.** Recent advances in deep learning have led to models that achieve high performance but which are also highly complex [20, 11]. Understanding these complex models is crucial for safe and reliable deployments of these systems in the real-world. One approach to improve our understanding of a model is through explanations. This can take many forms such as feature importance [31, 35, 23, 37], high level concepts [17, 44], counterfactual examples [39, 12, 25], robustness of gradients [41], or influential training samples [18, 43].

In contrast to generating post-hoc explanations of a given model, we aim to learn models given apriori explanations. There has been some recent work along such lines. Koh et al. [19], Zarlenga et al. [45] incorporates explanations within the model architecture by requiring a conceptual bottleneck layer. Ross et al. [33], Rieger et al. [32], Ismail et al. [15], Stacey et al. [36] use explanations to modify the learning procedure for any class of models: they incorporate explanations as a regularizer, penalizing models that do not exhibit apriori given explanations; Ross et al. [33] penalize input gradients, while Rieger et al. [32] penalize a Contextual Decomposition score [26]. Some of these suggest that constraining models via explanations leads to higher accuracies and more robustness to spurious correlation, but do not provide analytical guarantees. On the theoretical front, Li et al. [22] show that models that are easier to explain locally also generalize well. However, Bilodeau et al. [3] show that common feature attribution methods without additional assumptions on the learning algorithm or data distribution do no better than random guessing at inferring counterfactual model behavior.

**Learning Theory.** Our contribution is to provide an analytical framework for learning from explanations that quantify the benefits of explanation constraints. Our analysis is closely related to the framework of learning with side information. Balcan and Blum [2] shows how unlabeled data can help in semi-supervised learning through a notion of compatibility between the data and the target model. This work studies classical notions of side information (e.g., margin, smoothness, and co-training). Subsequent papers have adapted this learning theoretic framework to study the benefits of representation learning [10] and transformation invariance [34]. On the contrary, our paper focuses on the more recent notion of explanations. Rather than focus on the benefits of unlabeled data, we characterize the quality of different explanations. We highlight that constraints here are stochastic, as they depend on data points which differs from deterministic constraints that have been considered in existing literature, such as constraints on the norm of weights (i.e., L2 regularization).

**Self-Training.** Our work can also be connected to the self-training literature [5, 42, 40, 8], where we could view our variational objective as comprising a regularized (potentially simpler) teacher model that encodes these explanation constraints into a student model. Our variational objective (where we use simpler teacher models) is also related to distillation, which has also been studied in terms of gradients [6].

## 3   Learning from Explanation Constraints

Let $\mathcal{X}$ be the instance space and $\mathcal{Y}$ be the label space. We focus on binary classification where $\mathcal{Y} = \{-1, 1\}$, but which can be naturally generalized. Let $\mathcal{D}$ be the joint data distribution over $(\mathcal{X}, \mathcal{Y})$ and $\mathcal{D}_{\mathcal{X}}$ the marginal distribution over $\mathcal{X}$. For any classifier $h : \mathcal{X} \to \mathcal{Y}$, we are interested in its classification error $\mathrm{err}(h) := \mathrm{Pr}_{(x,y) \sim D}(h(x) \neq y)$, though one could also use other losses to define classification error. Our goal is to learn a classifier with small error from a family of functions $\mathcal{H}$. In this work, we use the words model and classifier interchangeably. Now, we formalize local explanations as functionals that take in a model and a test input, and output a vector:

**Definition 1** (Explanations). *Given an instance space $\mathcal{X}$, model hypothesis class $\mathcal{H}$, and an explanation functional $g : \mathcal{H} \times \mathcal{X} \to \mathbb{R}^r$, we say $g(h, x)$ is an explanation of $h$ on point $x$ induced by $g$.*

For simplicity, we consider the setting when $g$ takes a single data point and model as input, but this can be naturally extended to multiple data points and models. We can combine these explanations with prior knowledge on how explanations should look like at sample points in term of constraints.

**Definition 2** (Explanation Constraint Set). *For any instance space $\mathcal{X}$, hypothesis class $\mathcal{H}$, an explanation functional $g : \mathcal{H} \times \mathcal{X} \to \mathbb{R}^r$, and a family of constraint sets $\{C(x) \subseteq \mathbb{R}^r \mid x \in \mathcal{X}\}$, we say that $h \in \mathcal{H}$ satisfies the explanation constraints with respect to $C$ iff:*

$$g(h, x) \in C(x), \ \forall x \in \mathcal{X}.$$

In our definition, $C(x)$ represents values that we believe our explanations should take at a point $x$. For example, "an input gradient of a feature 1 must be larger than feature 2" can be represented by $g(h, x) = \nabla_x h(x)$ and $C(x) = \{(x_1, \ldots, x_d) \in \mathbb{R}^d \mid x_1 > x_2\}$. In practice, human annotators will be able to provide the constraint set $C(x')$ for a random sample $k$ data points $S_E = \{x'_1, \ldots, x'_k\}$ drawn i.i.d. from $\mathcal{D}_{\mathcal{X}}$. We then say that any $h \in \mathcal{H}$ $S_E$-satisfies the explanation constraints with respect to $C$ iff $g(h, x) \in C(x), \ \forall x \in S_E$. We note that the constraints depends on random samples $x'_i$ and therefore *are random*. To tackle this challenge, we can draw from the standard learning theoretic arguments to reason about probably approximately satisfying the constraints in expectation. Before doing so, we first consider the notion of explanation surrogate losses, which will allow us to generalize the setup above to a form that is amenable to practical estimators.

**Definition 3.** *(Explanation surrogate loss) An explanation surrogate loss $\phi : \mathcal{H} \times \mathcal{X} \to \mathbb{R}$ quantifies how well a model $h$ satisfies the explanation constraint $g(h, x) \in C(x)$. For any $h \in \mathcal{H}, x \in \mathcal{X}$:*

1. *$\phi(h, x) \geq 0$.*

2. *If $g(h, x) \in C(x)$ then $\phi(h, x) = 0$.*

For example, we could define $\phi(h, x) = 1\{g(h, x) \in C(x)\}$. Given such a surrogate loss, we can substitute the explanation constraint that $g(h, x) \in C(x)$ with the surrogate $\phi(h, x) \leq 0$. We now have the machinery to formalize how to reason about the random explanation constraints given a random set of inputs. First, denote the expected explanation loss as $\phi(h, \mathcal{D}) := \mathbb{E}_{x \sim \mathcal{D}}[\phi(h, x)]$. We are interested in models that satisfy the explanation constraints up to some slack $\tau$ (i.e. approximately) in expectation. We define a learnability condition of this explanation surrogate loss as EPAC (Explanation Probably Approximately Correct) learnability.

**Definition 4** (EPAC learnability). *For any $\delta \in (0, 1), \tau > 0$, the sample complexity of $(\delta, \tau)$ - EPAC learning of $\mathcal{H}$ with respect to a surrogate loss $\phi$, denoted $m(\tau, \delta; \mathcal{H}, \phi)$ is defined as the smallest $m \in \mathbb{N}$ for which there exists a learning rule $\mathcal{A}$ such that every data distribution $\mathcal{D}_{\mathcal{X}}$ over $\mathcal{X}$, with probability at least $1 - \delta$ over $S \sim \mathcal{D}^m$,*

$$\phi(\mathcal{A}(S), \mathcal{D}) \leq \inf_{h \in \mathcal{H}} \phi(h, \mathcal{D}) + \tau.$$

*If no such $m$ exists, define $m(\tau, \delta; \mathcal{H}, \phi) = \infty$. We say that $\mathcal{H}$ is EPAC learnable in the agnostic setting with respect to a surrogate loss $\phi$ if $\forall \delta \in (0, 1), \tau > 0, m(\tau, \delta; \mathcal{H}, \phi)$ is finite.*

*Furthermore, for a constant $\tau$, we denote any model $h \in \mathcal{H}$ with $\tau$-Approximately Correct Explanation where $\phi(h, \mathcal{D}) \leq \tau$, with a $\tau$ - CE models. We define the class of $\tau$ - CE models as*

$$\mathcal{H}_{\phi, \mathcal{D}, \tau} = \{h \in \mathcal{H} \ : \ \phi(h, \mathcal{D}) \leq \tau\}. \tag{1}$$

We simply use $\mathcal{H}_{\phi, \tau}$ to denote this class of CE models. From natural statistical learning theoretic arguments, a model that satisfies the random constraints in $S_E$ might also be a CE model.

**Proposition 3.1.** *Suppose a model $h$ $S_E$-satisfies the explanation constraints then*

$$\phi(h, \mathcal{D}_{\mathcal{X}}) \leq 2R_k(\mathcal{G}) + \sqrt{\frac{\ln(4/\delta)}{2k}},$$

*with probability at least $1 - \delta$, when $k = |S_E|$ and $\mathcal{G} = \{\phi(h, \cdot) \mid h \in \mathcal{H}\}$.*

We use $R_k(\cdot)$ to denote Rademacher complexity; please see Appendix A where we review this and related concepts. Note that even when $h$ satisfies the constraints exactly on $S$, we can only guarantee a bound on the expected surrogate loss $\phi(h, \mathcal{D}_{\mathcal{X}})$. We can achieve a bound similar to that in Proposition 3.1 via a single and simpler constraint on the empirical expectation $\phi(h, S_E) = $

$\frac{1}{|S_E|} \sum_{x \in S_E} \phi(h, x)$. We can then extend the above proposition to show that if $\phi(h, S_E) \leq \tau$, then $\phi(h, \mathcal{D}_{\mathcal{X}}) \leq \tau + 2R_k(\mathcal{G}) + \sqrt{\frac{\ln(4/\delta)}{2k}}$, with probability at least $1 - \delta$. Another advantage of such a constraint is that the explanation constraints could be noisy, or it may be difficult to satisfy them exactly, so $\tau$ also serves as a slack. The class $\mathcal{G}$ contains all surrogate losses of any $h \in \mathcal{H}$. Depending on the explanation constraints, $\mathcal{G}$ can be extremely large. We remark that the surrogate loss $\phi$ allows us to reason about satisfying an explanation constraint on a new data point and in expectation. However, for many constraints, $\phi$ does not have a closed-form or is unknown on an unseen data point. The question of which types of explanation constraints are generalizable may be of independent interest, and we further discuss this in Appendix B and provide further examples of learnable constraints in Appendix D.

**EPAC-ERM Objective.** Let us next discuss *combining* the two sources of information: the explanation constraints that we set up in the previous section, together with the usual set of labeled training samples $S = \{(x_1, y_1), \ldots, (x_n, y_n)\}$ drawn i.i.d. from $\mathcal{D}$ that informs the empirical risk. Combining these, we get what we call EPAC-ERM objective:

$$\min_{h \in \mathcal{H}} \frac{1}{n} \sum_{i=1}^{n} \ell(h, x_i, y_i) \ \text{ s.t. } \ \frac{1}{k} \sum_{i=1}^{k} \phi(h, x_i') \leq \tau. \tag{2}$$

We provide a learnability condition for a model that achieve both low average error and surrogate loss in Appendix F.

## 3.1 Generalization Bound

We assume that we are in a doubly agnostic setting. Firstly, we are agnostic in the usual sense that there need be no classifier in the hypothesis class $\mathcal{H}$ that perfectly labels $(x, y)$; instead, we hope to achieve the best error rate in the hypothesis class, $h^* = \arg\min_{h \in \mathcal{H}} \text{err}_{\mathcal{D}}(h)$. Secondly, we are also agnostic with respect to the explanations, so that the optimal classifier $h^*$ may not satisfy the explanation constraints exactly, so that it incurs nonzero surrogate explanation loss $\phi(h^*, D) > 0$.

**Theorem 3.2** (Generalization Bound for Agnostic Setting). *Consider a hypothesis class $\mathcal{H}$, distribution $\mathcal{D}$, and explanation loss $\phi$. Let $S = \{(x_1, y_1), \ldots, (x_n, y_n)\}$ be drawn i.i.d. from $\mathcal{D}$ and $S_E = \{x_1', \ldots, x_k'\}$ drawn i.i.d. from $\mathcal{D}_{\mathcal{X}}$. With probability at least $1 - \delta$, for $h \in \mathcal{H}$ that minimizes empirical risk $\text{err}_S(h)$ and has $\phi(h, S_E) \leq \tau$, we have*

$$\text{err}_D(h) \leq \text{err}_D(h^*_{\tau - \varepsilon_k}) + 2R_n(\mathcal{H}_{\phi, \tau + \varepsilon_k}) + 2\sqrt{\frac{\ln(4/\delta)}{2n}},$$

$$\varepsilon_k = 2R_k(\mathcal{G}) + \sqrt{\frac{\ln(4/\delta)}{2k}},$$

*when $\mathcal{G} = \{\phi(h, x) \mid h \in \mathcal{H}, x \in \mathcal{X}\}$ and $h^*_{\tau} = \arg\min_{h \in \mathcal{H}_{\phi, \tau}} \text{err}_{\mathcal{D}}(h)$.*

*Proof.* The proof largely follows the arguments in Balcan and Blum [2], but we use Rademacher complexity-based deviation bounds instead of VC-entropy. We defer the full proof to Appendix E. □

Our bound suggests that these constraints help with our learning by shrinking the hypothesis class $\mathcal{H}$ to $\mathcal{H}_{\phi, \tau + \varepsilon_k}$, reducing the required sample complexity. However, there is also a trade-off between reduction and accuracy. In our bound, we compare against the best classifier $h^*_{\tau - \varepsilon_k} \in \mathcal{H}_{\phi, \tau - \varepsilon_k}$ instead of $h^*$. Since we may have $\phi(h^*, \mathcal{D}) > 0$, if $\tau$ is too small, we may reduce $\mathcal{H}$ to a hypothesis class that does not contain any good classifiers. Recall that the generalization bound for standard supervised learning — in the absence of explanation constraints — is given by

$$\text{err}_D(h) \leq \text{err}_D(h^*) + 2R_n(\mathcal{H}) + 2\sqrt{\frac{\ln(2/\delta)}{2n}}.$$

We can see the difference between this upper bound and the upper bound in Theorem 3.2 here as a possible notion of the goodness of an explanation constraint. We further discuss this in Appendix C.

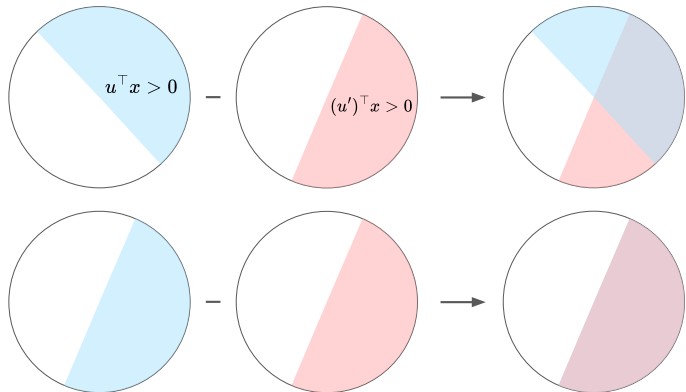

Figure 2: Visualization of the piecewise constant function of $\nabla_x h(x) - \nabla_x h'(x)$ when $h$ is a two layer NNs with 1 node. Background colors represent regions with non-zero value.

## 4 Gradient Explanations for Particular Hypothesis Classes

In this section, we further quantify the usefulness of explanation constraints on different concrete examples and characterize the Rademacher complexity of the restricted hypothesis classes. In particular, we consider an explanation constraint of a constraint on the input gradient. For example, we may want our model's gradient to be close to that of some $h' \in \mathcal{H}$. This translates to $g(h, x) = \nabla_x h(x)$ and $C(x) = \{x \in \mathbb{R}^d \mid \|x - \nabla_x h'(x)\| \leq \tau\}$ for some $\tau > 0$.

### 4.1 Gradient Explanations for Linear Models

We now consider the case of a uniform distribution on a sphere, and we use the symmetry of this distribution to derive an upper bound on the Rademacher complexity (full proof to Appendix H).

**Theorem 4.1** (Rademacher complexity of linear models with a gradient constraint, uniform distribution on a sphere). *Let $\mathcal{D}_\mathcal{X}$ be a uniform distribution on a unit sphere in $\mathbb{R}^d$, let $\mathcal{H} = \{h : x \mapsto \langle w_h, x \rangle \mid w_h \in \mathbb{R}^d, \|w_h\|_2 \leq B\}$ be a class of linear models with weights bounded by a constant $B$. Let $\phi(h, x) = \theta(w_h, w_{h'})$ be a surrogate loss where $\theta(u, v)$ is an angle between $u, v$. We have*

$$R_n(\mathcal{H}_{\phi, \tau}) \leq \frac{B}{\sqrt{n}} \left( \sin(\tau) \cdot p + \frac{1-p}{2} \right),$$

*where $p = \mathrm{erf}\left( \frac{\sqrt{d} \sin(\tau)}{\sqrt{2}} \right)$ and $\mathrm{erf}(x) = \frac{2}{\sqrt{\pi}} \int_0^x e^{-t^2} dt$ is the standard error function.*

The standard upper bound on the Rademacher complexity of linear models is $\frac{B}{\sqrt{n}}$. Our bound has a nice interpretation; we shrink our bound by a factor of $(\frac{1-p}{2} + \sin(\tau)p)$. We remark that $d$ increases, we observe that $p \to 1$, so the term $\sin(\tau)p$ dominates this factor. As a consequence, we get that our bound is now scaled by $\sin(\tau) \approx \tau$ and the the Rademacher complexity scales down by a factor of $\tau$. This implies that given $n$ labeled data, to achieve a fast rate $\mathcal{O}(\frac{1}{n})$, we need $\tau$ to be as good as $O(\frac{1}{\sqrt{n}})$.

### 4.2 Gradient Explanations for Two Layer Neural Networks

**Theorem 4.2** (Rademacher complexity of two layer neural networks ($m$ hidden nodes) with a gradient constraint). *Let $\mathcal{X}$ be an instance space and $\mathcal{D}_\mathcal{X}$ be a distribution over $\mathcal{X}$ with a large enough support. Let $\mathcal{H} = \{h : x \mapsto \sum_{j=1}^m w_j \sigma(u_j^\top x) \mid w_j \in \mathbb{R}, u_j \in \mathbb{R}^d, \sum_{j=1}^m |w_j| \leq B, \|u_j\|_2 = 1\}$ be a class of two layer neural networks with a ReLU activation function and bounded weight. Assume that there exists some constant $C > 0$ such that $\mathbb{E}_{x \sim \mathcal{D}_\mathcal{X}}[\|x\|_2^2] \leq C^2$. Consider an explanation loss given by*

$$\phi(h, x) = \|\nabla_x h(x) - \nabla_x h'(x)\|_2 + \infty \cdot 1\{\|\nabla_x h(x) - \nabla_x h'(x)\| > \tau\}$$

*for some $\tau > 0$. Then, we have that $R_n(\mathcal{H}_{\phi, \tau}) \leq \frac{3\tau m C}{\sqrt{n}}$.*

*Proof.* (Sketch) The key ingredient is to identify the impact of the gradient constraint and the form of class $\mathcal{H}_{\phi, \tau}$. We provide an idea when we have $m = 1$ node. We write $h(x) = w\sigma(u^\top x)$ and $h'(x) = w'\sigma(u'^\top x)$. Note that $\nabla_x h(x) - \nabla_x h'(x) = wu 1\{u^\top x > 0\} - w'u' 1\{(u')^\top x > 0\}$ is

a piecewise constant function (Figure 2). Assume that the probability mass of each region is non-negative, our gradient constraint implies that the norm of each region cannot be larger than $\tau$.

1. If $u, u'$ have different directions, we have 4 regions in $\nabla_x h(x) - \nabla_x h'(x)$ and can conclude that $|w| < \tau, |w'| < \tau$.

2. If $u = u'$ have the same direction, we only have 2 regions in $\nabla_x h(x) - \nabla_x h'(x)$ and can conclude that $\|wu - w'u'\| = |w - w'| < \tau$.

The gradient constraint enforces a model to have the same node boundary $(u = u')$ with a small weight difference $|w - w'| < \tau$ or that node would have a small weight $|w| < \tau$. This finding allows us to determine the restricted class $\mathcal{H}_{\phi,\tau}$, and we can use this to bound the Rademacher complexity accordingly. For full details, see Appendix I. $\qquad\square$

We compare this with the standard Rademacher complexity of a two layer neural network [24],

$$R_n(\mathcal{H}) \leq \frac{2BC}{\sqrt{n}}.$$

We can do better than this standard bound if $\tau < \frac{2B}{3m}$. One interpretation for this is that we have a budget at most $\tau$ to change the weight of each node and for total $m$ nodes, we can change the weight by at most $\tau m$. We compare this to $B$ which is an upper bound on the total weight $\sum_{j=1}^{m} |w_j| \leq B$. Therefore, we can do better than a standard bound when we can change the weight by at most two thirds of the average weight $\frac{2B}{3m}$ for each node. We would like to point out that our bound does not depend on the distribution $\mathcal{D}$ because we choose a specific explanation loss that guarantees that the gradient constraint holds almost everywhere. Extending to a weaker loss such as $\phi(h, x) = \|\nabla_x h(x) - \nabla_x h'(x)\|$ is a future research direction. In contrast, our result for linear models uses a weaker explanation loss and depends on $\mathcal{D}$ (Theorem H.1). We also assume that there exists $x$ with a positive probability density at any partition created by $\nabla_x h(x)$. This is not a strong assumption, and it holds for any distribution where the support is the $\mathbb{R}^d$, e.g., Gaussian distributions.

## 5 Algorithms for Learning from Explanation Constraints

Although we have analyzed learning with explanation constraints, algorithms to solve this constrained optimization problem are non-trivial. In this setting, we assume that we have access to $n$ labeled data $\{(x_i, y_i)\}_{i=1}^{n}$, $m$ unlabeled data $\{x_{n+i}\}_{i=1}^{m}$, and $k$ data with explanations $\{(x_{n+m+i}, \phi(\cdot, x_{n+m+i}))\}_{i=1}^{k}$. We argue that in many cases, $n$ labeled data are the most expensive to annotate. The $k$ data points with explanations also have non-trivial cost; they require an expert to provide the annotated explanation or provide a surrogate loss $\phi$. If the surrogate loss is specified then we can evaluate it on any unlabeled data, otherwise these data points with explanations could be expensive. On the other hand, the $m$ data points can cheaply be obtained as they are completely unlabeled. We now consider existing approaches to incorporate this explanation information.

**EPAC-ERM:** Recall our EPAC-ERM objective from (2):

$$\min_{h \in \mathcal{H}} \frac{1}{n} \sum_{i=1}^{n} 1\{h(x_i) \neq y_i\} \text{ s.t. } \frac{1}{k} \sum_{j=n+m+1}^{n+m+k} \phi(h, x_j) \leq \tau$$

for some constant $\tau$. This constraint in general requires more complex optimization techniques (e.g., running multiple iterations and comparing values of $\tau$) to solve algorithmically. We could also consider the case where $\tau = 0$, which would entail the hypotheses satisfy the explanation constraints exactly, which however is in general too strong a constraint with noisy explanations.

**Augmented Lagrangian objectives:**

$$\min_{h \in \mathcal{H}} \frac{1}{n} \sum_{i=1}^{n} 1[h(x_i) \neq y_i] + \frac{\lambda}{k} \sum_{j=n+m+1}^{n+m+k} \phi(h, x_j)$$

As is done in prior work [32], we can consider an augmented Lagrangian objective. A crucial caveat with this approach is that the explanation surrogate loss is in general a much more complicated

functional of the hypothesis than the empirical risk. For instance, it might involve the gradient of the hypothesis when we use gradient-based explanations. Computing the gradients of such a surrogate loss can be more expensive compared to the gradients of the empirical risk. For instance, in our experiments, computing the gradients of the surrogate loss that involves input gradients is 2.5 times slower than that of the empirical risk. With the above objective, however, we need to compute the same number of gradients of both the explanation surrogate loss and the empirical risk. These computational difficulties have arguably made incorporing explanation constraints not as popular as they could be.

## 5.1 Variational Method

To alleviate these aforementioned computational difficulties, we propose a *new* variational objective

$$\min_{h \in \mathcal{H}} (1 - \lambda) \mathop{\mathbb{E}}_{(x,y) \sim \mathcal{D}} [\ell(h(x), y)] + \lambda \inf_{f \in \mathcal{H}_{\phi,\tau}} \mathop{\mathbb{E}}_{x \sim \mathcal{D}_{\mathcal{X}}} [\ell(h(x), f(x))],$$

where $\ell$ is some loss function and $t \geq 0$ is some threshold. The first term is the standard expected risk of $h$ while the second term can be viewed as a projection distance between $h$ and $\tau$-CE models. It can be seen that the optimal solution of **EPAC-ERM** would also be an optimal solution of our proposed variational objective. The advantage of this formulation however is that it decouples the standard expected risk component from the surrogate risk component. This allows us to solve this objective with the following iterative technique, drawing inspiration from prior work in posterior regularization [9, 14]. More specifically, let $h_t$ be the learned model at time $t$ and at each timestep $t$,

1. We project $h_t$ to the class of $\tau$-CE models.

$$f_{t+1,\phi} = \operatorname*{argmin}_{h \in \mathcal{H}} \frac{1}{m} \sum_{i=n+1}^{n+m} \ell(h(x_i), h_t(x_i)) \quad + \lambda \max \left( 0, \frac{1}{k} \sum_{i=n+m+1}^{n+m+k} \phi(h, x_i) - \tau \right).$$

   The first term is the difference between $h_t$ and $f$ on unlabeled data. The second term is the surrogate loss, which we want to be smaller than $t$. $\eta$ is a regularization hyperparameter.

2. We calculate $h_{t+1}$ that minimizes the empirical risk of labeled data and matches pseudolabels from $f_{t+1,\phi}$

$$h_{t+1,\phi} = \operatorname*{argmin}_{h \in \mathcal{H}} \frac{1}{n} \sum_{i=1}^{n} \ell(h(x_i), y_i) + \frac{1}{m} \sum_{i=n+1}^{n+m} \ell(h(x_i), f_{t+1,\phi}(x_i)).$$

   Here, the discrepancy between $h$ and $f_{t+1,\phi}$ is evaluated on the unlabeled data $\{x_j\}_{j=n+1}^{n+m}$.

The advantage of this decoupling is that we could use a differing number of gradient steps and learning rates for the projection step that involves the complicated surrogate loss when compared to the empirical risk minimization step. Secondly, we can simplify the projection iterate computation by replacing $\mathcal{H}_{\phi,\tau}$ with a simpler class of teacher models $\mathcal{F}_{\phi,\tau}$ for greater efficiency. Thus, the decoupled approach to solving the EPAC-ERM objective is in general more computationally convenient.

We initialize this procedure with some model $h_0$. We remark that could see this as a constraint regularized self-training where $h_t$ is a student model and $f_t$ is a teacher model. At each timestep, we project a student model to the closest teacher model that satisfies the constraint. The next student model then learns from both labeled data and pseudo labels from the teacher model. In the standard self-training, we do not have any constraint and we have $f_t = h_t$.

## 6 Experiments

We provide both synthetic and real-world experiments to support our theoretical results and clearly illustrate interesting tradeoffs of incorporating explanations. In our experiments, we compare our method against 3 baselines: (1) a standard supervised learning approach, (2) a simple Lagrangian-regularized method (that directly penalizes the surrogate loss $\phi$), and (3) self-training, which propagates the predictions of (1) and matches them on unlabeled data. We remark that (2) captures the essence of the method in Ross et al. [33], except there is no $\ell_2$ regularization term.

Our experiments demonstrate that the proposed variational approach is preferable to simple Lagrangian methods and other supervised methods in many cases. In particular, the variational approach leads to a higher accuracy under limited labeled data settings. In addition, our method leads

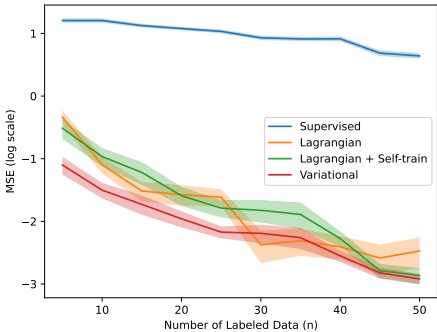

Figure 3: Comparison of MSE on regressing a linear model. Results are averaged over 5 seeds. $m = 1000, k = 20$.

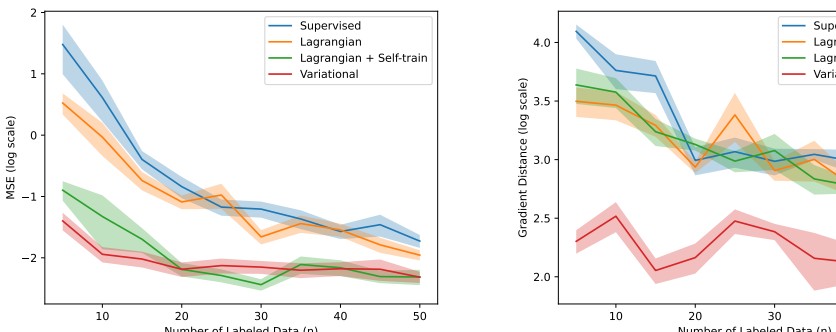

Figure 4: Comparison of MSE on regressing a two layer neural network (left) and $\ell_2$ distance over input gradients as we vary the amount of labeled data $n$ (right). Left is task performance and right is explanation constraint satisfcation. Results are averaged over 5 seeds. $m = 1000, k = 20$.

to models that satisfy the explanation constraints much more frequently than other baselines. We also compare to a Lagrangian-regularized + self-training baseline (first, we use the model (2) to generate pseudolabels for unlabeled data and then train a new model on both labeled and unlabeled data) in Appendix L. We remark that this baseline isn't a standard method in practice and does not fit nicely into a theoretical framework, although it seems to be the most natural approach to using unlabeled data in this procedure. More extensive ablations are deferred to Appendix N, and code to replicate our experiments will be released with the full paper.

## 6.1 Regression Task with Exact Gradient Information

In our synthetic experiments, we focus on a regression task where we try to learn some model contained in our hypothesis class. Our data is given by $\mathcal{X} = \mathbb{R}^d$, and we try to learn a target function $h^* : \mathcal{X} \to \mathbb{R}$. Our data distribution is given by $X \sim \mathcal{N}(0, \sigma^2 I)$, where $I$ is a $d \times d$ identity matrix. We generate $h^*$ by randomly initializing a model in the specific hypothesis class $\mathcal{H}$. We assume that we have $n$ labeled data, $m$ unlabeled data, and $k$ data with explanations.

We first present a synthetic experiment for learning with a perfect explanation, meaning that $\phi(h^*, S) = 0$. We consider the case where we have the *exact* gradient of $h^*$. Here, let $\mathcal{H}$ be a linear classifier and note that the exact gradient gives us the slope of the linear model, and we only need to learn the bias term. Incorporating these explanation indeed helps as both methods that include explanation constraints (Lagrangian and ours) perform much better (Figure 3).

We also demonstrate incorporating this information for two layer neural networks. We observe a clear difference between the simpler Lagrangian approach and our variational objective (Figure 4 - left). Our method is clearly the best in the setting with limited labeled data and matches the performance of the strong self-training baseline with sufficient labeled data. We note that this is somewhat expected, as these constraints primarily help in the setting with limited labeled data; with enough labeled data, standard PAC bounds suffice for strong performance.

We also analyze how strongly the approaches enforce these explanation constraints on new data points that are seen at test time (Figure 4 - right) for two layer NNs. We observe that our variational objective approaches have input gradients that more closely match the ground-truth target network's

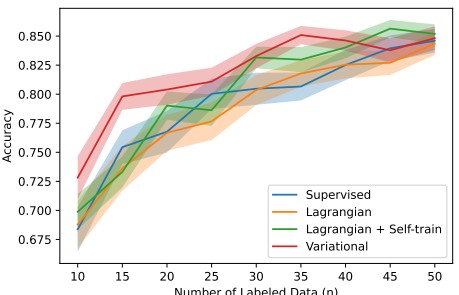 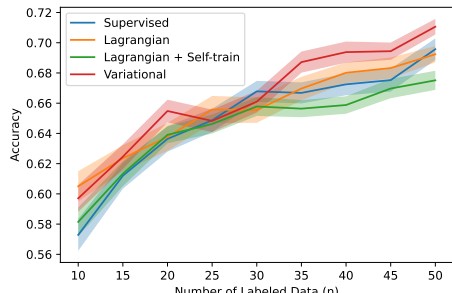

Figure 5: Comparison of accuracy on the YouTube (left) and the Yelp (right) datasets. Here, we let $m = 500, k = 150, T = 2, \tau = 0.0$. Results are averaged over 40 seeds.

input gradients. This demonstrates that, in the case of two layer NNs with gradient explanations, our approach best achieves both good performance and satisfying the constraints. Standard self-training achieves similar performance in terms of MSE but has no notion of satisfying the explanation constraints. The Lagrangian method does not achieve the same level of satisfying these explanations as it is unable to generalize and satisfy these constraints on new data.

## 6.2 Tasks with Imperfect Explanations

Assuming access to perfect explanations may be unrealistic in practice, so we present experiments when our explanations are imperfect. We present classification tasks (Figure 5) from a weak supervision benchmark [46]. In this setting, we obtain explanations through the approximate gradients of a single weak labeler, as is done in [? ]. More explicitly, weak labelers are heuristics designed by domain experts; one example is functions that check for the presence of particular words in a sentence (e.g., checking for the word "delicious" in a Yelp comment, which would indicate positive sentiment). We can then access gradient information from such weak labelers, which gives us a notion of feature importance about particular features in our data. We note that these examples of gradient information are rather *easy* to obtain, as we only need domain experts to specify simple heuristic functions for a particular task. Once given these functions, we can apply them easily over unlabeled data without requiring any example-level annotations.

We observe that our variational objective achieves better performance than all other baseline approaches on the majority of settings defined by the number of labeled data. We remark that the explanation in this dataset is a noisy gradient explanation along two feature dimensions, yet this still improves upon methods that do not incorporate this explanation constraint. Indeed, our method outperforms the Lagrangian approach, showing the benefits of iterative rounds of self-training over the unlabeled data. In addition to our real-world experiments, we present synthetic experiments with noisy gradients in Appendix K.1.

## 7 Discussion

Our work proposes a new learning theoretic framework that provides insight into how apriori explanations of desired model behavior can benefit the standard machine learning pipeline. The statistical benefits of explanations arise from constraining the hypothesis class: explanation samples serve to better estimate the population explanation constraint, which constrains the hypothesis class. This is to be contrasted with the statistical benefit of labeled samples, which serve to get a better estimate of the population risk. We provide instantiations of our analysis for the canonical class of gradient explanations, which captures many explanations in terms of feature importance. It would be of interest to provide corollaries for other types of explanations in future work. As mentioned before, the generality of our framework has larger implications towards incorporating constraints that are not considered as "standard" explanations. For example, this work can be leveraged to incorporate more general notions of side information and inductive biases. We also discuss the societal impacts of our approach in Appendix O. As a whole, our paper supports using further information (e.g., explanation constraints) in the standard learning setting.

# 8 Acknowledgements

This work was supported in part by DARPA under cooperative agreement HR00112020003, FA8750-23-2-1015, ONR grant N00014-23-1-2368, NSF grant IIS-1909816, a Bloomberg Data Science PhD fellowship and funding from Bosch Center for Artificial Intelligence and the ARCS Foundation.

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

# A  Uniform Convergence via Rademacher Complexity

A standard tool for providing performance guarantees of supervised learning problems is a generalization bound via uniform convergence. We will first define the Rademacher complexity and its corresponding generalization bound.

**Definition 5.** *Let $\mathcal{F}$ be a family of functions mapping $\mathcal{X} \to \mathbb{R}$. Let $S = \{x_1, \ldots, x_m\}$ be a set of examples drawn i.i.d. from a distribution $D_{\mathcal{X}}$. Then, the empirical Rademacher complexity of $\mathcal{F}$ is defined as*

$$R_S(\mathcal{F}) = \mathop{\mathbb{E}}_{\sigma} \left[ \sup_{f \in \mathcal{F}} \left( \frac{1}{m} \sum_{i=1}^{m} \sigma_i f(x_i) \right) \right]$$

*where $\sigma_1, \ldots, \sigma_m$ are independent random variables uniformly chosen from $\{-1, 1\}$.*

**Definition 6.** *Let $\mathcal{F}$ be a family of functions mapping $\mathcal{X} \to \mathbb{R}$. Then, the Rademacher complexity of $\mathcal{F}$ is defined as*

$$R_n(\mathcal{F}) = \mathop{\mathbb{E}}_{S \sim \mathcal{D}_{\mathcal{X}}^n} [R_S(\mathcal{F})].$$

*The Rademacher complexity is the expectation of the empirical Rademacher complexity, over $n$ samples drawn i.i.d. from the distribution $D_{\mathcal{X}}$.*

**Theorem A.1** (Rademacher-based uniform convergence). *Let $D_{\mathcal{X}}$ be a distribution over $\mathcal{X}$, and $\mathcal{F}$ a family of functions mapping $\mathcal{X} \to [0, 1]$. Let $S = \{x_1, \ldots, x_n\}$ be a set of samples drawn i.i.d. from $D_{\mathcal{X}}$, then with probability at least $1 - \delta$ over our draw $S$,*

$$|\mathbb{E}_{\mathcal{D}}[f(x)] - \hat{\mathbb{E}}_S[f(x)]| \le 2R_n(\mathcal{F}) + \sqrt{\frac{\ln(2/\delta)}{2n}}.$$

*This holds for every function $f \in \mathcal{F}$, and $\hat{\mathbb{E}}_S[f(x)]$ is expectation over a uniform distribution over $S$.*

This bound on the empirical Rademacher complexity leads to the standard generalization bound for supervised learning.

**Theorem A.2.** *For a binary classification setting when $y \in \{\pm 1\}$ with a zero-one loss, for $\mathcal{H} \subset \{h : \mathcal{X} \to \{-1, 1\}\}$ be a family of binary classifiers, let $S = \{(x_1, y_1), \ldots, (x_n, y_n)\}$ is drawn i.i.d. from $D$ then with probability at least $1 - \delta$, we have*

$$|\text{err}_{\mathcal{D}}(h) - \widehat{\text{err}_S}(h)| \le R_n(\mathcal{H}) + \sqrt{\frac{\ln(2/\delta)}{2n}},$$

*for every $h \in \mathcal{H}$ when*

$$\text{err}_{\mathcal{D}}(h) = \mathop{\Pr}_{(x,y) \sim \mathcal{D}}(h(x) \neq y)$$

*and*

$$\widehat{\text{err}}_S(h) = \frac{1}{n} \sum_{i=1}^{n} 1[h(x_i) \neq y_i]$$

*is the empirical error on $S$.*

For a linear model with a bounded weights in $\ell_2$ norm, the Rademacher complexity is $\mathcal{O}(\frac{1}{\sqrt{n}})$. We refer to the proof from Ma [24] for this result.

**Theorem A.3** (Rademacher complexity of a linear model ([24])). *Let $\mathcal{X}$ be an instance space in $\mathbb{R}^d$, let $\mathcal{D}_{\mathcal{X}}$ be a distribution on $\mathcal{X}$, let $\mathcal{H} = \{h : x \to \langle w_h, x \rangle \mid w_h \in \mathbb{R}^d, ||w_h||_2 \le B\}$ be a class of linear model with weights bounded by some constant $B > 0$ in $\ell_2$ norm. Assume that there exists a constant $C > 0$ such that $\mathbb{E}_{x \sim \mathcal{D}_{\mathcal{X}}}[||x||_2^2] \le C^2$. For any $S = \{x_1, \ldots, x_n\}$ is drawn i.i.d. from $\mathcal{D}_{\mathcal{X}}$, we have*

$$R_S(\mathcal{H}) \le \frac{B}{n} \sqrt{\sum_{i=1}^{n} ||x_i||_2^2}$$

*and*

$$R_n(\mathcal{H}) \le \frac{BC}{\sqrt{n}}.$$

Many of our proofs require the usage of Talgrand's lemma, which we now present.

**Lemma A.4** (Talgrand's Lemma [21]). *Let $\phi : \mathbb{R} \to \mathbb{R}$ be a $k$-Lipschitz function. Then for a hypothesis class $\mathcal{H} = \{h : \mathbb{R}^d \to \mathbb{R}\}$, we have that*

$$R_S(\phi \circ \mathcal{H}) \leq k R_s(\mathcal{H})$$

*where $\phi \circ \mathcal{H} = \{f : z \mapsto \phi(h(z)) | h \in \mathcal{H}\}$.*

## B  Generalizable Constraints

We know that constraints $C(x)$ capture human knowledge about how explanations at a point $x$ should behave. For any constraints $C(x)$ that are known apriori for all $x \in \mathcal{X}$, we can evaluate whether a model satisfies the constraints at a point $x \in \mathcal{X}$. This motivates us to discuss the ability of models to generalize from any finite samples $S_E$ to satisfy these constraints over $\mathcal{X}$ with high probability. Having access to $C(x)$ is equivalent to knowing how models should behave over *all* possible data points in terms of explanations, which may be too strong of an assumption. Nevertheless, many forms of human knowledge can be represented by a closed-form function $C(x)$. For example,

1. An explanation has to take value in a fixed range can be represented by $C(x) = \Pi_{i=1}^{r}[a_i, b_i], \forall x \in \mathcal{X}$.

2. An explanation has to stay in a ball around $x$ can be represented by $C(x) = \{u \in \mathbb{R}^d \mid \|u - x\|_2 \leq r\}$.

3. An explanation has to stay in a rectangle around $\frac{x}{3}$ can be represented by $C(x) = \{u \in \mathbb{R}^d \mid \frac{x_i}{3} - a_i \leq u_i \leq \frac{x_i}{3} + b_i, i = 1, \ldots, d\}$.

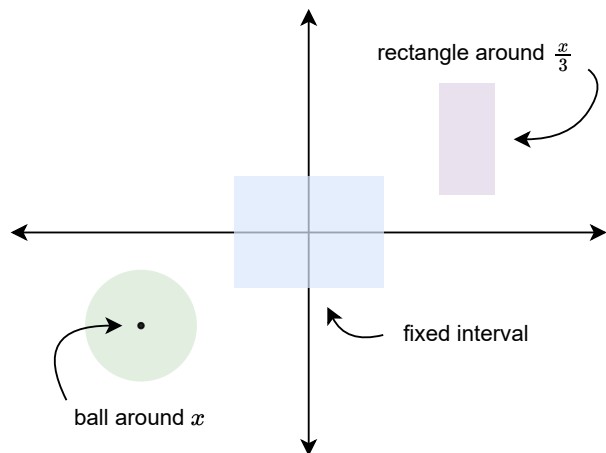

Figure 6: Illustration of examples of explanation constraints, given from some learnable class $C(x)$.

In this case, there always exists a surrogate loss that represents the explanation constraints $C(x)$; for example, we can set $\phi(h, x) = 1\{g(h, x) \in C(x)\}$. On the other hand, directly specifying explanation constraints through a surrogate loss would also imply that $C(x)$ is known apriori for all $x \in \mathcal{X}$. The task of generalization to satisfy the constraint on unseen data is well-defined in this setting. Furthermore, if a surrogate loss $\phi$ is specified, then we can evaluate $\phi(h, x)$ on any unlabeled data point without the need for human annotators which is a desirable property.

On the other hand, we usually do not have knowledge over all data points $x \in \mathcal{X}$; rather, we may only know these explanation constraints over a random sample of $k$ data points $S_E = \{x'_1, \ldots, x'_k\}$. If we do not know the constraint set $C(x)$, it is unclear what satisfying the constraint at an unseen data point $x$ means. Indeed, without additional assumptions, it may not make sense to think about generalization. For example, if there is no relationship between $C(x)$ for different values of $x$, then it is not possible to infer about $C(x)$ from $C(x'_i)$ for $i = 1, \ldots, k$. In this case, we could define

$$\phi(h, x) = 1\{g(h, x) \in C(x)\}1\{x \in S_E\},$$

where we are only interested in satisfying these explanation constraints over the finite sample $S_E$. For other data points, we have $\phi(h, x) = 0$. This guarantees that any model with low empirical explanation loss would also achieve loss expected explanation loss, although this does not have any particular implication on any notion of generalization to new constraints. Regardless, we note that our explanation constraints still reduce the size of the hypothesis class from $\mathcal{H}$ to $\mathcal{H}_{\phi,\tau}$, leading to an improvement in sample complexity.

The more interesting setting, however, is when we make an additional assumption that the true (unknown) surrogate loss $\phi$ exists and, during training, we only have access to instances of this surrogate loss evaluated on the sample $\phi(\cdot, x_i')$. We can apply a uniform convergence argument to achieve

$$\phi(h, \mathcal{D}_{\mathcal{X}}) \leq \phi(h, S_E) + 2R_k(\mathcal{G}) + \sqrt{\frac{\ln(4/\delta)}{2k}}$$

with probability at least $1 - \delta$ over $S_E$, drawn i.i.d. from $\mathcal{D}_{\mathcal{X}}$ and $\mathcal{G} = \{\phi(h, \cdot) | h \in \mathcal{H}\}$, $k = |S_E|$. Although the complexity term $R_k(\mathcal{G})$ is unknown (since $\phi$ is unknown), we can upper bound this by the complexity of a class of functions $\Phi$ (e.g., neural networks) that is large enough to well-approximate any $\phi(h, \cdot) \in \mathcal{G}$, meaning that $R_k(\mathcal{G}) \leq R_k(\Phi)$. Comparing to the former case when $C(x)$ is known for all $x \in \mathcal{X}$ apriori, the generalization bound has a term that increases from $R_k(\mathcal{G})$ to $R_k(\Phi)$, which may require more explanation-annotated data to guarantee generalization to new data points. We note that the simpler constraints lead to a simpler surrogate loss, which in turn implies a less complex upper bound $\Phi$. This means that simpler constraints are easier to learn.

Nonetheless, this is a more realistic setting when explanation constraints are hard to acquire and we do not have the constraints for all data points in $\mathcal{X}$. For example, Ross et al. [33] considers an image classification task on MNIST, and imposes an explanation constraint in terms of penalizing the input gradient of the background of images. In essence, the idea is that the background should be less important than the foreground for the classification task. In general, this constraint does not have a closed-form expression, and we do not even have access to the constraint for unseen data points. However, if we assume that a surrogate loss $\phi(h, \cdot)$ can be well-approximated by two layer neural networks, then our generalization bound allows us to reason about the ability of model to generalize and ignore background features on new data.

## C   Goodness of an explanation constraint

**Definition 7** (Goodness of an explanation constraint). *For a hypothesis class $\mathcal{H}$, a distribution $\mathcal{D}$ and an explanation loss $\phi$, the goodness of $\phi$ with respect to a threshold $\tau$ and $n$ labeled examples is:*

$$G_{n,\tau}(\phi, \mathcal{H}) = (R_n(\mathcal{H}) - R_n(\mathcal{H}_{\phi,\tau})) + (\text{err}_{\mathcal{D}}(h^*) - \text{err}_{\mathcal{D}}(h_t^*))$$

$$h^* = \arg\min_{h \in \mathcal{H}} \text{err}_{\mathcal{D}}(h), \quad h_\tau^* = \arg\min_{h \in \mathcal{H}_{\phi,\tau}} \text{err}_{\mathcal{D}}(h).$$

Here, we assume access to infinite explanation data so that $\varepsilon_k \to 0$. The goodness depends on the number of labeled examples $n$ and a threshold $t$. In our definition, a good explanation constraint leads to a reduction in the complexity of $\mathcal{H}$ while still containing a classifier with low error. This suggests that the benefits from explanation constraints exhibit diminishing returns as $n$ becomes large. In fact, as $n \to \infty$, we have $R_n(\mathcal{H}) \to 0$, $R_n(\mathcal{H}_{\phi,\tau}) \to 0$ which implies $G_n(\phi, \mathcal{H}) \to \text{err}_{\mathcal{D}}(h^*) - \text{err}_{\mathcal{D}}(h_\tau^*) \leq 0$. On the other hand, explanation constraints help when $n$ is small. For $t$ large enough, we expect $\text{err}_{\mathcal{D}}(h^*) - \text{err}_{\mathcal{D}}(h_\tau^*)$ to be small, so that our notion of goodness is dominated by the first term: $R_n(\mathcal{H}) - R_n(\mathcal{H}_{\phi,\tau})$, which has the simple interpretation of reduction in model complexity.

## D   Examples for Generalizable constraints

In this section, we look at the Rademacher complexity of $\mathcal{G}$ for different explanation constraints to characterize how many samples with explanation constraints are required in order to generalize to satisfying the explanation constraints on unseen data. We remark that this is a different notion of sample complexity; these unlabeled data require annotations of explanation constraints, not standard labels. In practice, this can be easier and less expertise might be necessary if define the surrogate loss $\phi$ directly. First, we analyze the case where our explanation is given by the gradient of a linear model.

**Proposition D.1** (Learning a gradient constraint for linear models)**.** *Let $\mathcal{D}$ be a distribution over $\mathbb{R}^d$. Let $\mathcal{H} = \{h : x \mapsto \langle w_h, x \rangle \mid w_h \in \mathbb{R}^d, \|w_h\|_2 \leq B\}$ be a class of linear models that pass through the origin. Let $\phi(h, x) = \theta(w_h, w_{h'})$ be a surrogate explanation loss. Let $\mathcal{G} = \{\phi(h, \cdot) \mid h \in \mathcal{H}\}$, then we have*

$$R_n(\mathcal{G}) \leq \frac{\pi}{2\sqrt{m}}.$$

*Proof.* For a linear separator, $\phi(h, \cdot)$ is a constant function over $\mathcal{X}$. The Rademacher complexity is given by

$$
\begin{aligned}
R_n(\mathcal{G}) &= \mathop{\mathbb{E}}_{x \sim D} \left[ \mathop{\mathbb{E}}_{\sigma} \left[ \sup_{\phi(h, \cdot) \in \mathcal{G}} \left( \frac{1}{m} \sum_{i=1}^m \sigma_i \phi(h, x_i) \right) \right] \right] \\
&= \mathop{\mathbb{E}}_{x \sim D} \left[ \mathop{\mathbb{E}}_{\sigma} \left[ \sup_{h \in \mathcal{H}} \left( \frac{1}{m} \sum_{i=1}^m \sigma_i \right) \theta(w_h, w_{h'}) \right] \right] \\
&= \mathop{\mathbb{E}}_{x \sim D} \left[ \mathop{\mathbb{E}}_{\sigma} \left[ \left( \frac{1}{m} \sum_{i=1}^m \sigma_i \right) \sup_{h \in \mathcal{H}} \theta(w_h, w_{h'}) \right] \right] \\
&= \frac{\pi}{2} \mathop{\mathbb{E}}_{\sigma} \left[ \left| \frac{1}{m} \sum_{i=1}^m \sigma_i \right| \right] \\
&\leq \frac{\pi}{2\sqrt{m}}.
\end{aligned}
$$

$\square$

We compare this with the Rademacher complexity of linear models which is given by $R_m(\mathcal{H}) \leq \frac{B}{\sqrt{m}}$. The upper bound does not depend on the upper bound on the weight $B$. In practice, we know that the gradient of a linear model is constant for any data point. This implies that knowing a gradient of a single point is enough to identify the gradient of the linear model.

We consider another type of explanation constraint that is given by a noisy model. Here, we could observe either a noisy classifier and noisy regressor, and the constraint could be given by having similar outputs to this noisy model. This is reminiscent of learning with noisy labels [27] or weak supervision [30, 29, 28]. In this case, our explanation $g$ is simply the hypothesis element $h$ itself, and our constraint is on the values that $h(x)$ can take. We first analyze this in the classification setting.

**Proposition D.2** (Learning a constraint given by a noisy classifier)**.** *Let $\mathcal{D}$ be a distribution over $\mathbb{R}^d$. Consider a binary classification task with $\mathcal{Y} = \{-1, 1\}$. Let $\mathcal{H}$ be a hypothesis class. Let $\phi(h, x) = 1[h(x) \neq h'(x)]$ be a surrogate explanation loss. Let $\mathcal{G} = \{\phi(h, \cdot) \mid h \in \mathcal{H}\}$, then we have*

$$R_n(\mathcal{G}) = \frac{1}{2} R_n(\mathcal{H}).$$

*Proof.*

$$
\begin{aligned}
R_n(\mathcal{G}) &= \mathop{\mathbb{E}}_{x \sim D} \left[ \mathop{\mathbb{E}}_{\sigma} \left[ \sup_{\phi(h, \cdot) \in \mathcal{G}} \left( \frac{1}{m} \sum_{i=1}^m \sigma_i \phi(h, x_i) \right) \right] \right] \\
&= \mathop{\mathbb{E}}_{x \sim D} \left[ \mathop{\mathbb{E}}_{\sigma} \left[ \sup_{h \in \mathcal{H}} \left( \frac{1}{m} \sum_{i=1}^m \sigma_i \left( \frac{1 - h(x)h'(x)}{2} \right) \right) \right] \right] \\
&= \mathop{\mathbb{E}}_{x \sim D} \left[ \mathop{\mathbb{E}}_{\sigma} \left[ \sup_{h \in \mathcal{H}} \left( \frac{1}{m} \sum_{i=1}^m \sigma_i \left( \frac{h(x)h'(x)}{2} \right) \right) \right] \right] \\
&= \mathop{\mathbb{E}}_{x \sim D} \left[ \mathop{\mathbb{E}}_{\sigma} \left[ \sup_{h \in \mathcal{H}} \left( \frac{1}{m} \sum_{i=1}^m \sigma_i \left( \frac{h(x)}{2} \right) \right) \right] \right] \\
&= \frac{1}{2} R_n(\mathcal{H}).
\end{aligned}
$$

$\square$

Here, to learn the restriction of $\mathcal{G}$ is on the same order of $R_n(\mathcal{H})$. For a given noisy regressor, we observe slightly different upper bound.

**Proposition D.3** (Learning a constraint given by a noisy regressor). *Let $\mathcal{D}$ be a distribution over $\mathbb{R}^d$. Consider a regression task with $\mathcal{Y} = \mathbb{R}$. Let $\mathcal{H}$ be a hypothesis class that $\forall h \in \mathcal{H}, -h \in \mathcal{H}$. Let $\phi(h, x) = |h(x) - h'(x)|$ be a surrogate explanation loss. Let $\mathcal{G} = \{\phi(h, \cdot) \mid h \in \mathcal{H}\}$, then we have*

$$R_n(\mathcal{G}) \leq 2R_n(\mathcal{H}).$$

*Proof.*

$$
\begin{aligned}
R_n(\mathcal{G}) &= \mathop{\mathbb{E}}_{x \sim D}\left[\mathop{\mathbb{E}}_{\sigma}\left[\sup_{\phi(h,\cdot) \in \mathcal{G}}\left(\frac{1}{m}\sum_{i=1}^{m}\sigma_i \phi(h, x_i)\right)\right]\right] \\
&= \mathop{\mathbb{E}}_{x \sim D}\left[\mathop{\mathbb{E}}_{\sigma}\left[\sup_{h \in \mathcal{H}}\left(\frac{1}{m}\sum_{i=1}^{m}\sigma_i |h(x_i) - h'(x_i)|\right)\right]\right] \\
&= \mathop{\mathbb{E}}_{x \sim D}\left[\mathop{\mathbb{E}}_{\sigma}\left[\sup_{h \in \mathcal{H}}\left(\frac{1}{m}\sum_{i=1}^{m}\sigma_i \max(0, h(x_i) - h'(x_i)) + \frac{1}{m}\sum_{i=1}^{m}\sigma_i \max(0, h'(x_i) - h(x_i))\right)\right]\right] \\
&\leq \mathop{\mathbb{E}}_{x \sim D}\left[\mathop{\mathbb{E}}_{\sigma}\left[\sup_{h \in \mathcal{H}}\left(\frac{1}{m}\sum_{i=1}^{m}\sigma_i \max(0, h(x_i) - h'(x_i))\right)\right]\right] + \\
&\qquad \mathop{\mathbb{E}}_{x \sim D}\left[\mathop{\mathbb{E}}_{\sigma}\left[\sup_{h \in \mathcal{H}}\left(\frac{1}{m}\sum_{i=1}^{m}\sigma_i \max(0, h'(x_i) - h(x_i))\right)\right]\right] \\
&\leq \mathop{\mathbb{E}}_{x \sim D}\left[\mathop{\mathbb{E}}_{\sigma}\left[\sup_{h \in \mathcal{H}}\left(\frac{1}{m}\sum_{i=1}^{m}\sigma_i (h(x_i) - h'(x_i))\right)\right]\right] + \mathop{\mathbb{E}}_{x \sim D}\left[\mathop{\mathbb{E}}_{\sigma}\left[\sup_{h \in \mathcal{H}}\left(\frac{1}{m}\sum_{i=1}^{m}\sigma_i (h'(x_i) - h(x_i))\right)\right]\right],
\end{aligned}
$$

where in the last line, we apply Talgrand's lemma A.4 and note that the max function $\max(0, h(x))$ is 1-Lipschitz; in the third line, we note that we break up the supremum as both terms by definition of the $\max$ function are non-negative. Then, noting that we do not optimize over $h'(x)$, we further simplify this as

$$
\begin{aligned}
R_n(\mathcal{G}) &\leq \mathop{\mathbb{E}}_{x \sim D}\left[\mathop{\mathbb{E}}_{\sigma}\left[\sup_{h \in \mathcal{H}}\left(\frac{1}{m}\sum_{i=1}^{m}\sigma_i h(x_i)\right)\right]\right] + \mathop{\mathbb{E}}_{x \sim D}\left[\mathop{\mathbb{E}}_{\sigma}\left[\sup_{h \in \mathcal{H}}\left(\frac{1}{m}\sum_{i=1}^{m}\sigma_i(-h(x_i))\right)\right]\right] \\
&\leq 2R_n(\mathcal{H}).
\end{aligned}
$$

$\square$

As mentioned before, knowing apriori surrogate loss $\phi$ might be too strong. In practice, we may only have access to the instances $\phi(\cdot, x_i)$ on a set of samples $S = \{x_1, \ldots, x_k\}$. We also consider the case when $\phi(h, x) = |h(x) - h'(x)|$ when $h'$ is unknown and $h'$ belongs to a learnable class $\mathcal{C}$.

**Proposition D.4** (Learning a constraint given by a noisy regressor from some learnable class $\mathcal{C}$). *Assume $\mathcal{D}$ is a distribution over $\mathbb{R}^d$. Let $\mathcal{H}$ and $\mathcal{D}$ be hypothesis classes. Let $\phi_{h'}(h, x) = |h(x) - h'(x)|$ be a surrogate explanation loss of a constraint corresponding to $h'$. Let $\mathcal{G}_{\mathcal{C}} = \{\phi_{h'}(h, \cdot) \mid h \in \mathcal{H}, h' \in \mathcal{C}\}$, then we have*

$$R_n(\mathcal{G}_{\mathcal{C}}) \leq 2R_n(\mathcal{H}) + 2R_n(\mathcal{C}).$$

*Proof.*

$$R_n(\mathcal{G}_\mathcal{C}) = \mathop{\mathbb{E}}_{x \sim D}\left[\mathop{\mathbb{E}}_\sigma\left[\sup_{\phi(h,\cdot) \in \mathcal{G}_\mathcal{C}}\left(\frac{1}{m}\sum_{i=1}^m \sigma_i \phi(h, x_i)\right)\right]\right]$$

$$= \mathop{\mathbb{E}}_{x \sim D}\left[\mathop{\mathbb{E}}_\sigma\left[\sup_{\substack{h \in \mathcal{H}, \\ h' \in \mathcal{C}}}\left(\frac{1}{m}\sum_{i=1}^m \sigma_i |h(x_i) - h'(x_i)|\right)\right]\right]$$

$$\leq \mathop{\mathbb{E}}_{x \sim D}\left[\mathop{\mathbb{E}}_\sigma\left[\sup_{\substack{h \in \mathcal{H}, \\ h' \in \mathcal{C}}}\left(\frac{1}{m}\sum_{i=1}^m \sigma_i \max(0, h(x_i) - h'(x_i))\right)\right]\right] \quad +$$

$$\mathop{\mathbb{E}}_{x \sim D}\left[\mathop{\mathbb{E}}_\sigma\left[\sup_{\substack{h \in \mathcal{H}, \\ h' \in \mathcal{C}}}\left(\frac{1}{m}\sum_{i=1}^m \sigma_i \max(0, h'(x_i) - h(x_i))\right)\right]\right]$$

$$\leq \mathop{\mathbb{E}}_{x \sim D}\left[\mathop{\mathbb{E}}_\sigma\left[\sup_{\substack{h \in \mathcal{H}, \\ h' \in \mathcal{C}}}\left(\frac{1}{m}\sum_{i=1}^m \sigma_i (h(x_i) - h'(x_i))\right)\right]\right] \quad +$$

$$\mathop{\mathbb{E}}_{x \sim D}\left[\mathop{\mathbb{E}}_\sigma\left[\sup_{\substack{h \in \mathcal{H}, \\ h' \in \mathcal{C}}}\left(\frac{1}{m}\sum_{i=1}^m \sigma_i (h'(x_i) - h(x_i))\right)\right]\right]$$

where the lasts line again holds by an application of Talgrand's lemma. In this case, we indeed are optimizing over $h'$, so we get that

$$R_n(\mathcal{G}_\mathcal{C}) \leq 2 \cdot \mathop{\mathbb{E}}_{x \sim D}\left[\mathop{\mathbb{E}}_\sigma\left[\sup_{h \in \mathcal{H}}\left(\frac{1}{m}\sum_{i=1}^m \sigma_i(h(x_i))\right)\right]\right] + 2 \cdot \mathop{\mathbb{E}}_{x \sim D}\left[\mathop{\mathbb{E}}_\sigma\left[\sup_{h' \in \mathcal{C}}\left(\frac{1}{m}\sum_{i=1}^m \sigma_i(h'(x_i))\right)\right]\right]$$

$$= 2R_n(\mathcal{H}) + 2R_n(\mathcal{C}).$$

$\square$

We remark that while this value is much larger than that of $R_n(\mathcal{H})$, we only need information about $\phi(h, x)$ and *not* the true label. Therefore, in many cases, this is preferable and not as expensive to learn.

## E   Proof of Theorem 3.2

We consider the agnostic setting of Theorem 3.2. Here, we have two notions of deviations; one is deviation in a model's ability to satisfy explanations, and the other is a model's ability to generalize to correctly produce the target function.

*Proof.* From Rademacher-based uniform convergence, for any $h \in \mathcal{H}$, with probability at least $1 - \delta/2$ over $S_E$

$$|\phi(h, \mathcal{D}) - \phi(h, S_E)| \leq 2R_k(\mathcal{G}) + \sqrt{\frac{\ln(4/\delta)}{2k}} = \varepsilon_k$$

Therefore, with probability at least $1 - \delta/2$, for any $h \in \mathcal{H}_{\phi, t - \varepsilon_k}$ we also have $\phi(h, S_E) \leq t$ and for any $h$ with $\phi(h, S_E) \leq t$, we have $h \in \mathcal{H}_{\phi, t + \varepsilon_k}$. In addition, by a uniform convergence bound, with probability at least $1 - \delta/2$, for any $h \in \mathcal{H}_{\phi, t + \varepsilon_k}$

$$|\text{err}_\mathcal{D}(h) - \text{err}_S(h)| \leq R_n(\mathcal{H}_{\phi, t + \varepsilon_k}) + \sqrt{\frac{\ln(4/\delta)}{2n}}.$$

Now, let $h'$ be the minimizer of $\mathrm{err}_S(h)$ given that $\phi(h, S_E) \leq t$. By previous results, with probability $1 - \delta$, we have $h' \in \mathcal{H}_{\phi, t+\varepsilon_k}$ and

$$\mathrm{err}_\mathcal{D}(h') \leq \mathrm{err}_S(h') + R_n(\mathcal{H}_{\phi, t+\varepsilon_k}) + \sqrt{\frac{\ln(4/\delta)}{2n}}$$

$$\leq \mathrm{err}_S(h^*_{t-\varepsilon_k}) + R_n(\mathcal{H}_{\phi, t+\varepsilon_k}) + \sqrt{\frac{\ln(4/\delta)}{2n}}$$

$$\leq \mathrm{err}_\mathcal{D}(h^*_{t-\varepsilon_k}) + 2R_n(\mathcal{H}_{\phi, t+\varepsilon_k}) + 2\sqrt{\frac{\ln(4/\delta)}{2n}}.$$

$\square$

# F  EPAC-PAC learnability

We note that in our definition of EPAC learnability, we are only concerned with whether a model achieves a lower surrogate loss than $\tau$. However, the objective of minimizing the EPAC-ERM objective is to achieve both low average error and low surrogate loss. We characterize this property as EPAC-PAC learnability.

**Definition 8** (EPAC-PAC learnability). *For any $\delta \in (0, 1), \tau > 0$, the sample complexity of $(\delta, \tau, \gamma)$ - EPAC learning of $\mathcal{H}$ with respect to a surrogate loss $\phi$, denoted $m(\delta, \tau, \gamma; \mathcal{H}, \phi)$ is defined as the smallest $m \in \mathbb{N}$ for which there exists a learning rule $\mathcal{A}$ such that every data distribution $\mathcal{D}_\mathcal{X}$ over $\mathcal{X}$, with probability at least $1 - \delta$ over $S \sim \mathcal{D}^m$,*

$$\phi(\mathcal{A}(S), \mathcal{D}) \leq \inf_{h \in \mathcal{H}} \phi(h, \mathcal{D}) + \tau$$

*and*

$$\mathrm{err}_\mathcal{D}(\mathcal{A}(S)) \leq \inf_{h \in \mathcal{H}} \mathrm{err}_\mathcal{D}(h) + \gamma.$$

*If no such $m$ exists, define $m(\delta, \tau, \gamma; \mathcal{H}, \phi) = \infty$. We say that $\mathcal{H}$ is EPAC-PAC learnable in the agnostic setting with respect to a surrogate loss $\phi$ if $\forall \delta \in (0, 1), \tau > 0$, $m(\delta, \tau, \gamma; \mathcal{H}, \phi)$ is finite.*

# G  A Generalization Bound in the Realizable Setting

In this section, we assume that we are in the doubly realizable [2] setting where there exists $h^* \in \mathcal{H}$ such that $\mathrm{err}_\mathcal{D}(h^*) = 0$ and $\phi(h^*, \mathcal{D}) = 0$. The optimal classifier $h^*$ lies in $\mathcal{H}$ and also achieve zero expected explanation loss. In this case, we want to output a hypothesis $h$ that achieve both zero empirical risk and empirical explanation risk.

**Theorem G.1** (Generalization bound for the doubly realizable setting). *For a hypothesis class $\mathcal{H}$, a distribution $\mathcal{D}$ and an explanation loss $\phi$. Assume that there exists $h^* \in \mathcal{H}$ that $\mathrm{err}_\mathcal{D}(h^*) = 0$ and $\phi(h^*, \mathcal{D}) = 0$. Let $S = \{(x_1, y_1), \ldots, (x_n, y_n)\}$ is drawn i.i.d. from $\mathcal{D}$ and $S_E = \{x'_1, \ldots, x'_k\}$ drawn i.i.d. from $\mathcal{D}_\mathcal{X}$. With probability at least $1 - \delta$, for any $h \in \mathcal{H}$ that $\mathrm{err}_S(h) = 0$ and $\phi(h, S_E) = 0$, we have*

$$\mathrm{err}_D(h) \leq R_n(\mathcal{H}_{\phi, \varepsilon_k}) + \sqrt{\frac{\ln(2/\delta)}{2n}}$$

*when*

$$\varepsilon_k = 2R_k(\mathcal{G}) + \sqrt{\frac{\ln(2/\delta)}{2k}}$$

*when $\mathcal{G} = \{\phi(h, x) \mid h \in \mathcal{H}, x \in \mathcal{X}\}$.*

*Proof.* We first consider only classifiers than has low empirical explanation loss and then perform standard supervised learning. From Rademacher-based uniform convergence, for any $h \in \mathcal{H}$, with probability at least $1 - \delta/2$ over $S_E$

$$\phi(h, \mathcal{D}) \leq \phi(h, S_E) + 2R_k(\mathcal{G}) + \sqrt{\frac{\ln(2/\delta)}{2k}}$$

when $\mathcal{G} = \{\phi(h, x) \mid h \in \mathcal{H}, x \in \mathcal{X}\}$. Therefore, for any $h \in \mathcal{H}$ with $\phi(h, S_E) = 0$, we have $h \in \mathcal{H}_{\phi, \varepsilon_k}$ with probability at least $1 - \delta/2$. Now, we can apply the uniform convergence on $\mathcal{H}_{\phi, \varepsilon_k}$. For any $h \in \mathcal{H}_{\phi, \varepsilon_k}$ with $\mathrm{err}_S(h) = 0$, with probability at least $1 - \delta/2$, we have

$$\mathrm{err}_{\mathcal{D}}(h) \leq R_n(\mathcal{H}_{\phi, \varepsilon_k}) + \sqrt{\frac{\ln(2/\delta)}{2n}}.$$

Therefore, for $h \in \mathcal{H}$ that $\phi(h, S_E) = 0, \mathrm{err}_S(h) = 0$, we have our desired guarantee. $\qquad\square$

We remark that, since our result relies on the underlying techniques of the Rademacher complexity, our result is on the order of $O(\frac{1}{\sqrt{n}})$. In the (doubly) realizable setting, this is somewhat loose, and more complicated techniques are required to produce tighter bounds. We leave this as an interesting direction for future work.

## H   Rademacher Complexity of Linear Models with a Gradient Constraint

We calculate the empirical Rademacher complexity of a linear model under a gradient constraint.

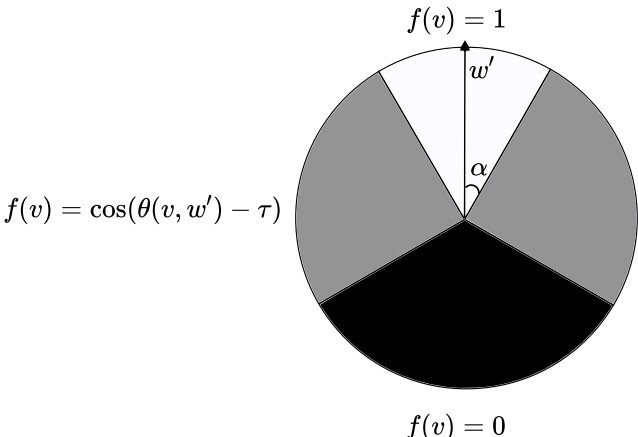

$$f(v) = 1$$
$$w'$$
$$\alpha$$
$$f(v) = \cos(\theta(v, w') - \tau)$$
$$f(v) = 0$$

Figure 7: Illustration of different value of a function $f(v)$.

**Theorem H.1** (Empirical Rademacher complexity of linear models with a gradient constraint). *Let $\mathcal{X}$ be an instance space in $\mathbb{R}^d$, let $\mathcal{D}_{\mathcal{X}}$ be a distribution on $\mathcal{X}$, let $\mathcal{H} = \{h : x \to \langle w_h, x \rangle \mid w_h \in \mathbb{R}^d, \|w_h\|_2 \leq B\}$ be a class of linear model with weights bounded by some constant $B > 0$ in $\ell_2$ norm. Assume that there exists a constant $C > 0$ such that $\mathbb{E}_{x \sim \mathcal{D}_{\mathcal{X}}}[\|x\|_2^2] \leq C^2$. Assume that we have an explanation constraint in terms of gradient constraint; we want the gradient of our linear model to be close to the gradient of some linear model $h'$. Let $\phi(h, x) = \theta(w_h, w_{h'})$ be an explanation surrogate loss when $\theta(u, v)$ is an angle between $u, v$. For any $S = \{x_1, \dots, x_n\}$ is drawn i.i.d. from $\mathcal{D}_{\mathcal{X}}$, we have*

$$R_S(\mathcal{H}_{\phi, \tau}) = \frac{B}{n} \mathbb{E}_\sigma \left[ \|v\| f(v) \right].$$

*when $v = \sum_{i=1}^n x_i \sigma_i$ and*

$$f(v) = \begin{cases} 1 & \text{when } \theta(v, w') \leq \tau \\ \cos(\theta(v, w') - \tau) & \text{when } \tau \leq \theta(v, w') \leq \frac{\pi}{2} + \tau \\ 0 & \text{when } \theta(v, w') \geq \frac{\pi}{2} + \tau. \end{cases}$$

For the proof, we refer to Appendix H for full proof. We compare this with the standard bound on linear models which is given by

$$R_S(\mathcal{H}) = \frac{B}{n} \mathbb{E}_\sigma [\|v\|].$$

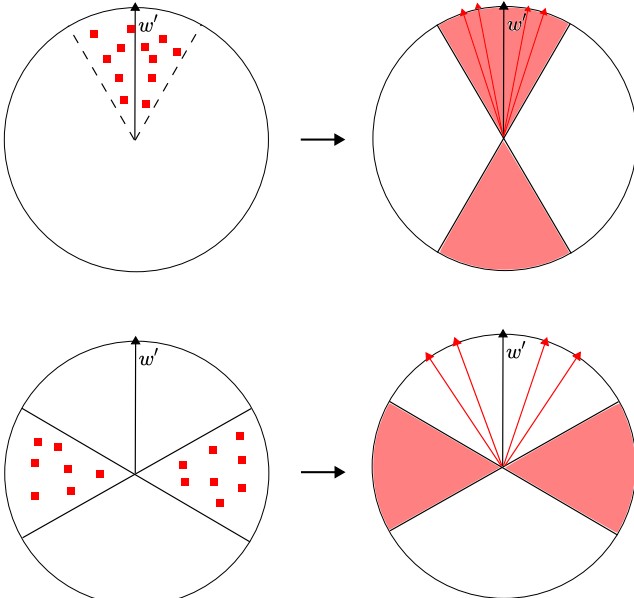

Figure 8: Benefits of an explanation constraint also depend on the data distribution. We represent data points $x_i$ with red squares (Left). The possible regions for $v = \sum_{i=1}^{n} x_i \sigma_i$ are the shaded areas (Right). When the data is highly correlated with $w'$, $v$ would lie in a region where $f(v)$ is large (Top) and this implies that the gradient constraints provide less benefits. On the other hand, when the data is almost orthogonal to $w'$, $v$ would lie in a region with a small value of $f(v)$ (Bottom) which leads to more benefits from the gradient constraints

.

The benefits of the explanation constraints depend on the underlying data distribution; in the case of linear models with a gradient constraint, this depends on an angle between $v = \sum_{i=1}^{n} x_i \sigma_i$ and $w'$. The explanation constraint reduces the term inside the expectation by a factor of $f(v)$ depending on $\theta(v, w')$. When $\theta(v, w') \leq \tau$ then $f(v) = 1$ which implies that there is no reduction. The value of $f(v)$ decreases as the angle between $\theta(v, w')$ increases and reaches $f(v) = 0$ when $\theta(v, w') \geq \frac{\pi}{2} + \tau$. When the data is concentrated around the area of $w'$, the possible regions for $v$ would be close to $w'$ or $-w'$ (Figure 8 (Top)). The value of $f(v)$ in this region would be either $1$ or $0$ and the reduction would be $\frac{1}{2}$ on average. In essence, this means that the gradient constraint of being close to $w'$ does not actually tell us much information beyond the information from the data distribution. On the other hand, when the data points are nearly orthogonal to $w'$, the possible regions for $v$ would lead to a small $f(v)$ (Figure 8 (Bottom)). This can lead to a large reduction in complexity. Intuitively, when the data is nearly orthogonal to $w'$, there are many valid linear models including those not close in angle to $w'$. The constraints allows us to effectively shrink down the class of linear models that are close to $w'$.

*Proof.* (Proof of Theorem H.1) Recall that $\mathcal{H}_{\phi,\tau} = \{h : x \rightarrow \langle w_h, x \rangle \mid w_h \in \mathbb{R}^d, \|w_h\|_2 \leq B, \theta(w_h, w_{h'}) \leq \tau\}$. For a set of sample $S$, the empirical Rademacher complexity of $\mathcal{H}_{\phi,\tau}$ is given by

$$R_S(\mathcal{H}_{\phi,\tau}) = \frac{1}{n} \mathbb{E}_\sigma \left[ \sup_{h \in \mathcal{H}_{\phi,\tau}} \sum_{i=1}^{n} h(x_i) \sigma_i \right]$$

$$= \frac{1}{n} \mathbb{E}_\sigma \left[ \sup_{\substack{\|w_h\|_2 \leq B \\ \theta(w_h, w_{h'}) \leq \tau}} \sum_{i=1}^{n} \langle w_h, x_i \rangle \sigma_i \right]$$

$$= \frac{1}{n} \mathbb{E}_\sigma \left[ \sup_{\substack{\|w_h\|_2 \leq B \\ \theta(w_h, w_{h'}) \leq \tau}} \langle w_h, \sum_{i=1}^{n} x_i \sigma_i \rangle \right].$$

For a vector $w' \in \mathbb{R}^d$ with $\|w'\|_2 = 1$, and a vector $v \in \mathbb{R}^d$, we will claim the following,

1. If $\theta(v, w') \leq \tau$, we have
$$\sup_{\substack{\|w\|_2 \leq B \\ \theta(w,w') \leq \tau}} \langle w, v \rangle = B\|v\|.$$

2. If $\frac{\pi}{2} + \tau \leq \theta(v, w') \leq \pi$, we have
$$\sup_{\substack{\|w\|_2 \leq B \\ \theta(w,w') \leq \tau}} \langle w, v \rangle = 0.$$

3. If $\tau \leq \theta(v, w') \leq \frac{\pi}{2} + \tau$, we have
$$\sup_{\substack{\|w\|_2 \leq B \\ \theta(w,w') \leq \tau}} \langle w, v \rangle = B\|v\| \cos(\theta(v, w') - \tau)$$

For the first claim, we can see that if $\theta(v, w') \leq \tau$, we can pick $w = \frac{Bv}{\|v\|}$ and achieve the optimum value. For the second claim, we use the fact that $\theta(\cdot, \cdot)$ satisfies a triangle inequality and for any $w$ that $\theta(w, w') \leq \tau$, we have

$$\theta(v, w) + \theta(w, w') \geq \theta(v, w')$$
$$\theta(v, w) \geq \theta(v, w') - \theta(w, w')$$
$$\theta(v, w) \geq \frac{\pi}{2} + \tau - \tau = \frac{\pi}{2}.$$

This implies that for any $w$ that $\theta(w, w') \leq \tau$, we have $\langle w, v \rangle = \|w\|\|v\| \cos(\theta(v, w)) \leq 0$ and the supremum is given by 0 where we can set $\|w\| = 0$. For the third claim, we know that $\langle w, v \rangle$ is maximum when the angle between $v, w$ is the smallest. From the triangle inequality above, we must have $\theta(w, w') = \tau$ to be the largest possible value so that we have the smallest lower bound $\theta(v, w) \geq \theta(v, w') - \theta(w, w')$. In addition, the inequality holds when $v, w', w$ lie on the same plane. Since we do not have further restrictions on $w$, there exists such $w$ and we have

$$\sup_{\substack{\|w\|_2 \leq B \\ \theta(w,w') \leq \tau}} \langle w, v \rangle = B\|v\| \cos(\theta(v, w') - \tau)$$

as required. One can calculate a closed form formula for $w$ by solving a quadratic equation. Let $w = \frac{B\tilde{w}}{\|\tilde{w}\|}$ when $\tilde{w} = v + \lambda w'$ for some constant $\lambda > 0$ such that $\theta(w, w') = \tau$. With this we have an equation

$$\frac{\langle \tilde{w}, w' \rangle}{\|\tilde{w}\|} = \cos(\tau)$$
$$\frac{\langle v + \lambda w', w' \rangle}{\|v + \lambda w'\|} = \cos(\tau)$$

Let $\mu = \langle v, w' \rangle$, solving for $\lambda$, we have

$$\frac{\mu + \lambda}{\sqrt{\|v\|^2 + 2\lambda\mu + \lambda^2}} = \cos(\tau)$$
$$\mu^2 + 2\mu\lambda + \lambda^2 = \cos^2(\tau)(\|v\|^2 + 2\lambda\mu + \lambda^2)$$
$$\sin^2(\tau)\lambda^2 + 2\sin^2(\tau)\mu\lambda + \mu^2 - \cos^2(\tau)\|v\|^2 = 0$$
$$\lambda^2 + 2\mu\lambda + \frac{\mu^2}{\sin^2(\tau)} - \cot^2(\tau)\|v\|^2 = 0$$

Solve this quadratic equation, we have

$$\lambda = -\mu \pm \cot(\tau)\sqrt{\|v\|^2 - \mu^2}.$$

Since $\lambda > 0$, we have $\lambda = -\mu + \cot(\tau)\sqrt{\|v\|^2 - \mu^2}$. We have

$$
\begin{aligned}
\tilde{w} &= v + \lambda w' \\
&= v + (-\mu + \cot(\tau)\sqrt{\|v\|^2 - \mu^2})w' \\
&= v - \langle v, w'\rangle w' + \cot(\tau)w'\sqrt{\|v\|^2 - \mu^2}.
\end{aligned}
$$

With these claims, we have

$$
\begin{aligned}
R_S(\mathcal{H}_{\phi,\tau}) &= \frac{1}{n}\mathbb{E}_\sigma\left[\sup_{\substack{\|w_h\|_2 \leq B \\ \theta(w_h, w_{h'}) \leq \tau}} \langle w_h, \sum_{i=1}^n x_i\sigma_i\rangle\right] \\
&= \frac{B}{n}\mathbb{E}_\sigma\left[\|v\|1\{\theta(v, w') \leq \tau\} + \|v\|1\{\tau \leq \theta(v, w') \leq \frac{\pi}{2} + \tau\}\cos(\theta(v, w') - \tau)\right] \\
&= \frac{B}{n}\mathbb{E}_\sigma\left[\|v\|f(v)\right].
\end{aligned}
$$

$\square$

**Theorem H.2** (Rademacher complexity of linear models with gradient constraint, uniform distribution on a sphere). *Let $\mathcal{X}$ be an instance space in $\mathbb{R}^d$, let $\mathcal{D}_\mathcal{X}$ be a uniform distribution on a unit sphere in $\mathbb{R}^d$, let $\mathcal{H} = \{h : x \to \langle w_h, x\rangle \mid w_h \in \mathbb{R}^d, \|w_h\|_2 \leq B\}$ be a class of linear model with weights bounded by some constant $B > 0$ in $\ell_2$ norm. Assume that there exists a constant $C > 0$ such that $\mathbb{E}_{x \sim \mathcal{D}_\mathcal{X}}[\|x\|_2^2] \leq C^2$. Assume that we have an explanation constraint in terms of gradient constraint; we want the gradient of our linear model to be close to the gradient of some linear model $h'$. Let $\phi(h, x) = \theta(w_h, w_{h'})$ be an explanation surrogate loss when $\theta(u, v)$ is an angle between $u, v$. We have*

$$
R_n(\mathcal{H}_{\phi,\tau}) = \frac{B}{\sqrt{n}}\left(\sin(\tau) \cdot p + \frac{1-p}{2}\right),
$$

*where*

$$
p = \operatorname{erf}\left(\frac{\sqrt{d}\sin(\tau)}{\sqrt{2}}\right).
$$

*Proof.* From Theorem H.1, we have that

$$
\begin{aligned}
R_n(\mathcal{H}_{\phi,\tau}) &= \mathbb{E}[R_S(\mathcal{H}_{\phi,\tau})] \\
&= \frac{B}{n}\mathbb{E}_\mathcal{D}\left[\mathbb{E}_\sigma\left[\|v\|1\{\theta(v, w') \leq \tau\} + \|v\|1\{\tau \leq \theta(v, w') \leq \frac{\pi}{2} + \tau\}\cos(\theta(v, w') - \tau)\right]\right] \\
&= \frac{B}{n}\mathbb{E}_\mathcal{D}\left[\mathbb{E}_\sigma\left[\|v\|1\{\theta(v, w') \leq \frac{\pi}{2} - \tau\} + \|v\|1\{\frac{\pi}{2} - \tau \leq \theta(v, w') \leq \frac{\pi}{2} + \tau\}\cos(\theta(v, w') - \tau)\right]\right]
\end{aligned}
$$

when $v = \sum_{i=1}^n x_i\sigma_i$. Because $x_i$ is drawn uniformly from a unit sphere, in expectation $\theta(v, w')$ has a uniform distribution over $[0, \pi]$ and the distribution $\|v\|$ for a fixed value of $\theta(v, w')$ are the same for all $\theta(v, w') \in [0, \pi]$. From Trigonometry, we note that

$$
\cos(\frac{\pi}{2} - 2\tau + a) + \cos(\frac{\pi}{2} - a) = \sin(2\tau - a) + \sin(a) \leq 2\sin(\tau).
$$

By the symmetry property and the uniformity of the distribution of $\theta(v, w')$ and $\|v\|$.

$$\mathbb{E}_{\mathcal{D}}\left[\mathbb{E}_\sigma\left[\|v\|\mathbb{1}\{\frac{\pi}{2} - \tau \le \theta(v, w') \le \frac{\pi}{2} + \tau\}\cos(\theta(v, w') - \tau)\right]\right]$$

$$= \mathbb{E}_{\mathcal{D}}\left[\mathbb{E}_\sigma\left[\|v\|\mathbb{1}\{0 \le \theta(v, w') \le 2\tau\}\cos(\frac{\pi}{2} + \theta(v, w') - \tau)\right]\right]$$

$$= \mathbb{E}_{\mathcal{D}}\left[\mathbb{E}_\sigma\left[\|v\|(\mathbb{1}\{0 \le \theta(v, w') \le \tau\}\cos(\frac{\pi}{2} + \theta(v, w') - \tau) + \mathbb{1}\{\tau \le \theta(v, w') \le 2\tau\}\cos(\frac{\pi}{2} + \theta(v, w') - \tau))\right]\right]$$

$$= \mathbb{E}_{\mathcal{D}}\left[\mathbb{E}_\sigma\left[\|v\|(\mathbb{1}\{0 \le \theta(v, w') \le \tau\}\cos(\frac{\pi}{2} + \theta(v, w') - \tau) + \mathbb{1}\{0 \le 2\tau - \theta(v, w') \le \tau\}\cos(\frac{\pi}{2} - (2\tau - \theta(v, w'))))\right]\right]$$

$$= \mathbb{E}_{\mathcal{D}}\left[\mathbb{E}_\sigma\left[\|v\|(\mathbb{1}\{0 \le \theta(v, w') \le \tau\}\cos(\frac{\pi}{2} + \theta(v, w') - \tau) + \mathbb{1}\{0 \le \tilde\theta(v, w') \le \tau\}\cos(\frac{\pi}{2} - \tilde\theta(v, w')))\right]\right]$$

$$= \mathbb{E}_{\mathcal{D}}\left[\mathbb{E}_\sigma\left[\|v\|(\mathbb{1}\{0 \le \theta(v, w') \le \tau\}\cos(\frac{\pi}{2} + \theta(v, w') - \tau) + \mathbb{1}\{0 \le \theta(v, w') \le \tau\}\cos(\frac{\pi}{2} - \theta(v, w')))\right]\right]$$

$$\le \mathbb{E}_{\mathcal{D}}\left[\mathbb{E}_\sigma\left[\|v\|\mathbb{1}\{\frac{\pi}{2} - \tau \le \theta(v, w') \le \frac{\pi}{2} + \tau\}\sin(\tau)\right]\right]$$

when $\tilde\theta(v, w') = \frac{\pi}{2} - \theta(v, w')$. We have

$$R_n(\mathcal{H}_{\phi,\tau}) \le \frac{B}{n}\mathbb{E}_{\mathcal{D}}\left[\mathbb{E}_\sigma\left[\|v\|\mathbb{1}\{\theta(v, w') \le \frac{\pi}{2} - \tau\} + \|v\|\mathbb{1}\{\frac{\pi}{2} - \tau \le \theta(v, w') \le \frac{\pi}{2} + \tau\}\sin(\tau)\right]\right]$$

$$= \frac{B}{n}\mathbb{E}_{\mathcal{D}}\left[\mathbb{E}_\sigma\left[\|v\|\right]\right]\left(\Pr(\theta(v, w') \le \frac{\pi}{2} - \tau) + \Pr(\frac{\pi}{2} - \tau \le \theta(v, w') \le \frac{\pi}{2} + \tau)\sin(\tau)\right)$$

The last equation follows from the symmetry and uniformity properties. We can bound the first expectation

$$\mathbb{E}_{\mathcal{D}}[\mathbb{E}_\sigma\|v\|]] = \mathbb{E}_{\mathcal{D}}[\mathbb{E}_\sigma\|\sum_{i=1}^n x_i\sigma_i\|]]$$

$$\le \mathbb{E}_{\mathcal{D}}[\sqrt{\mathbb{E}_\sigma\|\sum_{i=1}^n x_i\sigma_i\|^2]}]$$

$$= \mathbb{E}_{\mathcal{D}}[\sqrt{\mathbb{E}_\sigma\sum_{i=1}^n \|x_i\|^2\sigma_i^2]}]$$

$$\le C\sqrt{n}.$$

Next, we can simply note that, since our data is distributed over a unit sphere, each data has norm no greater than 1. Therefore, we know that $C = 1$ is indeed an upper bound on $\mathbb{E}_{x \sim \mathcal{D}_\mathcal{X}}[\|x\|_2^2]$. For the probability term, we note that in expectation $v$ has the same distribution as a random vector $u$ drawn uniformly from a unit sphere. We let this be some probability $p$:

$$p = \Pr\left(\frac{\pi}{2} - \tau \le \theta(v, w') \le \frac{\pi}{2} + \tau\right) = \Pr\left(|\langle u, w'\rangle| \le \sin(\tau)\right).$$

We know that the projection $\langle u, w'\rangle \sim \mathcal{N}(0, \frac{1}{d})$. Then, we have that $|\langle u, w'\rangle|$ is given by a Folded Normal Distribution, which has a CDF given by

$$\Pr\left(|\langle u, w'\rangle| \le \sin(\tau)\right) = \frac{1}{2}\left[\text{erf}\left(\frac{\sqrt{d}\sin(\tau)}{\sqrt{2}}\right) + \text{erf}\left(\frac{\sqrt{d}\sin(\tau)}{\sqrt{2}}\right)\right]$$

$$= \text{erf}\left(\frac{\sqrt{d}\sin(\tau)}{\sqrt{2}}\right).$$

We then observe that

$$\Pr\left(\theta(v, w') \le \frac{\pi}{2} - \tau\right) = \frac{1}{2}\left(1 - \Pr\left(\frac{\pi}{2} - \tau \le \theta(v, w') \le \frac{\pi}{2} + \tau\right)\right)$$

$$= \frac{1 - p}{2}$$

Plugging this in yields the following bound

$$R_n(\mathcal{H}_{\phi,\tau}) = \frac{B}{\sqrt{n}}\left(\sin(\tau)\cdot p + \frac{1-p}{2}\right),$$

where

$$p = \text{erf}\left(\frac{\sqrt{d}\sin(\tau)}{\sqrt{2}}\right).$$

$\square$

# I  Rademacher Complexity for Two Layer Neural Networks with a Gradient Constraint

Here, we present the full proof of the generalization bound for two layer neural networks with gradient explanations. In our proof, we use two results from Ma [24]. One result is a technical lemma, and the other is a bound on the Rademacher complexity of two layer neural networks.

**Lemma I.1.** *Consider a set $S = \{x_1, ..., x_n\}$ and a hypothesis class $\mathcal{F} \subset \{f : \mathbb{R}^d \to \mathbb{R}\}$. If*

$$\sup_{f\in\mathcal{F}}\sum_{i=1}^{n}f(x_i)\sigma_i \geq 0 \ \text{for any} \ \sigma_i \in \{\pm 1\}, i = 1,...,n,$$

*then, we have that*

$$\mathbb{E}_{\sigma}\left[\sup_{f\in\mathcal{F}}|\sum_{i=1}^{n}f(x_i)\sigma_i|\right] \leq 2\mathbb{E}_{\sigma}\left[\sup_{f\in\mathcal{F}}\sum_{i=1}^{n}f(x_i)\sigma_i\right].$$

**Theorem I.2** (Rademacher complexity for two layer neural networks [24]). *Let $\mathcal{X}$ be an instance space and $\mathcal{D}_{\mathcal{X}}$ be a distribution over $\mathcal{X}$. Let $\mathcal{H} = \{h : x \mapsto \sum_{i=1}^{m} w_i\sigma(u_i^{\top}x)|w_i \in \mathbb{R}, u_i \in \mathbb{R}^d, \sum_{i=1}^{m}|w_i|\|u_i\|_2 \leq B\}$ be a class of two layer neural networks with $m$ hidden nodes with a ReLU activation function $\sigma(x) = \max(0, x)$. Assume that there exists some constant $C > 0$ such that $\mathbb{E}_{x\sim\mathcal{D}_{\mathcal{X}}}[\|x\|_2^2] \leq C^2$. Then, for any $S = \{x_1, \ldots, x_n\}$ is drawn i.i.d. from $\mathcal{D}_{\mathcal{X}}$, we have that*

$$R_S(\mathcal{H}) \leq \frac{2B}{n}\sqrt{\sum_{i=1}^{n}\|x_i\|_2^2}$$

*and*

$$R_n(\mathcal{H}) \leq \frac{2BC}{\sqrt{n}}.$$

We defer interested readers to [24] for the full proof of this result. Here, the only requirement of the data distribution is that $\mathbb{E}_{x\sim\mathcal{D}_{\mathcal{X}}}[\|x\|_2^2] \leq C^2$. We now present our result in the setting of two layer neural networks with one hidden node $m = 1$ to provide clearer intuition for the overall proof.

**Theorem I.3** (Rademacher complexity for two layer neural networks ($m = 1$) with gradient constraints). *Let $\mathcal{X}$ be an instance space and $\mathcal{D}_{\mathcal{X}}$ be a distribution over $\mathcal{X}$. Let $\mathcal{H} = \{h : x \mapsto w\sigma(u^{\top}x)|w \in \mathbb{R}, u \in \mathbb{R}^d, |w| \leq B, \|u\| = 1\}$. Without loss of generality, we assume that $\|u\| = 1$. Assume that there exists some constant $C > 0$ such that $\mathbb{E}_{x\sim\mathcal{D}_{\mathcal{X}}}[\|x\|_2^2] \leq C^2$. Our explanation constraint is given by a constraint on the gradient of our models, where we want the gradient of our learnt model to be close to a particular target function $h' \in \mathcal{H}$. Let this be represented by an explanation loss given by*

$$\phi(h, x) = \|\nabla_x h(x) - \nabla_x h'(x)\|_2 + \infty \cdot 1\{\|\nabla_x h(x) - \nabla_x h'(x)\| > \tau\}$$

*for some $\tau > 0$. Let $h'(x) = w'\sigma((u')^{\top}x)$ the target function, then we have*

$$R_n(\mathcal{H}_{\phi,\tau}) \leq \frac{\tau C}{\sqrt{n}} \qquad \text{if } |w'| > \tau,$$

$$R_n(\mathcal{H}_{\phi,\tau}) \leq \frac{3\tau C}{\sqrt{n}} \qquad \text{if } |w'| \leq \tau.$$

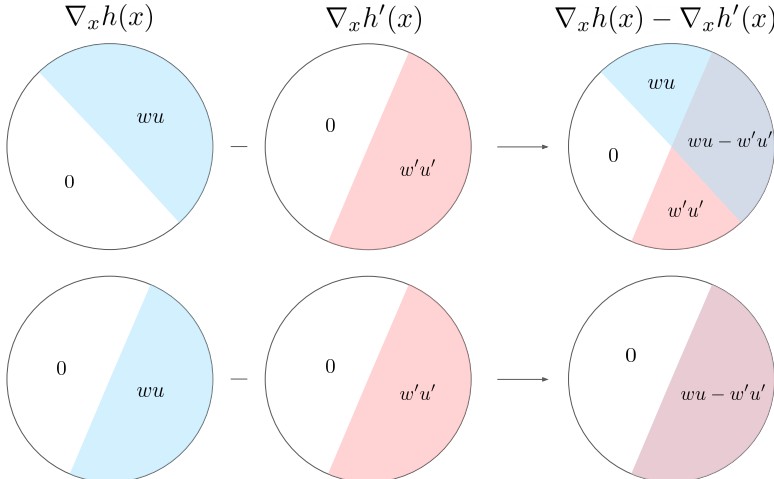

Figure 9: Visualization of the piecewise constant function of $\nabla_x h(x) - \nabla_x h'(x)$ over 4 regions.

*Proof.* Our choice of $\phi(h, x)$ guarantees that, for any $h \in \mathcal{H}_{\phi,\tau}$, we have that $\|\nabla_x h(x) - \nabla_x h'(x)\| \le \tau$ almost everywhere. We note that for $h(x) = w\sigma(u^\top x)$, the gradient is given by $\nabla_x h(x) = wu\mathbf{1}\{u^\top x > 0\}$, which is a piecewise constant function over two regions (i.e., $u^\top x > 0, u^\top x \le 0$), captured by Figure I.

We now consider $\nabla_x h(x) - \nabla_x h'(x)$, and we have 3 possible cases.

**Case 1:** $\theta(u, u') > 0$
This implies that the boundaries of $\nabla_x(h)$ and $\nabla_x h'(x)$ are different. Then, we have that $\nabla_x h(x) - \nabla_x h'(x)$ is a piecewise constant function with 4 regions, taking on values

$$\nabla_x h(x) - \nabla_x h'(x) = \begin{cases} wu - w'u' & \text{when } u^\top x > 0, (u')^\top x > 0 \\ wu & \text{when } u^\top x > 0, (u')^\top x < 0 \\ -w'u' & \text{when } u^\top x < 0, (u')^\top x > 0 \\ 0 & \text{when } u^\top x < 0, (u')^\top x < 0 \end{cases}$$

If we assume that each region has probability mass greater than 0 then our constraint $\|\nabla_x h(x) - \nabla_x h'(x)\|_2 \le \tau$ implies that $|w| = |w|\|u\| \le \tau, |w'| = |w'|\|u'\| \le \tau, \|wu - w'u'\| \le \tau$.

**Case 2:** $\theta(u, u') = 0$
This implies that the boundary of $\nabla_x h(x)$ and $\nabla_x h'(x)$ are the same. Then, $\nabla_x h(x) - \nabla_x h'(x)$ is a piecewise constant over two regions

$$\nabla_x h(x) - \nabla_x h'(x) = \begin{cases} wu - w'u' & \text{when } u^\top x > 0 \\ 0 & \text{when } u^\top x < 0 \end{cases}$$

This gives us that $|w - w'| = \|wu - w'u'\| \le \tau$.

**Case 3:** $\theta(u, u') = \pi$
Here, we have that the decision boundaries of $\nabla_x h(x)$ and $\nabla_x h'(x)$ are the same but the gradients are non-zero on different sides. Then, $\nabla_x h(x) - \nabla_x h'(x)$ is a piecewise constant on two regions

$$\nabla_x h(x) - \nabla_x h'(x) = \begin{cases} wu & \text{when } u^\top x > 0 \\ -w'u' & \text{when } u^\top x < 0 \end{cases}$$

This gives us that $|w| \le \tau$ and $|w'| \le \tau$.

These different cases tell us that the constraint $\|\nabla_x h(x) - \nabla_x h'(x)\| \le \tau$ reduces $\mathcal{H}$ into a class of models follows either

1. $u = u'$ and $|w - w'| < \tau$.

2. $u \neq u'$ and $|w| < \tau$. However, this case only possible when $|w'| < \tau$.

If $|w'| > \tau$, we know that we must only have the first case. Now, we can calculate the Rademacher complexity of our restricted class $\mathcal{H}_{\phi,\tau}$. We will again do this in separate cases.

**Case 1:** $|w'| > \tau$
For any $h \in \mathcal{H}_{\phi,\tau}$, we have that $u = u'$ and $|w - w'| < \tau$. For a sample $S = \{x_1, ..., x_n\}$,

$$R_s(\mathcal{H}_{\phi,\tau}) = \frac{1}{n}\mathbb{E}_\sigma \left[ \sup_{h \in \mathcal{H}_{\phi,\tau}} \sum_{i=1}^n h(x_i)\sigma_i \right]$$

$$= \frac{1}{n}\mathbb{E}_\sigma \left[ \sup_w \sum_{i=1}^n w\sigma((u')^\top x_i)\sigma_i \right] \qquad (\text{ as } u = u')$$

$$= \frac{1}{n}\mathbb{E}_\sigma \left[ \sup_w w \left( \sum_{i=1}^n \sigma((u')^\top x_i)\sigma_i \right) \right].$$

Since, $|w - w'| < \tau$,

$$w' - \tau < w < w' + \tau$$

Then, we can compute the supremum over $w$ as

$$w = \begin{cases} w' - \tau & \text{if } \left( \sum_{i=1}^n \sigma((u')^\top x_i)\sigma_i \right) < 0 \\ w' + \tau & \text{if } \left( \sum_{i=1}^n \sigma((u')^\top x_i)\sigma_i \right) \geq 0 \end{cases}$$

Therefore, we have

$$\sup_w w \left( \sum_{i=1}^n \sigma((u')^\top x_i)\sigma_i \right) = \left( w' + \tau \operatorname{sign} \left( \sum_{i=1}^n \sigma((u')^\top x_i)\sigma_i \right) \right) \cdot \left( \sum_{i=1}^n \sigma((u')^\top x_i)\sigma_i \right).$$

Now, we can calculate the Rademacher complexity as

$$R_S(\mathcal{H}_{\phi,\tau}) = \frac{1}{n}\mathbb{E}_\sigma \left[ w' \left( \sum_{i=1}^n \sigma((u')^\top x_i)\sigma_i \right) + \tau | \sum_{i=1}^n \sigma((u')^\top x_i)\sigma_i | \right]$$

$$= \frac{\tau}{n}\mathbb{E}_\sigma \left[ | \sum_{i=1}^n \sigma((u')^\top x_i)\sigma_i | \right]$$

$$\leq \frac{\tau}{n}\sqrt{\mathbb{E}_\sigma \left[ \| \sum_{i=1}^n \sigma((u')^\top x_i)\sigma_i \|^2 \right]} \qquad (\text{Jensen's inequality})$$

$$= \frac{\tau}{n}\sqrt{\mathbb{E}_\sigma \left[ \sum_{i=1}^n \sigma((u')^\top x_i)^2\sigma_i^2 \right]} \qquad (\text{since } \sigma_i, \sigma_j \text{ are independent with mean } 0)$$

$$\leq \frac{\tau}{n}\sqrt{\sum_{i=1}^n ((u')^\top x_i)^2}$$

$$\leq \frac{\tau}{n}\sqrt{\sum_{i=1}^n \|x_i\|^2}.$$

Combining this with the fact that $\mathbb{E}\left[\|x\|^2\right] \leq C^2$, we have

$$R_n(\mathcal{H}_{\phi,\tau}) = \mathbb{E}[R_S(\mathcal{H}_{\phi,\tau})]$$

$$\leq \frac{\tau}{n}\mathbb{E}[\sqrt{\sum_{i=1}^{n}\|x_i\|^2}]$$

$$\leq \frac{\tau}{n}\sqrt{\mathbb{E}[\sum_{i=1}^{n}\|x_i\|^2]} \quad \text{(Jensen's inequality)}$$

$$\leq \frac{\tau C}{\sqrt{n}}.$$

**Case 2:** $|w'|\|u'\| < \tau$.
We know that $\mathcal{H}_{\phi,\tau} = \mathcal{H}_{\phi,\tau}^{(1)} \bigcup \mathcal{H}_{\phi,\tau}^{(2)}$ when

$$\mathcal{H}_{\phi,\tau}^{(1)} = \{h \in \mathcal{H} | h : x \to w\sigma(u^\top x), u = u', |w - w'| < \tau\}$$

$$\mathcal{H}_{\phi,\tau}^{(2)} = \{h \in \mathcal{H} | h : x \to w\sigma(u^\top x), \|u\| = 1, u \neq u', |w| < \tau\}$$

We have

$$R_S(\mathcal{H}_{\phi,\tau}) = \frac{1}{n}\mathbb{E}_\sigma\left[\sup_{h \in \mathcal{H}_{\phi,\tau}} \sum_{i=1}^{n} h(x_i)\sigma_i\right]$$

$$\leq \frac{1}{n}\mathbb{E}_\sigma\left[\sup_{h \in \mathcal{H}_{\phi,\tau}^{(1)}} \sum_{i=1}^{n} h(x_i)\sigma_i + \sup_{h \in \mathcal{H}_{\phi,\tau}^{(2)}} \sum_{i=1}^{n} h(x_i)\sigma_i\right]$$

$$= R_S(\mathcal{H}_{\phi,\tau}^{(1)}) + R_S(\mathcal{H}_{\phi,\tau}^{(2)})$$

The second line holds as $\sup_{x \in A \cup B} f(x) \leq \sup_{x \in A} f(x) + \sup_{x \in B} f(x)$ when $\sup_{x \in A} f(x) \geq 0$ and $\sup_{x \in B} f(x) \geq 0$. We know that both of these supremums be greater than zero, as we can recover the value of 0 with $w = 0$. From **Case 1**, we know that

$$R_n(\mathcal{H}_{\phi,\tau}^{(1)}) \leq \frac{\tau C}{\sqrt{n}}.$$

We also note that $\mathcal{H}_{\phi,\tau}^{(2)}$ is a class of two layer neural networks with weights with norms bounded by $\tau$. From Theorem I.2, we have that

$$R_n(\mathcal{H}_{\phi,\tau}^{(2)}) \leq \frac{2\tau C}{\sqrt{n}}.$$

Therefore, in **Case 2**,

$$R_n(\mathcal{H}_{\phi,\tau}) \leq \frac{3\tau C}{\sqrt{n}}.$$

as required. $\qquad\square$

Now, we consider in the general setting (i.e., no restriction on $m$).

**Theorem I.4** (Rademacher complexity for two layer neural networks with gradient constraints ).
*Let $\mathcal{X}$ be an instance space and $\mathcal{D}_\mathcal{X}$ be a distribution over $\mathcal{X}$ with a large enough support. Let $\mathcal{H} = \{h : x \mapsto \sum_{j=1}^{m} w_j\sigma(u_j^\top x) | w_j \in \mathbb{R}, u_j \in \mathbb{R}^d, \|u_j\|_2 = 1, \sum_{j=1}^{m} |w_j| \leq B\}$. Assume that there exists some constant $C > 0$ such that $\mathbb{E}_{x \sim \mathcal{D}_\mathcal{X}}[\|x\|_2^2] \leq C^2$. Our explanation constraint is given by a constraint on the gradient of our models, where we want the gradient of our learnt model to be close to a particular target function $h' \in \mathcal{H}$. Let this be represented by an explanation loss given by*

$$\phi(h, x) = \|\nabla_x h(x) - \nabla_x h'(x)\|_2 + \infty \cdot 1\{\|\nabla_x h(x) - \nabla_x h'(x)\| > \tau\}$$

*for some $\tau > 0$. Then, we have that*

$$R_n(\mathcal{H}_{\phi,\tau}) \leq \frac{3\tau mC}{\sqrt{n}}.$$

*To be precise,*

$$R_n(\mathcal{H}_{\phi,\tau}) \leq \frac{(2m+q)\tau C}{\sqrt{n}}.$$

*when $q$ is the number of node $j$ of $h'$ such that $|w'_j| < \tau$.*

We note that this result indeed depends on the number of hidden dimensions $m$; however, we note that in the general case (Theorem I.2), the value of $B$ is $O(m)$ as it is a sum over the values of each hidden node. We now present the proof for the more general version of our theorem.

*Proof.* For simplicity, we first assume that any $h \in \mathcal{H}$ has that $\|u_j\| = 1, \forall j$. Consider $h \in \mathcal{H}$, we write $h = \sum_{j=1}^{m} w'_j \sigma((u'_j)^\top x)$ and let $h'(x) = \sum_{j=1}^{m} w'_j \sigma((u'_j)^\top x)$ be a function for our gradient constraint. The gradient of a hypothesis $h$ is given by

$$\nabla_x h(x) = \sum_{j=1}^{m} w_j u_j \cdot 1\{u_j^\top x > 0\},$$

which is a piecewise constant function over at most $2^m$ regions. Then, we consider that

$$\nabla_x h(x) - \nabla_x h'(x) = \sum_{j=1}^{m} w_j u_j \cdot 1\{u_j^\top x > 0\} - \sum_{j=1}^{m} w'_j u'_j \cdot 1\{(u'_j)^\top x > 0\},$$

which is a piecewise constant function over at most $2^{2m}$ regions. We again make an assumption that each of these regions has a non-zero probability mass. Our choice of $\phi(h, x)$ guarantees that the norm of the gradient in each region is less than $\tau$. Similar to the case with $m = 1$, we will show that the gradient constraint leads to a class of functions with the same decision boundary or neural networks that have weights with a small norm.

Assume that among $u_1, ..., u_m$ there are $k$ vectors that have the same direction as $u'_1, ..., u'_m$. Without loss of generality, let $u_j = u'_j$ for $j = 1, ..., k$. In this case, we have that $\nabla_x h(x) - \nabla_x h'(x)$ is a piecewise function over $2^{2m-k}$ regions. As each region has non-zero probability mass, for each $j \in \{1, ..., k\}$, we know that $\exists x$ such that

$$u_j^\top x = (u'_j)^\top x > 0, \qquad u_i^\top x < 0 \text{ for } i \neq j, \qquad (u'_i)^\top x < 0 \text{ for } i \neq j.$$

In other words, we can observe a data point from each region that uniquely defines the value of a particular $w_j, u_j$. In this case, we have that

$$\nabla_x h(x) - \nabla_x h'(x) = w_j u_j - w'_j u'_j$$
$$= (w_j - w'_j) u'_j.$$

From our gradient constraint, we know that $\|\nabla_x h(x) - \nabla_x h'(x)\| \leq \tau, \forall x$, which implies that $|w_j - w'_j| \leq \tau$ for $j = 1, ..., k$.

On the other hand, for the remaining $j = k+1, ..., m$, we know that there exists $x$ such that

$$u_j^\top x > 0, \qquad u_i^\top x < 0 \text{ for } i \neq j, \qquad (u'_i)^\top x < 0 \text{ for } i = 1, ..., m.$$

Then, we have that $\nabla_x h(x) = w_j u_j$, and our constraint implies that $|w_j|\|u_j\| = |w_j| \leq \tau$. Similarly, we have that $|w'_j|\|u'_j\| = |w'_j| < \tau$, for $j = k+1, ..., m$. We can conclude that $\mathcal{H}_{\phi,\tau}$ is a class of two layer neural networks with $m$ hidden nodes (assuming $\|u_i\| = 1$) that for each node $w_j \sigma(u_j^\top x)$ satisfies

    1. There exists $l \in [m]$ that $u_j = u'_l$ and $|w_j - w'_l| < \tau$.

2. $|w_j| < \tau$

We further note that for a node $w_l'\sigma((u_l')^\top x)$ in $h'(x)$ that has that a high weight $|w_l'| > \tau$, there must be a node $w_j\sigma(u_j^\top x)$ in $h$ with the same boundary $u_j = u_l$. Otherwise, there is a contradiction with $|w_l'| < \tau$ for all nodes in $h'$ without a node in $h$ with the same boundary. We can utilize this characterization of the restricted class $\mathcal{H}_{\phi,\tau}$ to bound the Rademacher complexity of the class. Let

$$\mathcal{H}' = \{h : x \mapsto \sum_{j=1}^{m} w_j'\sigma((u_j')^\top x)a_j \mid a_j \in \{0,1\} \text{ and for } j \text{ that } |w_j'| > \tau, a_j = 1\}.$$

This is a class of two layer neural networks with at most $m$ nodes such that each node is from $h'$. We also have a condition that if the weight of the $j$-th node in $h'$ is greater than $\tau$, the $j$-th node must be present in any member of this class. Let

$$\mathcal{H}^{(\tau)} = \{h : x \mapsto \sum_{j=1}^{m} w_j\sigma((u_j)^\top x)a_j \mid w_j \in \mathbb{R}, u_j \in \mathbb{R}^d, |w_j| < \tau, \|u_j\| = 1\}.$$

be a class of two layer neural networks with $m$ nodes such that the weight of each node is at most $\tau$. We claim that for any $h \in \mathcal{H}_{\phi,\tau}$ there exists $h_1 \in \mathcal{H}', h_2 \in \mathcal{H}^{(\tau)}$ that $h = h_1 + h_2$. For any $h \in \mathcal{H}_{\phi,\tau}$, let $p_h : [m] \to [m] \cup \{0\}$ be a function that match a node in $h$ with the node with the same boundary in $h'$. Formally,

$$p_h(j) = \begin{cases} l & \text{when } u_j = u_l' \\ 0 & \text{otherwise.} \end{cases}$$

The function $p_h$ maps $j$ to 0 if there is no node in $h'$ with the same boundary. Let $w_0' = 0, u_0' = [0, \ldots, 0]$, we can write

$$
\begin{aligned}
h(x) &= \sum_{j=1}^{m} w_j\sigma(u_j^\top x) \\
&= \sum_{j=1}^{m} w_j\sigma(u_j^\top x) - w_{p_h(j)}'\sigma((u')_{p_h(j)}^\top x) + w_{p_h(j)}'\sigma((u')_{p_h(j)}^\top x) \\
&= \underbrace{\sum_{p_h(j)\neq 0} (w_j - w_{p_h(j)}')\sigma((u')_{p_h(j)}^\top x) + \sum_{p_h(j)=0} w_j\sigma(u_j^\top x)}_{\in \mathcal{H}^{(\tau)}} + \underbrace{\sum_{p(j)\neq 0} w_{p_h(j)}'\sigma((u')_{p_h(j)}^\top x)}_{\in \mathcal{H}'}.
\end{aligned}
$$

The first term is a member of $\mathcal{H}^{(\tau)}$ because we know that $|w_j - w_{p(j)}'| < \tau$ or $|w_j| < \tau$. The second term is also a member of $\mathcal{H}'$ since for any $l$ that $|w_l'| > \tau$, there exists $j$ that $p_h(j) = l$. Therefore, we can write $h$ in terms of a sum between a member of $\mathcal{H}'$ and $\mathcal{H}^{(\tau)}$. This implies that

$$R_n(\mathcal{H}_{\phi,\tau}) \le R_n(\mathcal{H}') + R_n(\mathcal{H}^{(\tau)}).$$

From Theorem I.2, we have that

$$R_n(\mathcal{H}_{\phi,\tau}^{(\tau)}) \le \frac{2\tau m C}{\sqrt{n}}.$$

Now, we will calculate the Rademacher complexity of $\mathcal{H}'$. For $S = \{x_1, \ldots, x_n\}$,

$$
\begin{aligned}
R_S(\mathcal{H}') &= \frac{1}{n}\mathbb{E}_\sigma\left[\sup_{h \in \mathcal{H}'} \sum_{i=1}^n h(x_i)\sigma_i\right] \\
&= \frac{1}{n}\mathbb{E}_\sigma\left[\sup_{h \in \mathcal{H}'} \sum_{i=1}^n (\sum_{j=1}^m w'_j\sigma((u'_j)^\top x_i)a_j)\sigma_i\right] \\
&= \frac{1}{n}\mathbb{E}_\sigma\left[\sup_{h \in \mathcal{H}'} \sum_{i=1}^n (\sum_{|w'_j|<\tau} w'_j\sigma((u'_j)^\top x_i)a_j + \sum_{|w'_j|>\tau} w'_j\sigma((u'_j)^\top x_i))\sigma_i\right] \\
&= \frac{1}{n}\mathbb{E}_\sigma\left[\sup_{a_j \in \{0,1\}} \sum_{i=1}^n \sum_{|w'_j|<\tau} w'_j\sigma((u'_j)^\top x_i)a_j\sigma_i\right] \\
&= \frac{1}{n}\mathbb{E}_\sigma\left[\sup_{a_j \in \{0,1\}} \sum_{|w'_j|<\tau} a_j(w'_j\sum_{i=1}^n \sigma((u'_j)^\top x_i)\sigma_i)\right].
\end{aligned}
$$

To achieve the supremum, if $w'_j\sum_{i=1}^n \sigma((u'_j)^\top x_i)\sigma_i > 0$ we need to set $a_j = 1$, otherwise, we need to set $a_j = 0$. Therefore,

$$
\begin{aligned}
R_S(\mathcal{H}') &= \frac{1}{n}\mathbb{E}_\sigma\left[\sup_{a_j \in \{0,1\}} \sum_{|w'_j|<\tau} a_j(w'_j\sum_{i=1}^n \sigma((u'_j)^\top x_i)\sigma_i)\right] \\
&= \frac{1}{n}\mathbb{E}_\sigma\left[\sum_{|w'_j|<\tau} \sigma(w'_j\sum_{i=1}^n \sigma((u'_j)^\top x_i)\sigma_i)\right] \\
&= \frac{1}{2n}\mathbb{E}_\sigma\left[\sum_{|w'_j|<\tau} (w'_j\sum_{i=1}^n \sigma((u'_j)^\top x_i)\sigma_i) + |w'_j\sum_{i=1}^n \sigma((u'_j)^\top x_i)\sigma_i|\right] \qquad (\sigma(x) = \frac{x+|x|}{2}) \\
&= \frac{1}{2n}\mathbb{E}_\sigma\left[\sum_{|w'_j|<\tau} |w'_j\sum_{i=1}^n \sigma((u'_j)^\top x_i)\sigma_i|\right] \\
&\le \frac{1}{2n}\left(\sum_{|w'_j|<\tau} |w'_j|\right)\mathbb{E}_\sigma\left[\sup_{\|u\|=1} |\sum_{i=1}^n \sigma(u^\top x_i)\sigma_i|\right] \\
&\le \frac{1}{n}\left(\sum_{|w'_j|<\tau} |w'_j|\right)\mathbb{E}_\sigma\left[\sup_{\|u\|=1} \sum_{i=1}^n \sigma(u^\top x_i)\sigma_i\right] \qquad\qquad (\text{Lemma } I.1) \\
&\le \left(\sum_{|w'_j|<\tau} |w'_j|\right) \underbrace{\mathbb{E}_\sigma\left[\frac{1}{n}\sup_{\|u\|=1} \sum_{i=1}^n u^\top x_i\sigma_i\right]}_{\text{Empirical Rademacher complexity of a linear model}} \qquad (\text{Talagrand's Lemma}).
\end{aligned}
$$

From Theorem 4.1, we can conclude that

$$
R_n(\mathcal{H}') \le \sum_{|w'_j|<\tau} |w'_j|\frac{C}{\sqrt{n}} \le \frac{q\tau C}{\sqrt{n}} \le \frac{m\tau C}{\sqrt{n}}
$$

when $q$ is the number of nodes $j$ of $h'$ such that $|w'_j| < \tau$. Therefore,

$$R_n(\mathcal{H}') \leq \frac{(2m+q)\tau C}{\sqrt{n}} \leq \frac{3m\tau C}{\sqrt{n}}.$$

$\square$

A tighter bound is given by $\frac{(2m+q)\tau C}{\sqrt{n}}$ when $q$ is the number of $w'_j$ that $|w'_j| < \tau$. As $\tau \to 0$, we also have $q \to 0$. This implies that we have an upper bound of $\frac{2m\tau C}{\sqrt{n}}$ if $\tau$ is small enough. When comparing this to the original bound $\frac{2BC}{\sqrt{n}}$, we can do much better if $\tau \ll \frac{B}{m}$. We would like to point out that our bound does not depend on the distribution $\mathcal{D}$ because we choose a strong explanation loss

$$\phi(h,x) = \|\nabla_x h(x) - \nabla_x h'(x)\|_2 + \infty \cdot 1\{\|\nabla_x h(x) - \nabla_x h'(x)\| > \tau\}$$

which guarantees that $\|\nabla_x h(x) - \nabla_x h'(x)\|_2 \leq \tau$ almost everywhere. We also assume that we are in a high-dimensional setting $d \gg m$ and there exists $x$ with a positive probability density at any partition created by $\nabla_x h(x)$.

## J  Algorithmic Results for Two Layer Neural Networks with a Gradient Constraint

Now that we have provided generalization bounds for the restricted class of two layer neural networks, we also present an algorithm that can identify the parameters of a two layer neural network (up to a permutation of the weights). In practice, we might solve this via our variational objective or other simpler regularized techniques. However, we also provide a theoretical result for the required amount of data (given some assumptions about the data distribution) and runtime for an algorithm to exactly recover the parameters of these networks under gradient constraints.

We again know that the gradient of two layer neural networks with ReLU activations can be written as

$$\nabla_x f_{w,U}(x) = \sum_{i=1}^m w_i u_i \cdot 1\{u_i^T x > 0\},$$

where we consider $||u_i|| = 1$. Therefore, an exact gradient constraint given of the form of pairs $(x, \nabla_x f(x))$ produces a system of equations.

**Proposition J.1.** *If the values of $u_i$'s are known, we can identify the parameters $w_i$ with exactly $m$ fixed samples.*

*Proof.* We can select $m$ datapoints, which each achieve value 1 for the indicator value in the gradient of the two layer neural network. This would give us $m$ equations, which each are of the form

$$\nabla_x f_{w,U}(x_i) = w_i u_i.$$

Therefore, we can easily solve for the values of $w_i$, given that $u_i$ is known. $\square$

To make this more general, we now consider the case where $u_i$'s are not known but are at least linearly independent.

**Proposition J.2.** *Let the $u_i$'s be linearly independent. Assume that each region of the data (when partitioned by the values of $u_i$) has non-trivial support $> p$. Then, with probability $1 - \delta$, we can identify the parameters $w_i, u_i$ with $O\left(2^m + \frac{m+\log(\frac{1}{\delta})}{\log(\frac{1}{1-p})}\right)$ data points and in $O(2^{2m})$ time.*

*Proof.* Let us partition $\mathcal{X}$ into regions satisfying unique values of the binary vector $(1\{u_1^T x > 0\}, ..., 1\{u_m^T x > 0\})$, which by our assumption each have at least some probability mass $p$. First, we calculate the probability that we observe one data point with an explanation from each region in this partition. This is equivalent to sampling from a multinomial distribution with probabilities $(p_1, ..., p_{2^m})$, where $p_i \geq p, \forall i$. Then,

$$\Pr(\text{observe all regions in } n \text{ draws}) = 1 - \Pr(\exists i \text{ s.t. we do not observe region } i)$$
$$= 1 - 2^m(1-p)^n.$$

**Algorithm 1** Algorithm for identifying parameters of a two layer neural network, given exact gradient constraints

---

1: **Input:** We are given $M = \{\sum_{x \in C} x | C \in \mathcal{P}(\{x_1, ..., x_m\})\}$, with $\{x_1, ..., x_m\}$ linearly independent
2: **Output:** The set of basis elements $\{x_1, ..., x_m\}$
3: **function**
4:    $B = \{\}, S = \{\}$        {Set for basis vectors and set for a current sum of at least 2 elements}
5:    **for** $x \in M$ **do**
6:       **if** $x \in S$ **then**
7:          pass
8:       **else**
9:          $B = B \cup \{x\}$
10:          **if** $|B| = 2$ **then**
11:             $S = \{y_1 + y_2\}$, where $B = \{y_1, y_2\}$
12:          **else**
13:             $S = S \cup \{y + x | y \in S\}$                          {Updating sums from adding $x$}
14:          **end if**
15:          $O = B \cap S$                  {Computing overlap between current basis and sums}
16:          $B = B \setminus O$                  {Removing elements contained in pairwise span}
17:          $S = \{y - y_o | y \in S, y_o \in O\}$                  {Updating sums $S$ from removing set $O$}
18:       **end if**
19:    **end for**
20:    **return** $B$
21: **end function**

---

Setting this as no less than $1 - \delta$ leads to that $n \geq \frac{m + \log(\frac{1}{\delta})}{\log(\frac{1}{1-p})}$.

Given $O(2^m + \frac{m + \log(\frac{1}{\delta})}{\log(\frac{1}{1-p})})$ pairs of data and gradients, we will observe at least one pair from each region of the partition. Then, identifying the values of $u_i$'s and $w_i$'s is equivalent to identifying the datapoints that correspond to a value of the binary vector where only one indicator value is 1. These values can be identified in $O(2^{3m})$ time; the algorithm is given in Algorithm J.1. These results demonstrate that we can indeed learn the parameters (up to a permutation) of a two layer neural network given exact gradient information. $\qquad\square$

## J.1   Algorithm for Identifying Regions

We first note that identifying the parameters $u_i$'s and $w_i$'s of a two layer neural network is equivalent to identifying the values $\{x_1, ..., x_m\}$ from the set $\{\sum_{x \in C} x | C \in \mathcal{P}(\{x_1, ..., x_m\})\}$, where $\mathcal{P}$ denotes the power set. We also assume that $x_1, ..., x_m$ are linearly independent, so we cannot create $x_i$ from any linear combination of $x_j$'s with $j \neq i$. Then, we can identify the set $\{x_1, ..., x_m\}$ as in Algorithm 1. This algorithm runs in $O(2^{3m})$ time as it iterates through each point in $M$ and computes the overlapping set $O$ and resulting updated sum $S$, which takes $O(2^{2m})$ time. From the resulting set $B$, we can exactly compute values $u_i$ and $w_i$ up to a permutation.

## K   Additional Synthetic Experiments

We now present additional synthetic experiments that demonstrate the performance of our approach under settings with imperfect explanations and compare the benefits of using *different types* of explanations.

### K.1   Variational Objective is Better with Noisy Gradient Explanations

Here, we present the remainder of the results from the synthetic regression task of **??**, under more settings of noise $\epsilon$ added to the gradient explanation.

Again, we observe that our method does better than that of the Lagrangian approach and the self-training method. Under high levels of noise, the Lagrangian method does poorly. On the contrary,

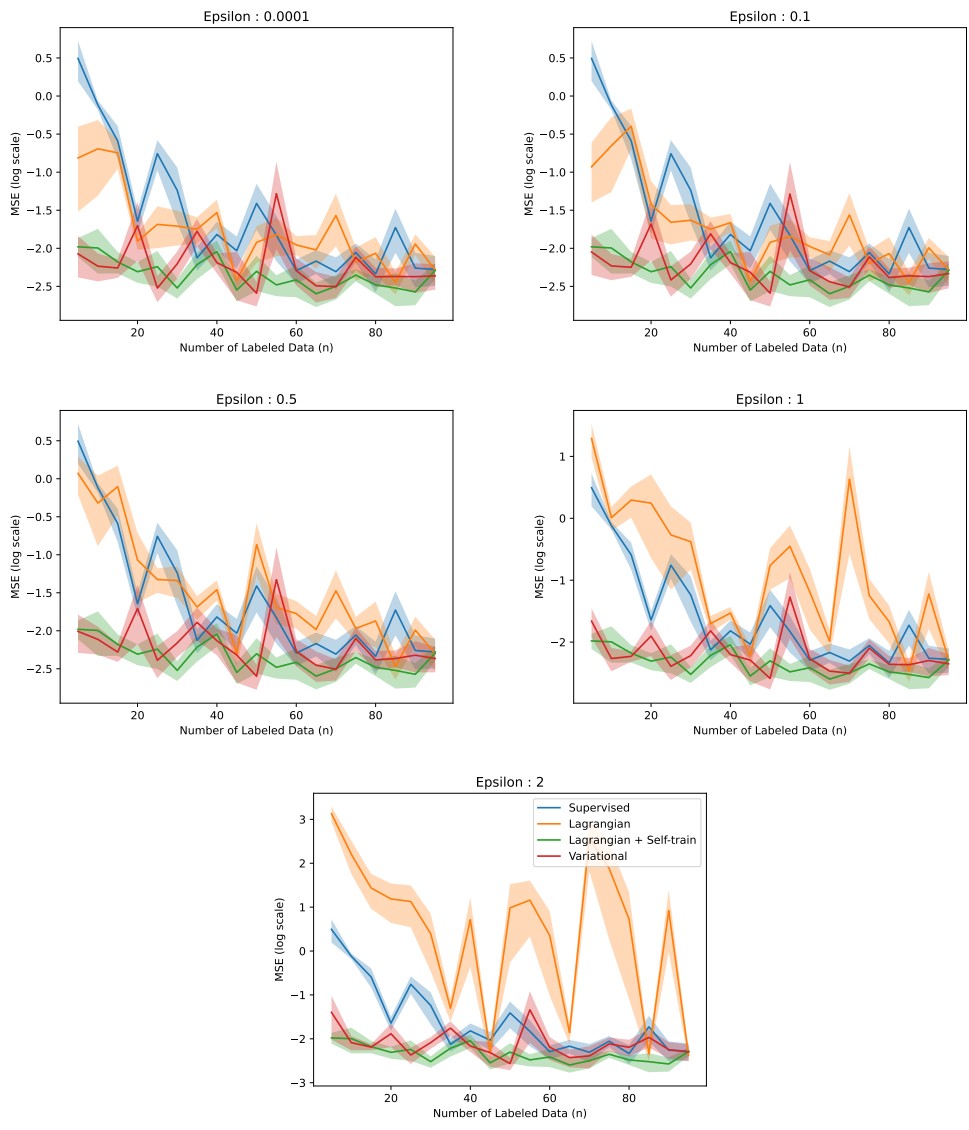

Figure 10: Comparison of MSE on regressing a two layer neural network with explanations of noisy gradients. $m = 1000, k = 20, \lambda = 10$. For the iterative methods, $T = 10$. Results are averaged over 5 seeds.

our method is resistant to this noise and also outperforms self-training significantly in settings with limited labeled data.

### K.2 Comparing Different Types of Explanations

Here, we present synthetic results to compare using different types of explanation constraints. We focus on comparing noisy gradients as before, as well as noisy classifiers, which are used in the setting of weak supervision [30]. Here, we generate our noisy classifiers as $h^*(x) + \epsilon$, where $\epsilon \sim \mathcal{N}(0, \sigma^2)$. We omit the results of self-training as it does not use any explanations, and we keep the supervised method as a baseline. Here, $t = 0.25$.

We observe different trends in performance as we vary the amount of noise in the noisy gradient or noisy classifier explanations. With any amount of noise and sufficient regularization ($\lambda$), this influences the overall performance of the methods that incorporate constraints. With few labeled data, using noisy classifiers helps outperform standard supervised learning. With a larger amount

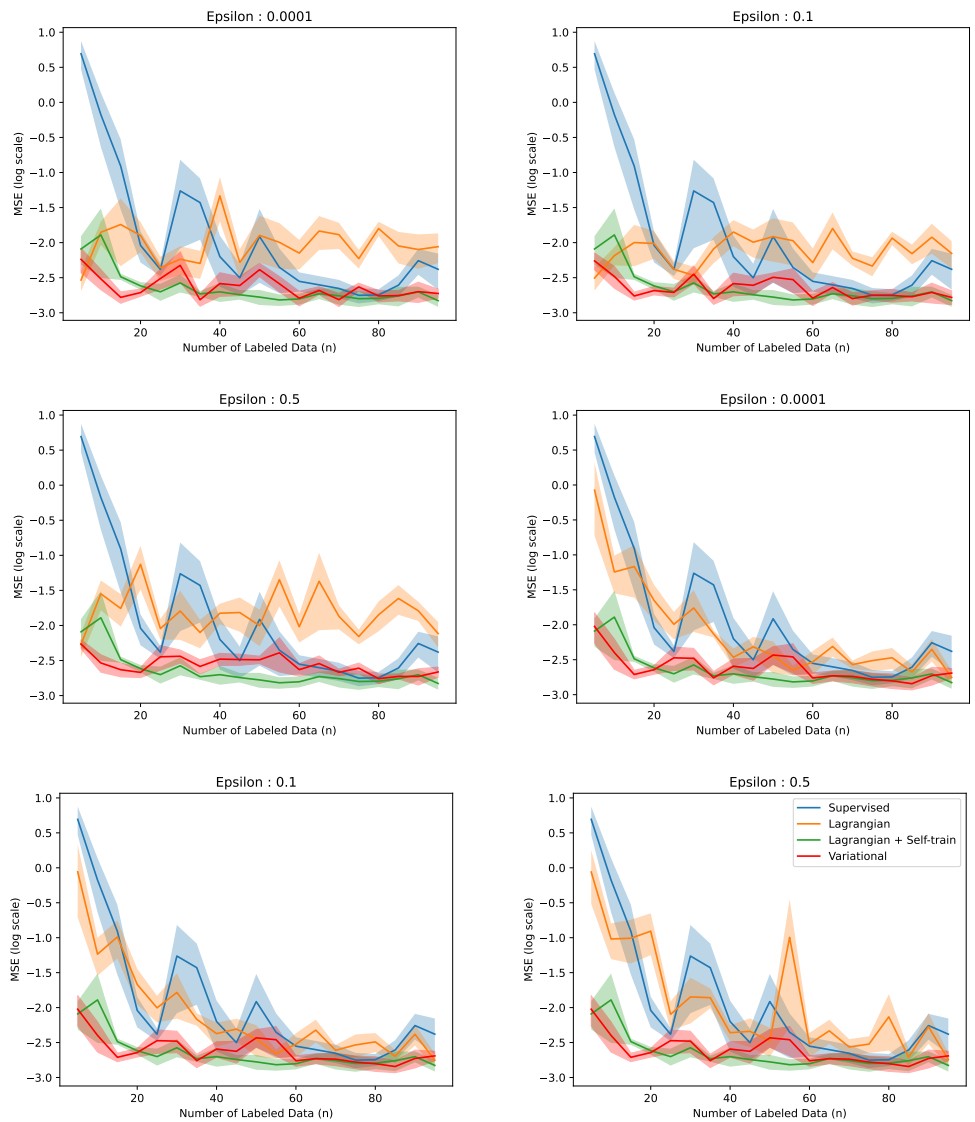

Figure 11: Comparison of MSE on regressing a two layer neural network with explanations as a noisy classifier (top) and noisy gradients (bottom). $m = 1000, k = 20$. For the iterative methods, $T = 10$. Results are averaged over 5 seeds. $\epsilon$ represents the variance of the noise added to the noisy classifier or noisy gradient.

of labeled data, this leads to no benefits (if not worse performance of the Lagrangian approach). However, with the noisy gradient, under small amounts of noise, the restricted class of hypothesis will still capture solutions with low error. Therefore, in this case, we observe that the Lagrangian approach outperforms standard supervised learning in the case with few labeled data and matches it with sufficient labeled data. Our method outperforms or matches both methods across all settings.

We consider another noisy setting, where noise has been added to the weights of a copy of the target two layer neural network. Here, we compare how this information impacts learning from the direct outputs (noisy classifier) or the gradients (noisy gradients) of that noisy copy (Figure 12).

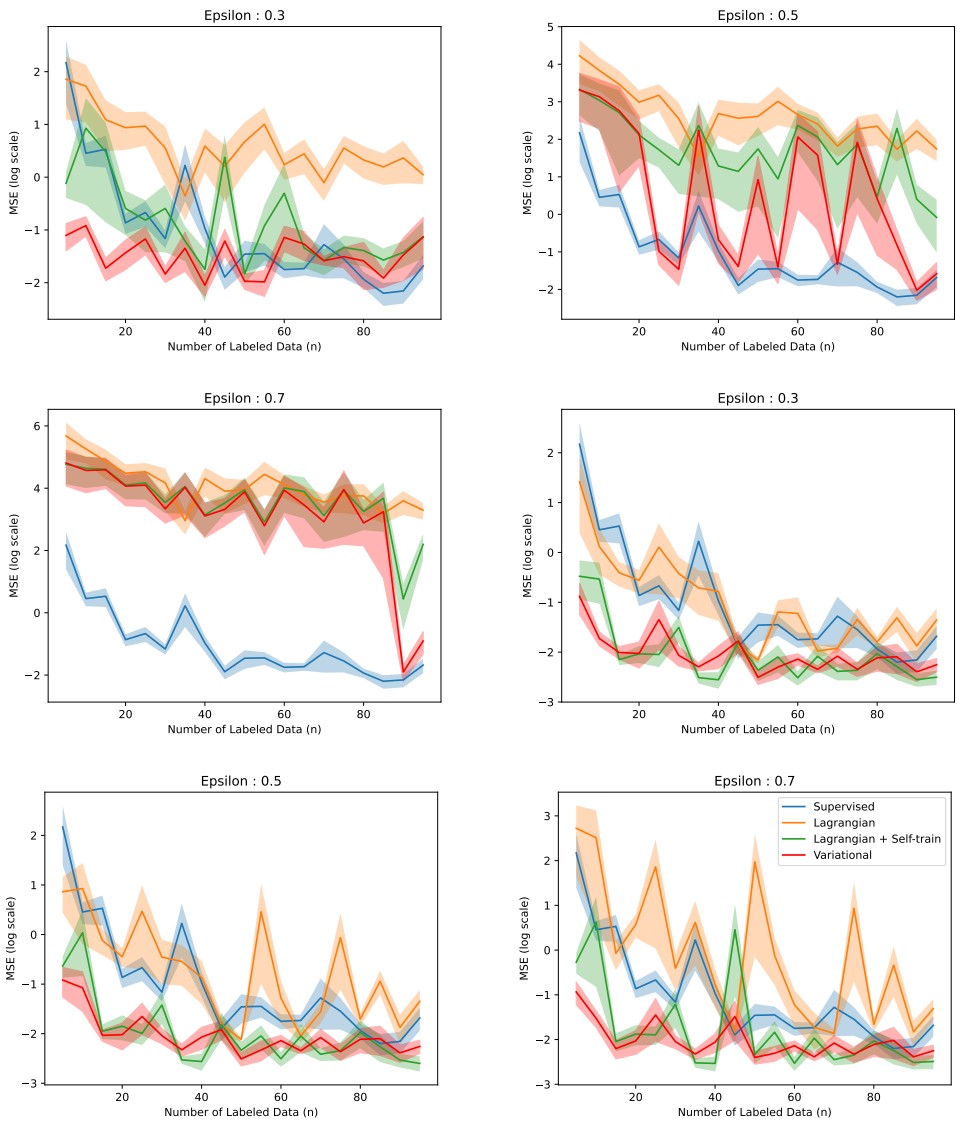

Figure 12: Comparison of MSE on regressing a two layer neural network with explanations as a noisy classifier (top) and noisy gradients (bottom). $m = 1000, k = 20$. For the iterative methods, $T = 10$. Results are averaged over 5 seeds. $\epsilon$ represents the variance of the noise added to the noisy classifier or noisy gradient.

## L  Additional Baselines

We compare against an additional baseline of a Lagrangian-regularized model + self-training on unlabeled data. We again note that this is not a standard method in practice and does not naturally fit into a theoretical framework, although we present it to compare against a method that uses both explanations and unlabeled data.

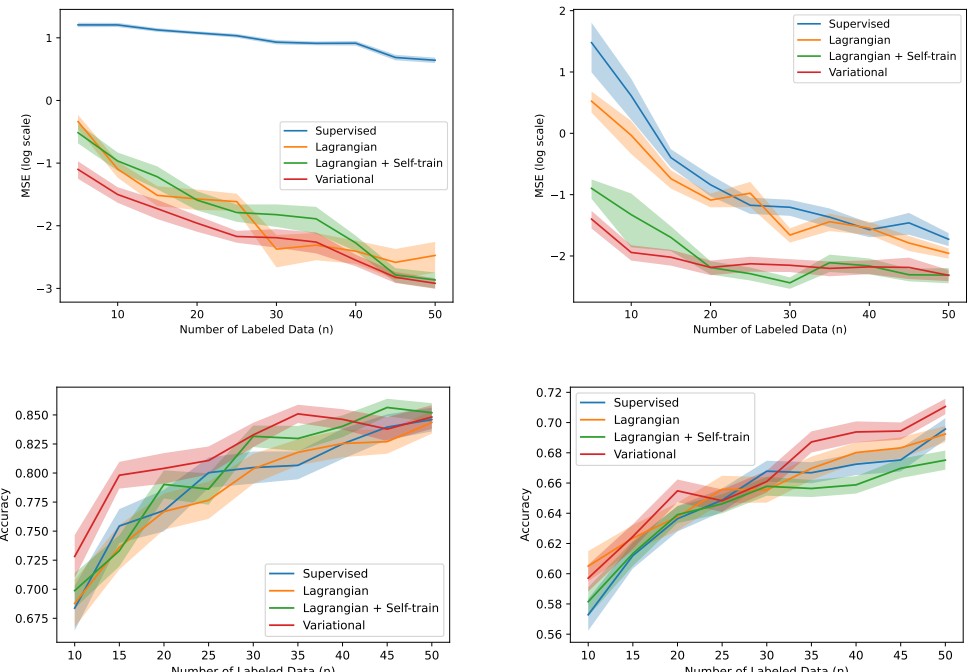

Figure 13: Comparison of MSE on regressing a linear model (top left) and two layer neural network (top right) with gradient explanations. $m = 1000, k = 20$. For the iterative methods, $T = 2$. Results are averaged over 5 seeds. Comparison of classification accuracy on the YouTube dataset (bottom left) and the Yelp dataset (bottom right). $m = 500, k = 150$. Results are averaged over 40 seeds.

We observe that our method outperforms this baseline (Figure 13), again especially in the settings with limited labeled data. We observe that although this new method indeed satisfies constraints, when performing only a single round of self-training, it no longer satisfies these constraints as much. Thus, this supports the use of our method to perform multiple rounds of projections onto a set of EPAC models.

## M  Experimental Details

For all of our synthetic and real-world experiments, we use values of $m = 1000, k = 20, T = 3, \tau = 0, \lambda = 1$, unless otherwise noted. For our synthetic experiments, we use $d = 100, \sigma^2 = 5$. Our two layer neural networks have hidden dimensions of size 10. They are trained with a learning rate of 0.01 for 50 epochs. We evaluate all networks on a (synthetic) test set of size 2000.

For our real-world data, our two layer neural networks have a hidden dimension of size 10 and are trained with a learning rate of 0.1 (YouTube) and 0.1 (Yelp) for 10 epochs. $\lambda = 0.01$ and gradient values computed by the smoothed approximation in [**?** ] has $c = 1$. Test splits are used as follows from the YouTube and Yelp datasets in the WRENCH benchmark [46].

We choose the initialization of our variational algorithm $h_0$ as the standard supervised model, trained using gradient descent.

# N Ablations

We also perform ablation studies in the same regression setting as Section 6. We vary parameters that determine either the experimental setting or hyperparameters of our algorithms.

## N.1 Number of Explanation-annotated Data

First, we vary the value of $k$ to illustrate the benefits of our method over the existing baselines.

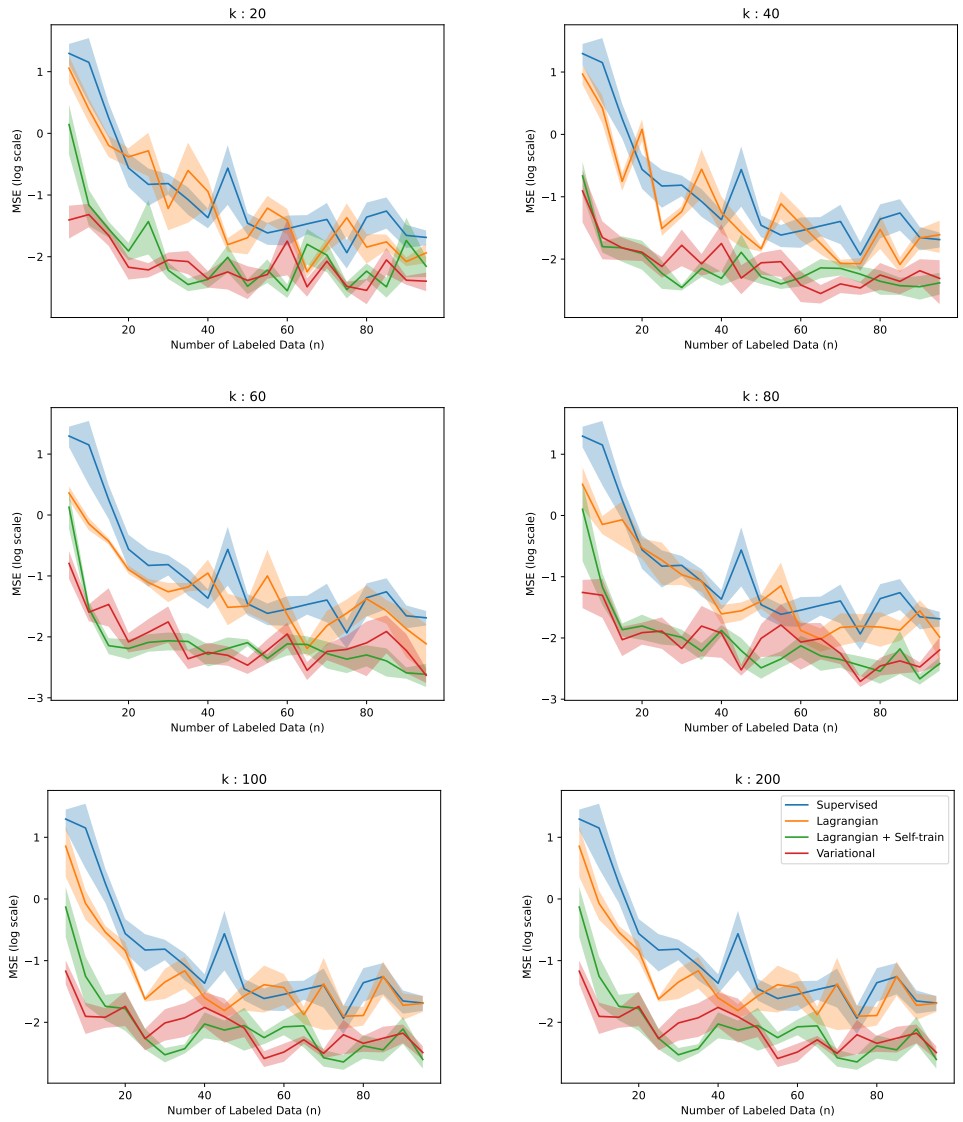

Figure 14: Comparison of MSE on regressing a two layer neural network over different amounts of explanation-annotated data $k$. $m = 1000$. For the iterative methods, $T = 10$. Results are averaged over 5 seeds.

We observe that our variational approach performs much better than a simple augmented Lagrangian method, which in turn does better than supervised learning with sufficiently large values of $k$. Our approach is always better than the standard supervised approach.

We also provide results for how well these methods satisfy these explanations over varying values of $k$.

## N.2 Simpler Teacher Models Can Maintain Good Performance

As noted before, we can use *simpler* teacher models to be regularized into the explanation-constrained subspace. This can lead to overall easier optimization problems, and we synthetically verify the impacts on the overall performance. In this experimental setup, we are regressing a two layer neural network with a hidden dimension size of 100, which is much larger than in our other synthetic experiments. Here, we vary over simpler teacher models by changing their hidden dimension size.

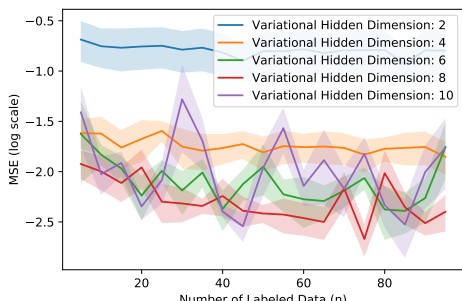

Figure 15: Comparison of MSE on regressing a two layer neural network over simpler teacher models (hidden dimension). Here, $k = 20, m = 1000, T = 10$. Results are averaged over 5 seeds.

We observe no major differences as we shrink the hidden dimension size by a small amount. For significantly smaller hidden dimensions (e.g., 2 or 4), we observe a large drop in performance as these simpler teachers can no longer fit the approximate projection onto our class of EPAC models accurately. However, slightly smaller networks (e.g., 6, 8) can fit this projection as well, if not better in some cases. This is a useful finding, meaning that our teacher can be a *smaller model* and get comparable results, showing that this simpler teacher can help with scalability without much or any drop in performance.

## N.3 Number of Unlabeled Data

As a main benefit of our approach is the ability to incorporate large amounts of unlabeled data, we provide a study as we vary the amount of unlabeled data $m$ that is available. When varying the amount of unlabeled data, we observe that the performance of self-training and our variational objective improves at similar rates.

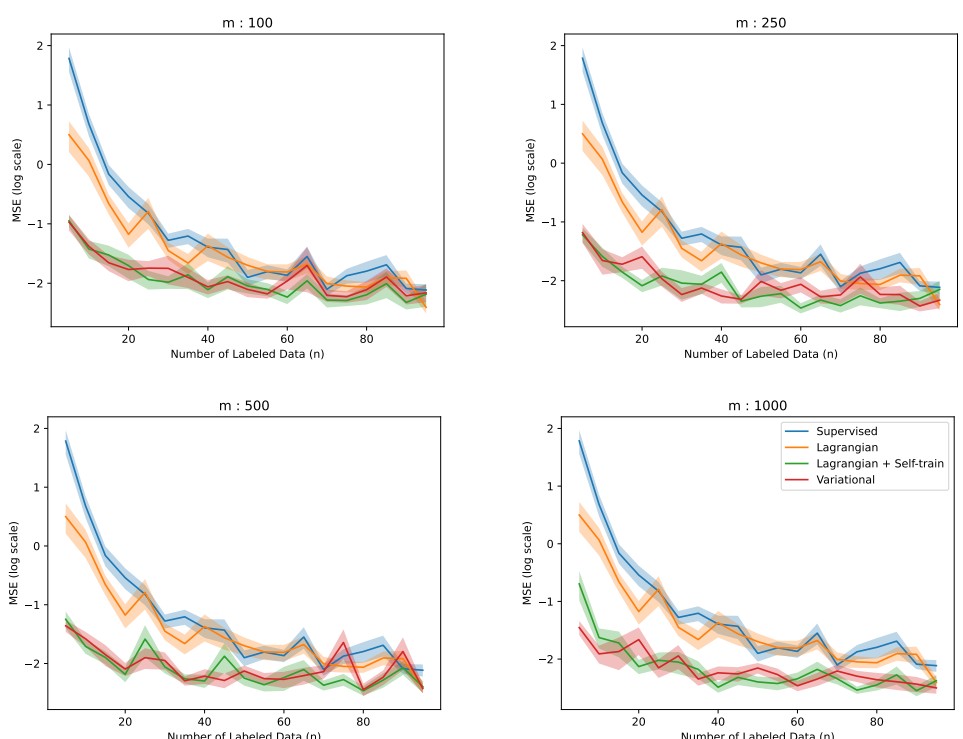

Figure 16: Comparison of MSE on regressing a two layer neural network over different values of $m$. $k = 20, T = 10$. Results are averaged over 5 seeds.

### N.4  Data Dimension

We also provide ablations as we vary the underlying data dimension $d$. As we increase the dimension $d$, we observe that the methods seem to achieve similar performance, due to the difficulty in modeling the high-dimensional data. Also, here gradient information is much harder to incorporate, as the input gradient itself is $d$-dimensional, so we do not see as much of a benefit of our approach as $d$ grows.

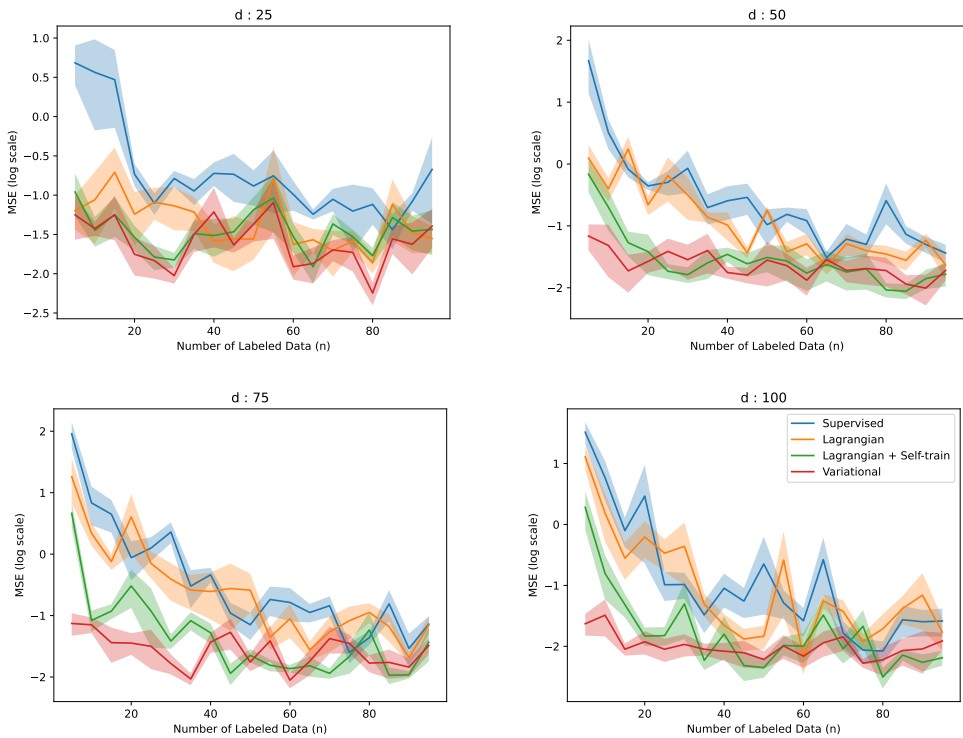

Figure 17: Comparison of MSE on regressing a two layer neural network over different underlying data dimensions $d$. $m = 1000, k = 20$. For the iterative methods, $T = 10$. Results are averaged over 5 seeds.

## N.5 Hyperparameters

First, we compare the different approaches over different values of regularization ($\lambda$) towards satisfying the explanation constraints. Here, we compare the augmented Lagrangian approach, the self-training approach, and our variational approach.

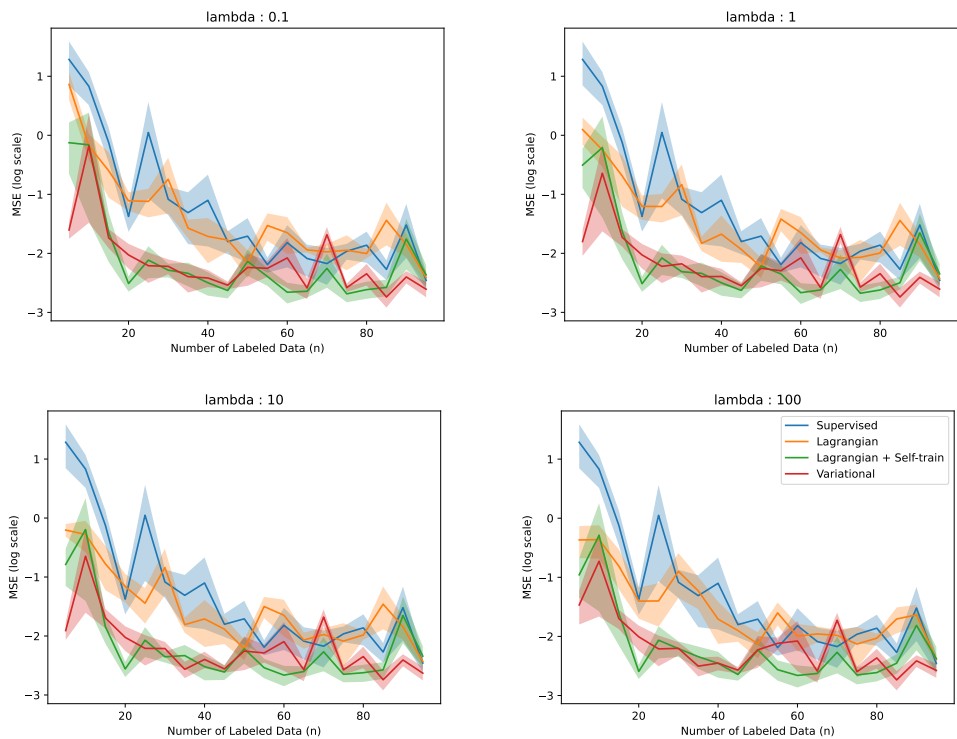

Figure 18: Comparison of MSE on regressing a two layer neural network over different values of $\lambda$. $m = 1000, k = 20$. For the iterative methods, $T = 10$. Results are averaged over 5 seeds.

We observe that there is not a significant trend as we change the value of $\lambda$ across the different methods. Since we know that our explanation is perfect (our restricted EPAC class contains the target classifier), increasing the value of $\lambda$ should help, until this constraint is met.

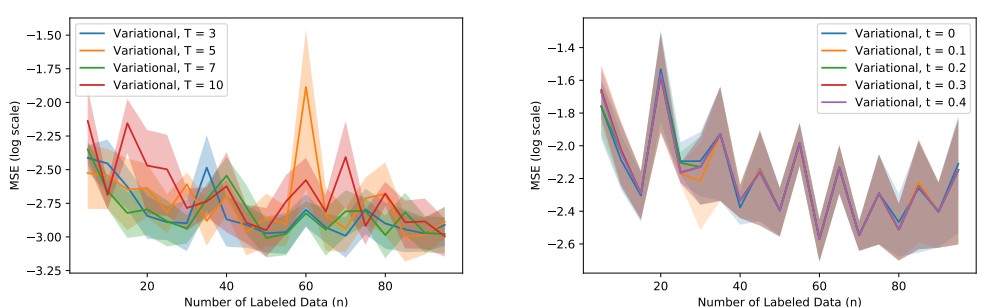

Figure 19: Comparison of MSE on regressing a two layer neural network over different values of $T$ (left) and $\tau$ (right) in our variational approach. $m = 1000, k = 20, \tau = 10, T = 10$, unless noted otherwise. Results are averaged over 5 seeds.

Next, we compare different hyperparameter settings for our variational approach. Here, we analyze trends as we vary the values of $T$ (number of iterations) and $\tau$ (threshold before adding hinge penalty).

We note that the value of $\tau$ does not significantly impact the performance of our method while increasing values of $T$ seems to generally benefit performance on this task.

## O  Social Impacts

While our proposed method has the potential to improve performance and efficiency in a variety of applications, our method could introduce new biases or reinforce existing biases in the data used to train the model. For example, if our explanations constraints are poorly specified and reflect biased behavior, this could lead to inaccurate or discriminatory predictions, which could have negative impacts on individuals or groups that are already marginalized. Therefore, it is important to note that these explanation constraints must be properly analyzed and specified to exhibit the desired behavior of our model.

