# OpenReview forum: "Learning with Explanation Constraints"
_NeurIPS.cc/2023/Conference — NeurIPS 2023 poster_

### Official Review · Reviewer_9kMr · 2023-07-05

**Soundness:** 3 good
**Presentation:** 3 good
**Contribution:** 4 excellent
**Rating:** 7
**Confidence:** 3

**Summary:**

This paper tries to answer an important and fundamental question: why can interpretability constraints on model parameters improve model performance. The authors developed an analytic framework. They first define constraints as a functional on a model function and a data point, and the model satisfying the constraint is hence represented as the functional value lies in a set. Following this setup, they defined the surrogate loss based on the constraint violation, and showed the generalization bound on this constraint violation loss. They showed that the constraints helped shrink the hypothesis class. They also showed that with gradient constraints, linear and 2-layer NN models could benefit in requiring fewer labeled data. They also discussed optimization algorithms through projections that could make the model satisfy the constraints.

**Strengths:**

This paper tries to answer an important and fundamental question: why can interpretability constraints on model parameters improve model performance. The presentation of this paper is clear, techniques sound, and contributions good.

**Weaknesses:**

The paper lacks a discussion on the stringency of their conditions in theorems, which makes it more difficult to judge the significance of the contribution.
The paper focus on gradient constraints and lacks a discussion and related work on model shape constraints, such as monotonicity and convexity.

**Questions:**

Please address the comments in the weakness section.

**Limitations:**

This paper doesn't have potential negative societal impact. The authors are encouraged to discuss the plausibility of the conditions of their theorems.

---

> ### Author Rebuttal · Authors · 2023-08-09
>
> > The paper lacks a discussion on the stringency of their conditions in theorems, which makes it more difficult to judge the significance of the contribution. The paper focus on gradient constraints and lacks a discussion and related work on model shape constraints, such as monotonicity and convexity.
>
> **Our response:** Our result presents the reduction in the Rademacher complexity of an EPAC model for a class of linear models and two-layer neural networks. (Theorem 3.1, 3.2). We have already provided a short discussion on the assumption of the gradient constraints for neural networks between lines 205-210. Our results apply to a general data distribution (see Theorem H.1 for linear models) and distributions where samples from this have a bounded norm in expectation (Theorem 3.2 for neural networks). This assumption on the bounded norm (in expectation) of data samples for NNs is a standard assumption in practice and is not restrictive of particular tasks; for example, a Gaussian distribution satisfies this assumption.
> We also make the assumption for our proof for neural networks that the support is large enough, or more specifically, that each partition of the input space (partitions being defined by values of the hidden layer of the neural network) has a positive probability mass. This is not a strong assumption, and it holds for any distribution where the support is the $R^d$, for example, Gaussian distributions. We will add these clarifications to our revision.
>
>
> We focus on gradient constraints since our work is motivated by a line of works in the field of explainability where such explanation constraints are used to train models [1,2], and input gradients are one of the canonical classes of explanations. We will discuss the other notions of constraints that you have mentioned in the camera-ready version.
>
> [1] Ross, Andrew Slavin, Michael C. Hughes, and Finale Doshi-Velez. "Right for the right reasons: training differentiable models by constraining their explanations." Proceedings of the 26th International Joint Conference on Artificial Intelligence. 2017.
>
> [2] Rieger, Laura, et al. "Interpretations are useful: penalizing explanations to align neural networks with prior knowledge." International conference on machine learning. PMLR, 2020.

---

> > ### Comment · Reviewer_9kMr · 2023-08-21
> >
> > Thank you for your response to my questions

---

### Official Review · Reviewer_w12b · 2023-07-05

**Soundness:** 2 fair
**Presentation:** 3 good
**Contribution:** 2 fair
**Rating:** 4
**Confidence:** 3

**Summary:**

Traditionally, learning from labelled examples has been well studied. However, with the advent of large models (and few-shot learning) and risk of misspecifications (spurious correlations), need for augmenting supervision such as with example explanations is gaining wider interest.
This paper deals with this timely topic of analysing when and how learning from explanations can improve sample efficiency.
Although learning from explanations is also promising in reducing systematic biases, this paper focusses on sample efficiency.
They also propose an interesting learning algorithm to accommodate explanation constraints.

I believe the topic and analysis is timely but I have a few concenrns regarding evaluation and analysis.

**Strengths:**

- Important practical problem.
- Well written and easy to follow.
- The proposed variational method is smart, which can exploit both labelled, unlabelled data along with explanations.

**Weaknesses:**

Disclaimer: I have not checked the proofs. But the statements seem correct.

Consider the following also as questions.

*How are explanations different?* Fundamentally, I did not understand why explanation constraints are given a status different from example labels. Can we not simply consider them to be additional labels and minimize the joint risk such as in the Lagrangian method or Rieger et.al. and handle the error bound using the classical analysis as shown between L161 and L162?
Especially since the explanations are imposed only approximately through objective functions.

*Clarification regarding Theorem 2.2.* From L 156: "... help with our learning by shrinking the hypothesis class $H$ to $H_{\phi,\tau+\epsilon_k}$". Although this sounds intuitively correct, from the definition of $H$, isn't $H_{\phi, \tau+\epsilon_k}$ larger than $H$?
Moreover, I do not see how constraints help improve the sample complexity since both $R_n(H_{\phi,\tau+\epsilon_k})$ and $err_D(h^*_{\tau-\epsilon_k})$ increase for a non-zero value of $\epsilon_k$?

*Motivation for variational method.* The need for variational method is motivated to avoid evaluating gradients of a potentially complicated surrogate loss: $\phi$, however even the variational method needs to evaluate the same when optimizing for $f$ in the first step.

*Evaluation concern*. Lagrangian method may have performed poorly because it is not exposed to unlabelled data unlike variational. What happens if we add an additional term to lagrangian objective on unlabelled data that is similar to self-training?
Otherwise, I do not see why lagrangian may poorly perform when compared with variational at least when we can compute gradient of gradients well (which is not much of a problem these days).

*Real world performance gains are unimpressive*. (Section 5.2 and Figure 5). I cannot clearly see the observation noted in L312-L313: "We observe that our variational objective achieves better performance than all other baseline approaches, across varying amounts of labeled data."
The weak labellers and the task should be better defined. How are the gradients (of $\phi$) computed (for say largrangian method) when weak learners are used for imposing constraints?

**Questions:**

Please see weaknesses above.

**Limitations:**

Addressed.

---

> ### Author Rebuttal · Authors · 2023-08-09
>
> > 1. How are explanations different? Fundamentally, I did not understand why explanation constraints are given a status different from example labels. Can we not simply consider them to be additional labels and minimize the joint risk such as in the Lagrangian method or Rieger et.al. and handle the error bound using the classical analysis as shown between L161 and L162? Especially since the explanations are imposed only approximately through objective functions.
>
> **Our response:** While you can do something like Rieger et.al., this does not fit into a standard learning theoretic framework, which analyzes when we only have class labeled examples. To theoretically analyze this setup, we need to use a learning theoretic framework that handles the classification loss and explanation loss separately. Explanation constraints here can be information that aren’t standard class labels for our data, e.g. ignoring the background. Since the explanations and labels are different, the classification loss and the explanation loss are different objects.
>
> >2. Clarification regarding Theorem 2.2. From L 156: "... help with our learning by shrinking the hypothesis class $H$ to $H_{\phi,\tau + \epsilon_k}$ ". Although this sounds intuitively correct, from the definition of $H$, isn't $H_{\phi,\tau + \epsilon_k}$ larger than $H$ ? Moreover, I do not see how constraints help improve the sample complexity since both $R_n(H_{\phi,\tau + \epsilon_k})$ and $err_D(h^*_{\tau - \varepsilon_k})$ increase for a non-zero value of  $\varepsilon_k$?
>
> **Our response:** No, $H_{\phi, \tau + \epsilon}$ is a subset of $H$ and therefore is smaller than that of $H$. We recall the definition of $H_{\phi, \tau + \epsilon_k}$ as any function in $H$ that the expected explanation loss is at most $\tau + \epsilon_k$. In fact, we can think of $H$ as $H_{\phi, \infty}$. Consequently, the complexity term $R_n(H_{\phi, \tau + \epsilon_k})$ is smaller than that of $R_n(H)$ - and we give bounds on how much smaller this is for 2 layer NNs and linear models in Thoerems 3.1 and 3.2.
>
> >3. Motivation for variational method. The need for variational method is motivated to avoid evaluating gradients of a potentially complicated surrogate loss: $\phi$, however even the variational method needs to evaluate the same when optimizing for $f$ in the first step.
>
> **Our response:** This is an astute observation; the underlying motivation for the variational method is to potentially achieve more efficient solutions to complicated constrained optimization problems. When compared to the standard Lagrangian penalized method, as is done in [1], the variational objective simplifies the Lagrangian objective even further by a sum of two terms, one of which is the empirical risk, and the other is a projection error onto the set of hypotheses that satisfy a bound on the explanation surrogate risk. A crucial advantage of such a decoupling is that we can reduce this to two steps (these are steps 1 and 2 in Section 4.1): a projection step of projecting current model onto the set of those with bounded explanation risk, and a synthesis step where we use the training labels as well as pseudo-labels from the projected model.
> The crucial advantage of this decoupling is that we can solve the projection step (this is step 1 in Section 4.1) with a **simpler** class of models for which the explanation surrogate gradients are more tractable, thus resolving a critical caveat with the Lagrangian objective. We could also use different learning rates and tolerances for the projection step, again with the aim of making the optimization problem more tractable.
> > 4. Evaluation concern. Lagrangian method may have performed poorly because it is not exposed to unlabelled data unlike variational. What happens if we add an additional term to lagrangian objective on unlabelled data that is similar to self-training? Otherwise, I do not see why lagrangian may poorly perform when compared with variational at least when we can compute gradient of gradients well (which is not much of a problem these days).
>
> **Our response:** We already have this comparison in the Appendix section L (additional baselines)! We also observe that a lagrangian with a single round of self-training afterwards is also worse in satisfying constraints on new test data, which in turn leads to worse performance in limited labeled data settings. This supports our method’s use of multiple rounds of incorporating these explanation constraints.
>
> >5. Real world performance gains are unimpressive. (Section 5.2 and Figure 5). I cannot clearly see the observation noted in L312-L313: "We observe that our variational objective achieves better performance than all other baseline approaches, across varying amounts of labeled data." The weak labellers and the task should be better defined. How are the gradients (of $\phi$) computed (for say largrangian method) when weak learners are used for imposing constraints?
>
> **Our response:** We will make this claim a bit less strong, but we still observe the following: “Our variational objective achieves better performance than all other baselines, on the majority of settings defined by the number of labeled data”
> The procedure for generating gradients is taken from a prior work [4]. The high level idea is that we can consider random inputs into these weak learners, and differentiate with respect to the parameters of this distribution to get a proxy for the gradient of these weak learners. In the case for weak learners that are hand-engineered lookup functions or regular expressions for language tasks, this corresponds to extracting feature indices for particular words or characters in the regular expression pattern.
>
> [4] D. Sam and J. Z. Kolter. Losses over labels: Weakly supervised learning via direct loss construction. https://arxiv.org/abs/2212.06921

---

> > ### Comment · Reviewer_w12b · 2023-08-14
> > **Further clarification**
> >
> > Thanks for the response, some of my concerns are now resolved.
> >
> > Points 2 and 4 are now resolved, thanks. I still have many other concerns.
> >
> > _Theoretical analysis_:
> > I am not yet convinced why explanation constraints are special. This issue was also raised by Reviewer 3fR6, but not yet resolved. Why not treat explanation constraints like any other constraints? It is not hard to see that constraints provide additional supervision, which would improve label efficiency. On the other extreme, supervision on explanations is not that different from label supervision. To elaborate on this, if the underlying hypothesis class is drawn from a GP prior, then observations on function gradient are readily accommodated just as observations on function values [1].
> >
> > More crisply, what is the significance of the theoretic results in relation to explanations beyond just saying that additional supervision through constraints improved label efficiency.
> >
> > _Experiments_
> >
> > Q:  Could you please clarify how $\lambda$ is obtained in the Augmented Lagrangian objective? Was $\lambda$ fixed? Was it obtained through alternate max-min steps typical of Lagrangian optimization?
> >
> > Q: Variational vs Lagrangian. The advantage of variational over Lagrangian is argued to be only computational. Figure 4 needs to be justified as to why Variational is satisfying explanation constraints better than Lagrangian.
> >
> > Few other issues:
> >
> > Datasets are non-standard. Rieger et.al. and Ross et.al. contain many other standard datasets like Decoy-MNIST or ISIC dataset. But the datasets used to report in this paper have not been used before for this problem to the best of my knowledge. The reason for choosing not to evaluate on standard datasets needs to be justified. Moreover, evaluation needs to be more thorough to demonstrate the generality to new explanations (like the one used in Rieger et.al.), which is also a concern shared by Reviewer o4F1.
> >
> > Moreover, I agree with Reviewer o4F1 that experiment and analysis are almost disjoint. Although I agree that the experiments corroborate label efficiency claim from theory, the connection between analysis and experiments is still very flimsy. For instance, theory did not explain why variational is better than Lagrangian or why Lagrangian failed to satisfy explanation constraints as shown in Fig. 4 (right) or explain the trend in rate of reduction of loss with number of training examples.
> >
> > I am updating my score accordingly.
> >
> > [1] Hennig, P., Osborne, M. A., and Kersting, H. P. Probabilistic Numerics: Computation as Machine Learning. Cambridge University Press, 2022. doi: 10.1017/9781316681411. (See chapter 4)

---

> > > ### Author Response · Authors · 2023-08-18
> > >
> > > We are glad to hear that points 2 and 4 are resolved! Regarding your other concerns:
> > >
> > > > I am not yet convinced why explanation constraints are special. This issue was also raised by Reviewer 3fR6, but not yet resolved. Why not treat explanation constraints like any other constraints? It is not hard to see that constraints provide additional supervision, which would improve label efficiency. On the other extreme, supervision on explanations is not that different from label supervision. To elaborate on this, if the underlying hypothesis class is drawn from a GP prior, then observations on function gradient are readily accommodated just as observations on function values [1].
> > > More crisply, what is the significance of the theoretic results in relation to explanations beyond just saying that additional supervision through constraints improved label efficiency.
> > >
> > > Sure, hopefully, this response will help clarify both what we term “explanation constraints” and what is novel about our contributions.
> > >
> > >
> > > We define “explanation constraints” as stochastic constraints on our model that “depend” on the data points. This differs from deterministic constraints that have been considered in existing literature such as l2 regularization. The explanation constraints are closely related to prior works on semi-supervised learning [2] which consider notions of margin or smoothness which are also data-dependent. On the contrary, our paper focuses on the more recent notion of explanations from explainability research. We have already provided this discussion in Appendix A.
> > >
> > > To reason about stochastic constraints, we require using tools from statistical learning theory to work with the constraints in expectation. This idea follows from prior work [2] that we have also cited in the proof sketch of Theorem 2.2.
> > >
> > > We believe that we have sufficient theoretical contributions in our paper. First, new definitions of explanation constraints and surrogate loss provide machinery to formalize how to reason about the random explanation constraints. Given these definitions, we discuss what kind of explanation constraints are learnable (Appendix C) and connect the literature on learning with explanations with the classical work on semi-supervised learning (Thm 2.2). Another contribution of our work is analyzing the special case of explanation constraints that are on the gradient of the model. We provide novel bounds for the reduction in Rademacher complexity for both linear models and two-layer neural networks with these constraints, which both require new proof techniques for these respective proofs.  Framing these definitions and providing new bounds in this different setting, we believe, is a novel contribution.
> > >
> > > [2] Balcan, Maria-Florina, and Avrim Blum. "A discriminative model for semi-supervised learning." Journal of the ACM (JACM) 57.3 (2010): 1-46.
> > >
> > > > To elaborate on this, if the underlying hypothesis class is drawn from a GP prior, then observations on function gradient are readily accommodated just as observations on function values [1].
> > >
> > > In some sense, labeled data can be thought of as a special case of an explanation constraint. However, to analyze this from a learning theoretic perspective, we must distinguish between the two of these to provide generalization bounds. You are completely right that they can both be handled (especially in practice), which is what is done in the Lagrangian regularized methods. This distinction needs to be made from a theoretical perspective for our analysis.
> > >
> > > > Q: Could you please clarify how $\lambda$  is obtained in the Augmented Lagrangian objective? Was $\lambda$ fixed? Was it obtained through alternate max-min steps typical of Lagrangian optimization?
> > >
> > > Lambda is a hyperparameter that is selected via a held-out validation set, where we are selecting based on highest validation accuracy.
> > >
> > > > Q: Variational vs Lagrangian. The advantage of variational over Lagrangian is argued to be only computational. Figure 4 needs to be justified as to why Variational is satisfying explanation constraints better than Lagrangian.
> > >
> > > The advantage of our variational approach over the Lagrangian method is not only computational, and we will clarify this in our revision. Another issue with the Lagrangian approach in our setting is that it does not leverage additional unlabeled data, which is helpful for the downstream task. However, we also argue that solely adding unlabeled data isn’t the only benefit of our approach.
> > >
> > >
> > > We come up with another baseline (Lagrangian + self-training) to determine whether this is the case in Appendix L, and our experiments show that simply using self-training on top of a Lagrangian-regularized model does not maintain the ability of the model to satisfy explanation constraints. Therefore, these experiments support the multiple iterative rounds of projections from our variational approach onto the class of models that satisfy explanation constraints.

---

> > > > ### Author Response · Authors · 2023-08-18
> > > >
> > > > > Datasets are non-standard. Rieger et.al. and Ross et.al. contain many other standard datasets like Decoy-MNIST or ISIC dataset. But the datasets used to report in this paper have not been used before for this problem to the best of my knowledge. The reason for choosing not to evaluate on standard datasets needs to be justified. Moreover, evaluation needs to be more thorough to demonstrate the generality to new explanations (like the one used in Rieger et.al.), which is also a concern shared by Reviewer o4F1.
> > > >
> > > > We provide results on synthetic data, as well as practical experimental settings with gradient constraints. We will try running experiments on these other datasets in the rebuttal period if we have time.
> > > >
> > > > > Moreover, I agree with Reviewer o4F1 that experiment and analysis are almost disjoint. Although I agree that the experiments corroborate label efficiency claim from theory, the connection between analysis and experiments is still very flimsy. For instance, theory did not explain why variational is better than Lagrangian or why Lagrangian failed to satisfy explanation constraints as shown in Fig. 4 (right) or explain the trend in rate of reduction of loss with number of training examples.
> > > >
> > > > We would like to remark that our theory aims to explain many practical methods that use explanation constraints. This includes both our variational methods and the Lagrangian-regularized baseline in Rieger et. al. and Ross et. all. The goal of our bounds is to show that incorporating this information, from a theoretical perspective, benefits us via a reduced sample complexity. As such, we believe that our theory is a valuable contribution that analyzes both existing work and our new method, providing support to many of these existing methods that use explanations.
> > > >
> > > >
> > > > We remark that our theory is not intended to argue that the variational method is better than the Lagrangian. Our variational objective is simply a new objective that we have proposed, and we empirically show on many tasks that it is better and has other computational benefits.

---

### Official Review · Reviewer_o4F1 · 2023-07-06

**Soundness:** 4 excellent
**Presentation:** 4 excellent
**Contribution:** 3 good
**Rating:** 7
**Confidence:** 3

**Summary:**

In this work, the authors theoretically investigate learning with constraints on the gradient of the model. Like other works in a similar vein, this paper focuses on gradient-based explanations and on neural networks. Unlike  related works, however, this paper formulates explanation constraints as being over the data distribution rather than locally to a single input. In this setting, the authors key contributions come from their focus on a learning theoretic analysis of models that satisfy their constraints (called EPAC models) which I find to be significant contribution.

This paper is timely given the current focus on trustworthy AI and an emphasis on explainable AI in proposed legislation. I feel the analysis is simplistic in some places. Additionally, it is unused in the experimental section, and a few key prior works are not mentioned. I do believe this work provides a solid learning theoretic foundation for the budding sub-field of machine learning from explanations (MLX) and for works studying different kinds of learning constraints (e.g., adversarial certification).

I will note that though I have given the paper a borderline score, I will happily update my score to an accept if the authors can substantially address my comments during the rebuttal period.

**Strengths:**


The paper tackles a considerably interesting problem that will certainly have impact on future works and can serve as an important basis for understanding a budding sub-field of machine learning research.

The formulations all appear sound and correct to the best of my knowledge.

The paper is very easy to follow as it is well written. I raise a few minor points about the writing below, but in general I appreciate the high-quality of the paper’s presentation.



**Weaknesses:**

The main drawback of the paper is that the theory and experiments are almost wholly disjointed. The first portion of the paper focuses specifically on theoretical guarantees using standard learning theoretic tools; however, I do note see where any of these tools are experimentally evaluated even in the toy experiments. Moreover, machine learning from explanation (MLX) papers and even robust explanation constraint papers focus on datasets and models that are considerably more sophisticated than what is proposed in this paper.

* EPAC-ERM objective is exactly the same as what is proposed in prior machine learning from explanation (MLX) works, but no discussion is given.
* For completeness the work should cite and at least discuss if not experimentally compare to papers [1][2] and [3] There are several scientifically vague sentences which I find inappropriate without further discussion of formalization. Three in particular come to mind:

Another part of the paper I find lacking is the existence of several vague scientific statements that are not backed up by formal argument or good intuition. I give three examples and my issues below:

“We argue that in many cases, n labeled data are the most expensive to annotate. The k data points with explanations also have non-trivial cost; they require an expert to provide the annotated explanation or provide a surrogate loss.” - Page 6. It seems that for images at least, eliciting explanation annotations from humans is considerably more costly. Moreover, the authors claim that they “argue” the case of having a loss function, the k data points with explanations would be less costly, but I see no such argument. It is not obvious to me that eliciting a high-quality loss function $\phi$ would be any less expensive than eliciting annotations themselves. Happy to hear an argument to the contrary, but on its own this statement feels unscientific and needs to be buttressed with an actual argument. To this end, I would find it very interesting, and I think it would also be interesting to the community working on these problems if a query/sample complexity was explicitly modeled and compared (i.e., the sample complexity of learning a loss function versus the sample complexity of learning a EPAC model from concretely provided labels)

“Computing the gradients of this surrogate loss  in turn is much more expensive compared to gradients of the empirical risk.“ - Page 7. Again, I feel this is a vague and unjustified statement. The gradient of the surrogate loss for explanations is of course more expensive than the standard classification loss, but in previous papers regularization of explanations based on gradients has been done for even ResNet-50 sized models without any hint of computational strain. So, while the general case of the loss might be considerably more difficult to differentiate, in the case of studying gradient-based explanations and their losses dont seem “much more expensive”

“Thus, the decoupled approach to solving the EPAC-ERM objective is in general more computationally convenient. “ - Page 7. I feel this statement would be easy enough to back up experimentally in the experimental setting put forward in this work. One can simply provide the numbers showing this. However, I am curious if this remains the case when the method is scaled up.

[1] Ross et. Al., 2017 https://arxiv.org/abs/1703.03717 (this one is cited but not compared experimentally which I think it ought to be)
[2] Wicker et. Al. 2022 https://arxiv.org/abs/2212.08507 (this ought to be cited, but is in a different vein so no experimental comparison needed)
[3] Czarnecki et. Al. 2017 https://arxiv.org/pdf/1706.04859.pdf (Similar aim, ought to be discussed again perhaps not experimentally compared)

**Questions:**

See the above section.

**Limitations:**

See the weaknesses section

---

> ### Author Rebuttal · Authors · 2023-08-09
>
> > 1. The main drawback of the paper is that the theory and experiments are almost wholly disjointed. The first portion of the paper focuses specifically on theoretical guarantees using standard learning theoretic tools; however, I do note see where any of these tools are experimentally evaluated even in the toy experiments.
>
> **Our response:** We believe that our theory and experiments are complementary. These standard learning theoretic tools are supported by our experiments - we illustrate in Figure 4 that the gradient constraint helps with performance, especially in settings with limited labeled data. This supports our theoretical result, that such models that include an explanation loss achieve lower sample complexity.
>
> > 2. Moreover, machine learning from explanation (MLX) papers and even robust explanation constraint papers focus on datasets and models that are considerably more sophisticated than what is proposed in this paper.
> EPAC-ERM objective is exactly the same as what is proposed in prior machine learning from explanation (MLX) works, but no discussion is given. For completeness the work should cite and at least discuss if not experimentally compare to papers [1][2] and [3] There are several scientifically vague sentences which I find inappropriate without further discussion of formalization. Three in particular come to mind:
>
> **Our response:**  Thank you for pointing this out. We will certaintly provide a more detailed discussion with the MLX literature.
>
> > 3. Another part of the paper I find lacking is the existence of several vague scientific statements that are not backed up by formal argument or good intuition. I give three examples and my issues below:
> “We argue that in many cases, n labeled data are the most expensive to annotate. The k data points with explanations also have non-trivial cost; they require an expert to provide the annotated explanation or provide a surrogate loss.” - Page 6. It seems that for images at least, eliciting explanation annotations from humans is considerably more costly. Moreover, the authors claim that they “argue” the case of having a loss function, the k data points with explanations would be less costly, but I see no such argument. It is not obvious to me that eliciting a high-quality loss function would be any less expensive than eliciting annotations themselves.
>
> **Our response:** Thank you for your great point. The cost of explanation annotations highly depends on how fine-grained you want your explanation constraint to be.
>
> For example, in [1], we just want the model to ignore the background on an image. This is a relatively simple constraint and it is possible to provide a closed-form explanation loss function. For example, if we have access to an image segmentation model. We can use this segmentation model to segment an image and then penalize the feature importance of the background pixel without the need for a human annotator.
>
> However, if we want more complex explanation constraints such as identifying the most important features of each image then we agree that the cost of annotation will be considerably more than the cost of labeled data, and providing a high-quality loss function in this case can be difficult. This would be an interesting venue for future research.
>
> We will provide a better clarification of this in the camera-ready version.
>
>
> > 4. To this end, I would find it very interesting, and I think it would also be interesting to the community working on these problems if a query/sample complexity was explicitly modeled and compared (i.e., the sample complexity of learning a loss function versus the sample complexity of learning a EPAC model from concretely provided labels)
>
> **Our response:** We do provide some discussion about this in Appendix C; here we discuss what types of explanation loss functions are learnable (given finite unlabeled data and “concretely provided labels” in the form of the value of this loss on those data). We also believe that this is an interesting question and that our work provides a preliminary discussion of this. We agree that modeling the complexity of designing/generating this surrogate loss is an interesting question for future work.
>
> > 5. “Computing the gradients of this surrogate loss in turn is much more expensive compared to gradients of the empirical risk.“ - Page 7. ...
>
> **Our response:** We agree that this is computable in practice for ResNet-50 sized models. That being said, the regularization of explanations based on the input gradients still take a comparatively longer time to run than learning with a standard (cross-entropy) loss.  For instance, on synthetic data with 2 layer neural networks, differentiating with respect to the norm of the input gradients is 2.5x slower than using the standard CE loss (for 10 examples over 100 epochs).
>
> > 6. “Thus, the decoupled approach to solving the EPAC-ERM objective is in general more computationally convenient. “ - Page 7...
>
> **Our response:** We provide experimental results in Appendix N.2 that show that smaller teacher models (e.g., smaller number of hidden dimensions) do not degrade the performance of our EPAC-ERM objective, which supports that this decoupled approach can be more computationally convenient. We agree that scaling this up to larger models is indeed an interesting future research agenda.
>
> > 7. [1] Ross et. Al., 2017 (this one is cited but not compared experimentally which I think it ought to be)...
>
> **Our response:** We want to point out that the method in Ross et. al. [1] is the same as the standard Lagrangian baseline in our paper; the explanation surrogate loss function here is the l2 norm of the input gradients (along the features defined in the paper), which exactly matches the second term in the objective of [1]. The only difference is the third term of Ross et. al.’s objective, which is a standard L2 regularization. We will make this clearer in the main text of our paper.

---

> > ### Comment · Reviewer_o4F1 · 2023-08-14
> > **Thank you for the clarifications!**
> >
> > I would like to thank the authors for their very detailed response. I found it quite convincing. After reading the response and re-reading my review, I think my point about the experimental section being disjoint from the theory was too harsh. Moreover, I have had time to more carefully examine the Appendices which I found interesting and comprehensive. I have increased my score with the assumption that in the final version the authors properly cite the MLX/explanation constraint literature and make the changes they say they will. Thanks again :)

---

### Official Review · Reviewer_zYz5 · 2023-07-08

**Soundness:** 3 good
**Presentation:** 3 good
**Contribution:** 3 good
**Rating:** 7
**Confidence:** 2

**Summary:**

Disclaimer: I am not an expert in Learning Theory, but I come from a background of interpretability and explanation model’s decision making. So I am familiar with the methods that this work is trying to describe and some related works. I will review this paper modestly, and hope authors and AC can understand.


In the papers, the authors studied recent methods in regularization of prediction models using explanations. An example of explanation constraints would be enforcing certain properties of the gradients of the objective w.r.t. to the data. Another would be requiring a separate teacher model that takes in the prediction model and data and require the teacher model to have specific desired outputs. Authors analyze these explanation constraints from a statistical learning perspective, such as presenting two different explanation constraints and the analysis of their learnability and generalization bounds. Then, the authors proposed the algorithm for learning with explanation constraints, and a variational method for solving the objective.  Experiments performed on synthetic data and simple real-world data demonstrates that their variational approach achieves better performance than standard approach.


**Strengths:**

The paper is overall well-organized and well-written. The notation seems clear and not cluttered. The formulation and definitions are also not difficult to follow, and the writing describes most of the motivation of formulations well. Since I am not an expert, I cannot comment on the novelty of the work. Nonetheless theoretical analysis of explanation methods or methods that incorporate explanations into training are rare. So this work could potentially be a start to introducing theory in explanation methods. The authors have cited sufficient related works in the appendix. The experiments in this work are solid, where authors compared with multiple baselines and sufficiently discussed the results. The setup of the experiments also seem fair and complete. Appendix also supplies a range of ablation study such as hyperparameters.

**Weaknesses:**

I do not observe any major weaknesses in this work. Although, in terms of writing, the transition into section 4.1 was a bit of a jump. It was not immediately clear to me why there is a need for variational method. Although in experiments it shows to obtain better performance that non-variational approaches, it wasn’t immediately obvious what prompted the authors to develop a variational approach, hence there seems to be a gap between going from Line 234 to line 236.

**Questions:**

1. The authors claim that in Line 29 “An attractive facet of the latter is that we can automatically generate model-based explanations given unlabeled data points.” Can the authors cite works that generate model-based explanations given unlabeled data or clarify what this means?

**Limitations:**

The authors did not address any limitation of their work, but addressed its social impact. One limitation could be that in this work, authors seem to focus on restricting gradients as a explanation constraints. It will be great if future work can also address other versions of explanation constraints, and potential to extensions such as feature selection methods as explanation methods. Analysis of models beyond two-layer neural network regime could potentially be discussed.

---

> ### Author Rebuttal · Authors · 2023-08-09
>
> > I do not observe any major weaknesses in this work. Although, in terms of writing, the transition into section 4.1 was a bit of a jump. It was not immediately clear to me why there is a need for variational method. Although in experiments it shows to obtain better performance that non-variational approaches, it wasn’t immediately obvious what prompted the authors to develop a variational approach, hence there seems to be a gap between going from Line 234 to line 236.
>
> **Our response:** Thank you for the feedback, we will provide an additional discussion in the transition into section 4.1. The main idea is that, especially in instances where computing the surrogate loss function is computationally intensive, the Lagrangian penalized objective may be expensive to optimize. Therefore, we can consider a different objective, through a projection onto a regularized class of models.
> While at first, this projection indeed requires computing the surrogate loss as well, the crucial advantage of this is that we can solve the projection step (this is step 1 in Section 4.1) with a **simpler class** of teacher models for which the explanation surrogate gradients are more tractable, thus resolving much of the computationally difficulty of the Lagrangian objective. We could also use different learning rates and tolerances for the projection step, again with the aim of making the optimization problem more tractable.
>
> > The authors claim that in Line 29 “An attractive facet of the latter is that we can automatically generate model-based explanations given unlabeled data points.” Can the authors cite works that generate model-based explanations given unlabeled data or clarify what this means?
>
> **Our response:** By “automatically generate model-based explanations given unlabeled data points”, we refer to the case when we can evaluate an explanation loss function phi without a domain expert. For example, if we want to encourage the model to not rely on the background features of an image and we have access to an image segmentation model. We can use this segmentation model to segment an image, giving us the background pixels in an image, which then we can penalize the feature importance of the background without the need for a human annotator. Another example presented in the paper is when we want to match the gradient of the model with the gradient of weak labelers (see Section 5.2). Since we have access to these weak labelers apriori, we can evaluate the explanation loss on any unlabeled data.

---

> > ### Comment · Reviewer_zYz5 · 2023-08-12
> > **Response**
> >
> > I thank the authors for their response. The authors have sufficiently addressed my questions and concerns. As of now, I do not have further questions.

---

### Official Review · Reviewer_3fR6 · 2023-07-25

**Soundness:** 3 good
**Presentation:** 2 fair
**Contribution:** 2 fair
**Rating:** 6
**Confidence:** 4

**Summary:**

This paper presents the concept of learning from explanations and theoretically shows that learning from explanations can improve the standard excess risk bounds in the agnostic setting. They treat explanations as constraints in the ERM procedure and introduce an extension of PAC model referred to as EPAC model. The paper introduces formal notions of explanations, EPAC learnability, explanation constraint sets etc. The study analyzes the impact of explanations on model learning using standard learning theoretic tools and characterizes the constraints for linear models and two-layer neural networks. The statistical advantages of explanations stem from their role in constraining the hypothesis class: explanation samples improve the estimation of the population explanation constraint, thereby further restricting the hypothesis class. Further, the paper provides an algorithmic solution based on variational approximation to solve the problem of learning with explanation constraints. The proposed solution works better than other methods like augmented Lagrangian.


**Strengths:**


Strengths:
-------------

1. The paper provides a theoretical framework for studying learning with explanations by formalizing various key notions like explanation functionals, explanation constraint set, EPAC model and EPAC learnability etc.


2. Their analysis provides insights into why learning with explanations could improve the overall excess error of the ERM solution. The key insight is that having explanation constraints in the ERM procedure effectively reduces the size of the model class leading to smaller excess risk upper bounds.


3. The paper also provides an algorithm solution to solve the ERM objective while satisfying the explanation constraints and the empirical results show that the proposed method works better than augmented Lagrangian and other baselines.


**Weaknesses:**

Weaknesses and Questions:
---------------------------------

1. Overall I like the ideas presented in the paper but it was a bit tedious to get exactly what is being done. It first appeared that the paper is proposing a method to learn the explanation functionals i.e. learning a model that can generate explanations by learning from some labeled explanation data. However, the goal is to study how using the explanation information improves the model performance. It would be helpful to improve the presentation to make it clear early.


2. I am not convinced about the notion of explanations introduced. It is assumed that explanations are some vectors. Shouldn’t explanations be human-readable and be in natural language? Then how do you pose the constraint sets in natural language? I might be missing something here as I have not followed explainability research.


3. I didn't quite get why explanation constraints are special? In general, one could put some constraints (e.g. regularization, etc.)  along with the ERM objective to reduce the size of the hypothesis class, which will lead to a reduction in the upper bound of the excess risk. Could you please provide some justification or motivation for why the explanation constraints are special?


4. Figure 2. could be improved with some annotations and markers instead of just colors alone. The figures in experiments might  be better when run with more random seeds.


**Questions:**

Please see above.

**Limitations:**

Yes.

---

> ### Author Rebuttal · Authors · 2023-08-09
>
> > 1. Overall I like the ideas presented in the paper but it was a bit tedious to get exactly what is being done. It first appeared that the paper is proposing a method to learn the explanation functionals i.e. learning a model that can generate explanations by learning from some labeled explanation data. However, the goal is to study how using the explanation information improves the model performance. It would be helpful to improve the presentation to make it clear early.
>
> **Our response:** Our paper does address both of these topics: (1) we provide a mathematical framework to analyze how explanation information improves model performance and (2) we propose a variational objective that can learn explanations through explanation-labeled data. We would also like to point out that other reviewers thought that “the paper is organized and well-written”(Reviewer zYz5), “The paper is very easy to follow as it is well written”(Reviewer o4F1), “Well written and easy to follow”(Reviewer w12b), and “The presentation of this paper is clear” (Reviewer 9kMr).
>
> > 2. I am not convinced about the notion of explanations introduced. It is assumed that explanations are some vectors. Shouldn’t explanations be human-readable and be in natural language? Then how do you pose the constraint sets in natural language? I might be missing something here as I have not followed explainability research.
>
> **Our response:**  While some explanations may be in natural language, in this work, we focus on the notion of local explanations considered in explainability research where explanations indeed take a vector form. For example, feature attribution methods [1,2,3] indicate how much each feature contributed to the model predictions, and these are all represented via real-valued vectors not in language.
>
>
> [1] Ribeiro, Marco Tulio, Sameer Singh, and Carlos Guestrin. "" Why should i trust you?" Explaining the predictions of any classifier." Proceedings of the 22nd ACM SIGKDD international conference on knowledge discovery and data mining. 2016.
>
> [2] Lundberg, Scott M., and Su-In Lee. "A unified approach to interpreting model predictions." Advances in neural information processing systems 30 (2017).
>
> [3] Sundararajan, Mukund, Ankur Taly, and Qiqi Yan. "Axiomatic attribution for deep networks." International conference on machine learning. PMLR, 2017.
>
> > 3. I didn't quite get why explanation constraints are special? In general, one could put some constraints (e.g. regularization, etc.) along with the ERM objective to reduce the size of the hypothesis class, which will lead to a reduction in the upper bound of the excess risk. Could you please provide some justification or motivation for why the explanation constraints are special?
>
> **Our response:**  We completely agree that our framework is much more general and can handle other notions of constraints - please see our Discussion section (line 326 - 329).
>
> In our work, we have drawn inspiration from the field of explainability where such constraints are used to train models [4,5]. The explanation constraints are provided by domain experts for which the constraint depends on the instance. For example, if we want to ignore the background of images, different images would have different constraints. This differs from standard constraints such as L2 regularization which depends only on the model parameters. We provide a specific bound for a gradient constraint which is a canonical class of explanations in the field.
>
> As you mentioned (and as we also say in our Discussion section), this framework and the result from Theorem 2.2 holds for any notion of constraints. It is important, however, that there are models that satisfy these constraints that are also able to achieve high standard accuracy.
>
> [4] Ross, Andrew Slavin, Michael C. Hughes, and Finale Doshi-Velez. "Right for the right reasons: training differentiable models by constraining their explanations." Proceedings of the 26th International Joint Conference on Artificial Intelligence. 2017.
>
> [5] Rieger, Laura, et al. "Interpretations are useful: penalizing explanations to align neural networks with prior knowledge." International conference on machine learning. PMLR, 2020.
> >4. Figure 2. could be improved with some annotations and markers instead of just colors alone. The figures in experiments might be better when run with more random seeds.
>
> **Our response:**  Thank you for your suggestion. We will update the figure in the camera ready with some more descriptive annotations.

---

> > ### Comment · Reviewer_3fR6 · 2023-08-11
> > **Response to author rebuttal**
> >
> > Dear authors,
> >
> > Thank you for responding to my questions. I think my concerns are not adequately addressed. My point 3 was to get better understanding of the theoretical results. They seem to be valid for any constrained ERM. How do we interpret them in the context of explanation constraints? I am looking for a bit more nuanced understanding of these results from explanations point of view. I am also not convinced about why these results are novel or interesting. In general one could put constraints on ERM objective (e.g. regularization etc.) and that will lead to reduction of hypothesis class and hence smaller upper bounds. In these cases the results are more refined, providing more insights on constraint parameters and how they affect the bounds. In this regard, I don't find Theorem 2.2 satisfactory.
> >
> > Do you have any lower bounds on the generalization errors to claim that learning with explanations is helpful? The current conclusions are based on comparing two upper bounds on the generalization errors.
> >
> > In my view, the presentation could be improved to make it easier to follow. Here are some of my suggestions,
> >
> > a. Please include some intuitive examples of explanation constraints early in the paper.
> >
> > b. May be have a few figures, showing these constraints and how they affect the learning procedure.
> >
> > c. Provide a brief summary of the explanation constraints considered in the literature and why they can be abstracted out mathematically the way it is done in the paper.

---

### Author Rebuttal · Authors · 2023-08-09

**General response**


We would like to thank all the reviewers for their efforts in providing thoughtful and attentive examinations of our work. We are glad to see that the reviewers highlighted a number of strengths, including:
1. The paper tries to answer an “interesting”(o4F1) and “important” problem (9kMr, w12b)
2. The paper has a “good contribution” (9kMr) and could potentially be “a start to introducing theory” in explanation methods (zYz5) and can “serve as an important basis” for understanding the subfield (o4F1)
3. The paper is “well-written” (zYz5, o4F1, w12b) and has a “clear presentation” (9kMr)
4. The proposed method “works better than other baselines” (3fR6) and the setup of the experiments also seems “fair and complete” (zYz5).

We respond to individual reviewer comments in the individual threads. Thank you again for your hard work and consideration.

---

### Decision · Program_Chairs · 2023-09-21

**Decision:**

Accept (poster)

**Comment:**

This paper joins an interesting area where additional information can be used on top of a labeled dataset for supervised learning. It proposes a framework for learning from information in the form of a type of explanation---here mathematically modeled as functionals. It provides learning-theoretic results, algorithms, and reasonably varied and convincing empirical results.

This paper formalizes notions that are often used casually and provides a comprehensive analysis. Such papers are often useful to clarify concepts and approaches that are only touched on intuitively. The reviewers agreed and largely found the work compelling.